# Endowing GPT-4 with a Humanoid Body: Building the Bridge Between Off-the-Shelf VLMs and the Physical World

**Yingzhao Jian, Zhongan Wang, Yi Yang & Hehe Fan\***
College of Computer Science and Technology, Zhejiang University. *Corresponding author.
`hehefan@zju.edu.cn`

## Abstract

Humanoid agents often struggle to handle flexible and diverse interactions in open environments. A common solution is to collect massive datasets to train a highly capable model, but this approach can be prohibitively expensive. In this paper, we explore an alternative solution: empowering off-the-shelf Vision-Language Models (VLMs, such as GPT-4) to control humanoid agents, thereby leveraging their strong open-world generalization to mitigate the need for extensive data collection. To this end, we present **BiBo** (**B**uilding humano**I**d agent **B**y **O**ff-the-shelf VLMs). It consists of two key components: (1) an **embodied instruction compiler**, which enables the VLM to perceive the environment and precisely translate high-level user instructions (e.g., *"have a rest"*) into low-level primitive commands with control parameters (e.g., *"sit casually, location: (1, 2), facing: $90°$"*); and (2) a diffusion-based **motion executor**, which generates human-like motions from these commands, while dynamically adapting to physical feedback from the environment. In this way, BiBo is capable of handling not only basic interactions but also diverse and complex motions. Experiments demonstrate that BiBo achieves an interaction task success rate of 90.2% in open environments, and improves the precision of text-guided motion execution by 16.3% over prior methods. The code is available at https://github.com/Shadow-Dream/BiBo.

## 1 Introduction

Humanoid agents, as a medium between digital intelligence and the physical world, have attracted extensive research interest, particularly in the domains of scene perception (Huang et al., 2024b; Qi et al., 2025b) and interaction (Xiao et al., 2023; Tevet et al., 2024). With recent advances, humanoid agents are capable of performing text-guided motions (Tevet et al., 2022; Yuan et al., 2024) and executing interactive tasks under predefined plans (Xu et al., 2024; Pan et al., 2025). However, flexibly handling user-intended scene interactions in open and dynamic physical environments remains a significant challenge. A straightforward strategy is to collect large-scale human–scene interaction data (Bhatnagar et al., 2022; Jiang et al., 2024) and train a highly capable model, as commonly done for robotic arms or wheeled platforms (Firoozi et al., 2025; Team et al., 2025). Unfortunately, due to the structural complexity of humanoid bodies and the vast diversity of physical world, such data-centric scaling becomes prohibitively expensive and difficult to generalize.

In contrast, off-the-shelf general-purpose Vision–Language Models (VLMs), such as GPT-4 (Achiam et al., 2023), Gemini (Team et al., 2023), and Qwen (Bai et al., 2023), demonstrate open-world reasoning and adaptability across a wide variety of tasks, without specific finetuning. This observation naturally raises an intriguing question: *Can we bypass costly data collection by directly leveraging these powerful off-the-shelf VLMs to control humanoid agents, thereby enabling more versatile interaction in the physical world?*

Motivated by this question, we introduce BiBo (**B**uilding humano**I**d agent **B**y **O**ff-the-shelf VLMs), a framework designed to endow off-the-shelf VLMs with the capability to control humanoid agents. BiBo is composed of two core components: 1) an **VLM-driven embodied instruction compiler** and 2) a **diffusion-based motion executor**, which jointly bridge the gap between high-level human

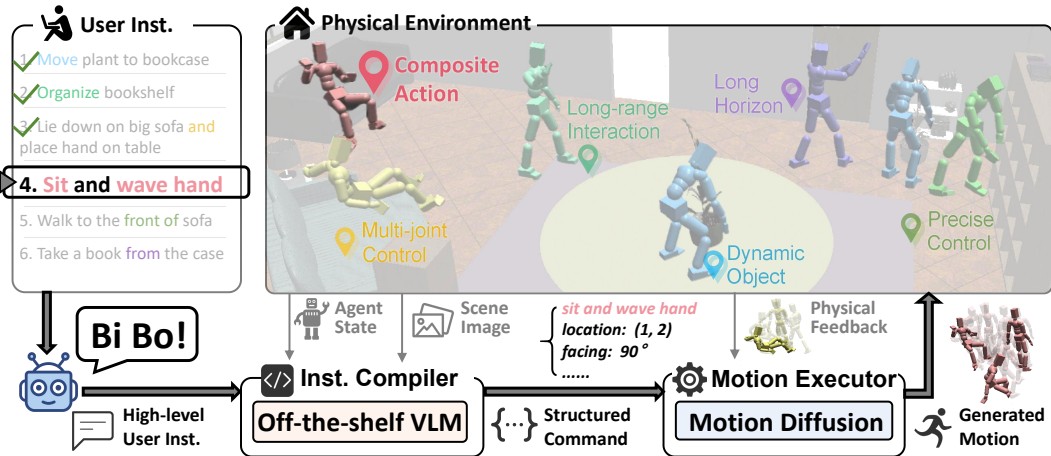

Figure 1: **BiBo** is a humanoid agent powered by an off-the-shelf VLM. It consists of an embodied instruction compiler (Inst. Compiler) and a diffusion-based motion executor. When the user provides a high-level instruction, the compiler observes the environment and translates it into the structured command for the executor. The executor then generates future motions for the humanoid agent, conditioned on both the command and the physical feedback from the environment. In this way, BiBo is able to perform diverse types of physical scene interactions.

instructions and low-level motor control required for physical humanoid interactions. This design is conceptually analogous to a computer, where the compiler and the assembler work together, operating the hardware to accomplish the tasks specified by programming language.

In computing system, a compiler translates source code written by high-level programming languages into low-level assembled language. Inspired by this, as in Fig. 1, the embodied instruction compiler in BiBo is designed to translate high-level natural language instructions into low-level executor commands, according to environmental context. To achieve this, the compiler first represents an action as a structured set of descriptors, encompassing the motion caption, key joint configurations, and other contextual details. Building on this structured representation, the compiler drives the VLM to reason over each descriptor in a coarse-to-fine manner, thereby producing an accurate and structured command that specifies the user intended action.

This generated command is then passed to the motion executor, which functions much like an assembler. Just as translating assembly commands into machine code, the motion executor interprets commands into full-body humanoid motions. Unlike a rigid rule-based assembler, our executor leverages a diffusion generator. Each time it receives a command, the generator extends future joint trajectories from the current motion, enabling diverse motion style and on-the-fly control.

However, during execution, collisions or external forces may cause the actual executed motion to deviate from the initially generated sequence. To handle such feedback, prior approaches (Tevet et al., 2024; Chen et al., 2024) extending future joint trajectories from the executed motion, rather than from those previously generated but unexecuted. This strategy enforces the diffusion to account for environmental context, but also introduces discontinuities between the current and previous generated motions. We resolve this by developing a novel application of the Latent Diffusion Model (LDM) (Chen et al., 2023). In our method, the diffusion extends future latent from the actual executed motion, enabling environmental awareness. A Variational Autoencoder (VAE) jointly decodes the latents of the previous and current generated motions, ensuring smooth transition.

According to the experiments, BiBo achieves an interaction task success rate of 90.2% under random generated physical environments using an off-the-shelf VLM (i.e., GPT-4o). Moreover, BiBo improves the precision of text-guided motion execution by 16.3% over prior methods. It handles complex motion execution, while also enabling infinite long-sequence synthesis and real-time interactive control through user instructions. In summary, our main contributions are threefold:

- We empower off-the-shelf VLMs for humanoid control through an embodied instruction compiler and a diffusion-based motion executor, bridging the gap between general-purpose VLM reasoning and low-level physical execution.

- The compiler introduces a structured humanoid action representation, enabling coarse-to-fine embodied reasoning, while advancing humanoid behavior planning and modeling.

- We develop a novel application of LDM to incorporate environmental feedback in motion generation, achieving state-of-the-art unlimited-length motion synthesis and offering insights for applying LDMs in broader domains.

## 2 RELATED WORK

### 2.1 HUMAN SCENE INTERACTION

When interacting with scene, humanoids perceive environments through training on real-world data, reinforcement learning (RL), and large language models (LLMs). Recent advances combine them: (1) using LLM to guide RL policies (Xiao et al., 2023; Shi et al., 2024), but still limits motion diversity; (2) using RL trackers (Luo et al., 2022; 2023) to follow diffusion-generated motions (Tevet et al., 2024), but causes discontinuities between generated and tracked motions. BiBo employs an off-the-shelf VLM to guide a latent diffusion model (LDM), promoting generalization and diversity, which achieve both smoothness and physical plausibility.

### 2.2 TEXT TO MOTION GENERATION

In text-to-motion, approaches can be broadly categorized into fixed- and arbitrary-length generation. For fixed-length generation, frameworks such as VAEs (Petrovich et al., 2021b; Bie et al., 2022), masked modeling (Pinyoanuntapong et al., 2024; Guo et al., 2024), and diffusion models (Tevet et al., 2022; Zhang et al., 2024; Dai et al., 2024; Chen et al., 2025) have been extensively explored. However, humanoids perform continuous arbitrary-length motion following user commands. To this end, some works adopt autoregressive next-token prediction (Jiang et al., 2023; Zhang et al., 2023), achieving high fidelity, while others (Chen et al., 2024; Han et al., 2024; Xiao et al., 2025) employ diffusion to extend future joint trajectories from past motion, improving efficiency. BiBo use latent diffusion with few denoising steps, enabling both high-fidelity generation and real-time control.

## 3 METHOD

### 3.1 OVERVIEW

BiBo is a humanoid agent powered by an off-the-shelf Vision-Language Model (VLM). As shown in Fig. 1, it comprises two main components: an embodied instruction compiler and a diffusion-based motion executor. Given a high-level user instruction, the compiler first collects observations of the current scene, and then prompts the VLM to generate a caption of the next motion to be executed. Next, the compiler guides the VLM to refine motion details through a three-stage visual question-answering (VQA) process. Finally, it formats these details into a command based on a structured motion representation, thereby instructing the executor to generate the corresponding motion.

The executor takes the command as a condition, extending future joint trajectories from the current motion. Due to collisions or external forces, the actual executed motion may differ from the predicted trajectories. To adapt to this feedback, we incorporate the actual performed motion into diffusion by developing a novel application of the Latent Diffusion Model (LDM). In diffusion, the model extends future motion latents from the actual executed motion, thereby adapting to the scene feedback; in the VAE, the model jointly decodes the previously generated and currently executed motions, ensuring smooth transitions between previous and current generated motions.

### 3.2 EMBODIED INSTRUCTION COMPILER

As in Fig. 2, the embodied instruction compiler enables the VLM to translate high-level user instructions into low-level executor commands, based on environmental observation. It consists of a three-stage visual question–answering process. The VLM first determines the next motion to execute, and analyzes its basic attributes, such as the motion caption and target object. Then, it reasons about the agent pose. Finally, the VLM locates the target positions of the key joints. The output of

Figure 2: The **embodied instruction compiler** takes in user instructions and environmental observations, and directs the VLM to generate the next motion command through a structured three-stage visual question–answering process. In the first stage, it analyzes the basic attributes of the motion (e.g., caption, key joints, target object). In the second stage, it reasons about the agent's pose during the interaction. Finally, it specifies the target positions for the key joints.

the compiler is a executor command $\mathcal{C}$, including the caption $c$, location $\boldsymbol{l} \in \mathbb{R}^2$, facing direction $f \in [-\pi, \pi]$, and joint targets $\mathbb{J} = \{(j, \boldsymbol{p}_j) : j \text{ is a key joint}\}$, where $\boldsymbol{p}_j \in \mathbb{R}^3$ is the joint target:

$$\mathcal{C} = \{c,\ \boldsymbol{l},\ f,\ \mathbb{J}\}. \tag{1}$$

The command $\mathcal{C}$ serves as a structured and simplified humanoid action representation, which reduces generation complexity while preserving the key information of a motion, controlling the diffusion to generate an interactive motion that fulfills the instruction. Details can be found in Sec. C.

**Basic Attribute Analysis.** In this step, the compiler inputs the user instruction with scene images and agent's status, and prompts the VLM to analyze the attributes of the next motion to be executed. These attributes include a motion caption $c$, an anchor object $o$ for agent localization, the key joints $j$ involved, and other complementary details that facilitate subsequent reasoning. To enhance accuracy, the final result is selected through majority voting across five parallel VLM instances.

**Agent Pose Reasoning.** The pose refers to the location $\boldsymbol{l}$ and facing direction $f$ of the agent. To predict the pose, directly outputting coordinates and angles is a straightforward way. However, current off-the-shelf VLMs struggle to handle numbers like 3D coordinates (Huang et al., 2024a; Qi et al., 2025a). As a result, we transform it into a visual identification task, which is more familiar to VLMs. As in Fig. 2, we put labels around the anchor object $o$, each corresponds to a position or direction. By choosing a label from the image, the VLM roughly locates the agent in the scene.

**Key Joint Generation.** When interacting with objects, we typically focus on a few key joints and their relative position to specific target points (e.g., when using a hand dryer, the hands are placed about 0.2m beneath the air outlet). Inspired by this, for each key joint, we first place a $8 \times 8$ grid of labels over the image of the anchor object, and allow the VLM to select one as the target point. Next, the VLM generates the joint's direction and distance relative to the target point. We provide a set of predefined directions: *[up, down, left, right, forward, backward, toward the object center, along the surface normal]*. This simplification reduces generation difficulty while covering most cases.

## 3.3 MOTION DIFFUSION EXECUTOR

The Motion Diffusion Executor is a Latent Diffusion Model, composed of a VAE and a Diffusion module. When command $\mathcal{C}$ comes, the VAE first encodes both the previously generated motion $\boldsymbol{M}_g \in \mathbb{R}^{F \times D}$ and the actual executed motion $\boldsymbol{M}_a \in \mathbb{R}^{F \times D}$ (i.e. the execution result of $\boldsymbol{M}_g$ under physical environment) into latent representations $\boldsymbol{S}_g \in \mathbb{R}^{L \times H}$ and $\boldsymbol{S}_a \in \mathbb{R}^{L \times H}$, respectively:

$$\boldsymbol{S}_a = \text{Encoder}(\boldsymbol{M}_a), \quad \boldsymbol{S}_g = \text{Encoder}(\boldsymbol{M}_g). \tag{2}$$

$F$ is the number of frames of the motion. $L$ is the number of latent tokens, where each token correspond to $\frac{F}{L}$ frames. $D$ and $H$ are the dimension of motion frame and latent. Then, the command $\mathcal{C}$, together with the latent of the executed motion $\boldsymbol{S}_a$ guide the denoising process to produce the latent of future motion $\boldsymbol{S}_f \in \mathbb{R}^{L \times H}$. The latents of the previous and current generated motion $\boldsymbol{S}_g$ and $\boldsymbol{S}_f$ are jointly decoded by the VAE to obtain the future joint trajectories $\boldsymbol{M}_f \in \mathbb{R}^{F \times D}$:

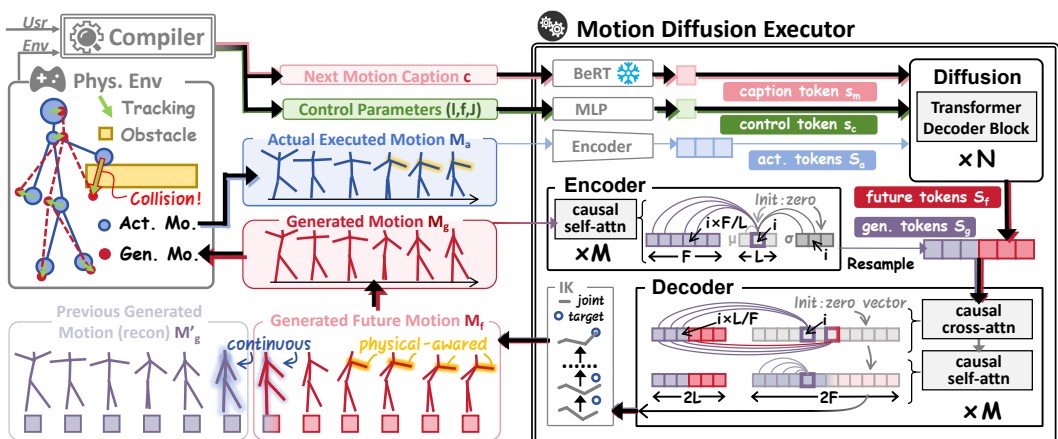

Figure 3: The **motion executor** is a Latent Diffusion Model. When receiving the command (motion caption and control parameters) from the compiler, the Diffusion extends the future latents $S_f$ from the actual executed motion tokens $S_a$, conditioned on the command tokens $s_m$ and $s_c$. Then, the previous and newly generated latents are jointly decoded by the VAE decoder. The decoder use casual attention, where each motion frame or latent token can only attend to its preceding tokens or frames. After IK optimization, a tracking policy drive humanoid joints to execute the newly generated motion $M_f$ in physical environment, producing the next execution result.

$$S_f = \text{Diffusion}(\mathcal{C}, S_a), \quad [M_g' : M_f] = \text{Decoder}([S_g : S_f]). \tag{3}$$

$S_f \in \mathbb{R}^{L \times H}$ is the generated latent of future motion. $M_g' \approx M_g \in \mathbb{R}^{F \times D}$ is the reconstructed previous generated motion. [:] represents the concatenation across length (or frame) dimension. By conditioning on $S_a$, the future motion $M_f$ is guided to account for physical feedback in $M_a$, while joint decoding enforces its continuity with $M_g$.

We additionally use inverse kinematics (IK) post-optimization to improve control accuracy. Finally, a reinforcement learned tracking policy drives humanoid joints to follow the generated joint trajectories in physical scene. For more details, please refer to Sec. D.5.

**VAE Design.** We use Transformer encoder-decoder architecture and propose causal self-cross attention. Specifically, during the attention process, each latent token or motion frame can only attend to its preceding tokens or frames. This design ensures the continuity between the previous and current generated motion $M_g$ and $M_f$. Specifically, as in Eq. 3, since the VAE ensures continuity in its decoded motions, the the generated future motion $M_f$ is continuous with reconstructed previous motion $M_g'$. Moreover, due to the causal mechanism:

$$[M_g' : M_f] = \text{Decoder}([S_g : S_f]) \;\Rightarrow\; M_g' = \text{Decoder}(S_g) \;\Rightarrow\; M_g' \approx M_g. \tag{4}$$

$M_f$ is continuous with $M_g'$, and $M_g' \approx M_g$. Therefore, $M_f$ is also continuous with $M_g$.

## 4 EXPERIMENTS

In this section, we analyze the capabilities of BiBo from two perspectives: task completion and motion quality. For task completion, we establish a set of tasks, and assess the success rate of BiBo and comparison methods under random generated scenes. For motion quality, we adopt standard motion quality metrics to quantitatively evaluate the synthesized and executed motions, and conduct case studies with visualizations to analyze the qualitative results. For more details and results, please refer to Sec. E and F.

### 4.1 TASK COMPLETION

**Task Setting.** We define six types of single interaction tasks. Task success if the required criterion is met for over 30 frames. The tasks include:

• *Reach* is considered successful if the agent reaches within 0.5 meters of the target location.

Table 1: Comparison of **task success rates** for different methods under randomly generated scenes and initial poses. A single task involves navigating to the interaction position and performing the interaction, whereas a composite task consists of multiple simultaneous or sequential single interactions. BiBo (our) performs online planning during evaluation, while other methods use ground truth action plan. The **bold** and underline represent the best and second-best performance, respectively. BiBo achieves the highest success rate across all tasks.

| Method (%) | Single Interaction | | | | | | Composite Task | | |
|---|---|---|---|---|---|---|---|---|---|
| | Reach ↑ | Watch ↑ | Sit ↑ | Sleep ↑ | Touch ↑ | Lift ↑ | Simple ↑ | Medium ↑ | Hard ↑ |
| UniHSI(Xiao et al., 2023) | 93.28 | - | 81.03 | 85.11 | 69.62 | - | - | - | - |
| HumanVLA (Xu et al., 2024) | 56.58 | - | - | - | - | 44.90 | - | - | - |
| TokenHSI (Pan et al., 2025) | 94.55 | - | 72.95 | 33.33 | - | 48.19 | - | - | - |
| CLoSD (Tevet et al., 2024) | 85.83 | 87.76 | 76.99 | 34.67 | 42.55 | 7.71 | 26.47 | 7.05 | 2.38 |
| BiBo (ours) | **99.18** | **99.62** | 95.84 | **94.89** | 86.05 | 65.42 | 58.82 | 36.54 | 27.78 |
| BiBo (ours, GT plan) | 98.91 | 99.06 | **96.75** | 93.33 | **87.23** | **70.41** | **61.76** | **44.23** | **42.86** |

• *Watch* evaluates the understanding of object orientation. It success if the agent stands in front of the target and facing to it, with an angular error of less than $\pi/6$.

• *Sit & Sleep* evaluate interaction with objects of varying shapes and functions. Sit success if the hips are within 0.1m of the seat area, subject to an upward force, and the torso faces forward. Sleep success if the legs, arms, and torso are all within 0.1m of the sleep area, with the average force directed upward and the torso facing upward. The angular errors should be less than $\pi/4$.

• *Touch* is successful if the correct hand is within 0.1m of the target, and subject force.

• *Lift* evaluates dynamic object manipulation. Success if the target being lifted more than 0.25m above the ground.

We additionally introduce composite tasks, each consisting of multiple interactions. They succeeds if all interactions are accomplished in the specified order. Composite tasks are categorized into simple, medium and hard. **1) Simple** tasks consist of $< 4$ interactions. **2) Medium** tasks include user intent understanding (e.g., the room is too dark) and dynamic object manipulation (i.e., lift, transport). **3) Hard** tasks involve long-horizon ($> 10$ interactions) and simultaneous interactions with multiple objects (e.g., sit on sofa and put a hand on the side table).

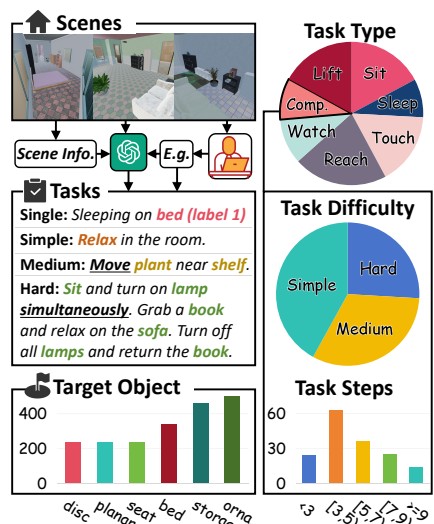

Figure 4: Summary of the random generated scene dataset. The tasks are constructed by a semi-automatic approach. The dataset contains various object categories, task types and difficulties, evaluating a wide range of interaction abilities of humanoid agents.

**Dataset.** To evaluate BiBo's ability to accomplish different tasks in physical environments, we randomly construct 100 scenes using InfiniGen (Raistrick et al., 2023). These scenes include different rooms and layouts, containing 73 object categories with randomized parameters controlling their shapes and styles. For each scene, we construct 6–18 single interaction and 1–3 composite tasks by a semi-automatic approach. Specifically, as in Fig. 4, volunteers first annotate task examples, after which the LLM generates tasks conditioned on the examples and scene information. This process results in a total of 1,365 single tasks and 162 composite tasks. All tasks are given through natural language, and evaluated three times under different initialization. Details can be found in Sec. B.1.

**Comparison Method.** As comparison methods, UniHSI (Xiao et al., 2023) focuses on contact interactions, HumanVLA (Xu et al., 2024) follows the VLA paradigm to address transportation tasks, TokenHSI (Pan et al., 2025) trains task tokens to manage multiple skills, and CLoSD (Tevet et al., 2024) employs motion diffusion, enabling diverse scene interactions. As these methods lack an applicable planner, they use a programmatically generated ground-truth plan, with details provided in Sec. E. For BiBo, we evaluated both the ground truth plan and online planning.

**Result.** As shown in Tab. 1, BiBo achieves an average success rate of $90.2\%$ in single interaction, and $41.0\%$ in composite task. Compared with other methods, BiBo achieves an average improvement of $12.5\%$ and $29.1\%$ on single interaction and composite tasks, respectively. In online planning, BiBo achieves success rates close to those of the ground-truth plan (within $4.38\%$).

Table 2: Impact of different components in compiler and executor on the task success rates. Act. and Gen. represents the actual executed motion and previous generated motion. The **bold** and underline represent the best and second-best performance, respectively. The results demonstrate the effectiveness of designs in BiBo.

| Method (%) | Single Interaction | | | | | | Composite Task | | |
|---|---|---|---|---|---|---|---|---|---|
| | Reach ↑ | Watch ↑ | Sit ↑ | Sleep ↑ | Touch ↑ | Lift ↑ | Simple ↑ | Medium ↑ | Hard ↑ |
| BiBo (ours, w/o Voting) | **99.18** | 98.87 | 91.13 | 88.67 | 85.82 | 59.75 | 49.51 | 32.05 | 22.22 |
| BiBo (ours, w/o Label) | 97.00 | 90.21 | 48.59 | 46.44 | 64.89 | 58.73 | 26.96 | 17.31 | 7.94 |
| BiBo (ours, w/o Act.) | 98.09 | 98.87 | 84.18 | 73.78 | 81.80 | 28.34 | 28.43 | 16.02 | 10.32 |
| BiBo (ours, w/o Gen.) | 98.64 | 99.25 | 95.62 | 93.78 | 84.40 | 56.58 | 57.35 | 30.77 | 23.81 |
| BiBo (ours, w/o IK) | 98.82 | 99.25 | **95.96** | 92.89 | 48.94 | 6.80 | 31.37 | 13.46 | 3.17 |
| BiBo (ours) | **99.18** | **99.62** | 95.84 | **94.89** | **86.05** | **65.42** | **58.82** | **36.54** | **27.78** |

Table 3: Comparison between **text-to-motion** methods on the HumanML3D. We evaluates efficiency, motion quality, and physical plausibility. ↑, ↓ and → means the higher, smaller and closer to the ground truth is preferred. Arbi. and Phys. indicate the support for arbitrary-length and physical plausible generation. The **bold** and underline represents the best and second-best performance.

| Method | Efficiency | Motion Quality | | | | | | Physical Plausibility | | |
|---|---|---|---|---|---|---|---|---|---|---|
| | AITS ↓ | Arbi. | FID ↓ | R.P.@1 ↑ | R.P.@2 ↑ | R.P.@3 ↑ | Diversity → | Phys. | Pen. ↓ | Float ↓ Skate ↓ |
| *Ground Truth* | - | - | 0.001 | 0.514 | 0.706 | 0.800 | 9.503 | ✗ | 0.00 | 22.9 0.21 |
| MDM (Tevet et al., 2022) | 24.74 | ✗ | 0.423 | 0.406 | 0.603 | 0.719 | 9.559 | ✗ | 0.15 | 28.6 0.33 |
| MotionLCM (Dai et al., 2024) | **0.042** | ✗ | 0.072 | 0.510 | 0.703 | 0.797 | 9.598 | ✗ | 0.65 | 27.4 0.81 |
| MotionStreamer (Xiao et al., 2025) | 2.584 | ✓ | 0.084 | 0.432 | 0.615 | 0.716 | **9.549** | ✗ | 0.17 | 23.2 0.68 |
| MoGenTS (Yuan et al., 2024) | 0.836 | ✗ | **0.046** | 0.521 | 0.713 | 0.804 | 9.617 | ✗ | 0.80 | 26.2 0.96 |
| MoConVQ (Yao et al., 2024) | - | ✗ | 3.279 | 0.309 | 0.504 | 0.614 | 8.010 | ✓ | 0.25 | 32.0 0.29 |
| DiP (Tevet et al., 2024) | 0.257 | ✓ | 0.210 | 0.466 | 0.653 | 0.754 | 9.570 | ✗ | **0.14** | 24.5 0.65 |
| CLoSD (Tevet et al., 2024) | 2.873 | ✓ | 2.861 | 0.367 | 0.553 | 0.665 | 8.256 | ✓ | 0.30 | **20.2** **0.01** |
| BiBo (ours) | 0.047 | ✓ | 0.076 | **0.542** | **0.738** | **0.829** | 9.606 | ✗ | 0.32 | 25.3 0.74 |
| BiBo (ours, Phy.) | - | ✓ | 1.883 | 0.411 | 0.604 | 0.716 | 8.298 | ✓ | 0.19 | 20.6 **0.01** |

Experimental results demonstrate that effectiveness of designs in BiBo, revealing the potential of general-purpose VLMs in controlling humanoids.

**Ablation Study.** We conduct ablation studies to validate the proposed designs in BiBo. Specifically, in the compiler, we introduce a voting mechanism, and leverage image labels to facilitate the visual reasoning. In the executor, we generate future motion condition on the actual executed motion and previous generated motion, and apply IK for precise control. The results in Tab.2 show that Voting and Label improve task success rate by $4.1\%$ and $22.9\%$. IK plays a crucial role in precise joint control, while executed and generated motions affect tasks that rely on physical feedback and precise interaction.

Table 4: Comparison of **control accuracy** across methods using MAE. ↓ means smaller is better. **Bold** is the best performance.

| Method | Head ↓ | Hand ↓ | Foot ↓ |
|---|---|---|---|
| DiP (Tevet et al., 2024) | 0.0663 | 0.0830 | 0.0540 |
| MotionLCM (Dai et al., 2024) | 0.0952 | 0.1470 | 0.0955 |
| BiBo (ours) | **0.0310** | **0.0571** | **0.0335** |
| BiBo (ours, w/o LDM) | 0.0705 | 0.0918 | 0.0608 |

## 4.2 MOTION QUALITY

**Setting.** We evaluate the motion quality from both quantitative and qualitative perspectives. On the quantitative side, we report Average Inference Time per Sequence (AITS) reflects the computational overhead (Chen et al., 2023); Fréchet Inception Distance (FID), R Precision and Diversity evaluate the motion fidelity and diversity (Guo et al., 2022b); Penetration, Float, and Skate reflect the physical plausibility (Yuan et al., 2023); Mean Absolute Error (MAE) evaluates the control precision. Details can be found in Sec. B.2.3. On the qualitative side, we conduct human evaluations and visual inspections, and further perform case studies that analyze both successful and failure cases.

**Dataset.** We adopt the HumanML3D dataset. Following common practice, we use motion sequences whose lengths fall inside $[40, 200]$ frames. A total of 24,545 motion episodes paired with 66,633 motion captions from the training split are used to train the motion diffusion executor, while 4,646 motion episodes paired with 12,536 captions from the test split are reserved for evaluation.

**Comparison Method.** We adopt state-of-the-art methods for both fixed-length and arbitrary-length generation, including physical and non-physical approaches. Details are provided in the Sec. E.

**Quantitative Result.** As in Tab. 3, BiBo handles on-the-fly control ($> 20$Hz), and demonstrates advantages (non-physical $+3.5\%$ and physical $+7.3\%$ relatively) in text alignment (R.P.) across comparison methods. It improves motion realism (FID) in real-time arbitrary-length generation

Table 5: **Motion discontinuity** evaluated by average joint acceleration $\bar{a}$.

| Method $(m^2/s^2)$ | $\bar{a}$ |
|---|---|
| CLoSD (Tevet et al., 2024) | 0.0610 |
| BiBo (ours, w/o LDM) | 0.0879 |
| BiBo (ours, w/o Causal) | 0.0626 |
| BiBo (ours, w/o Gen.) | 0.0698 |
| BiBo (ours, w/o Act.) | **0.0370** |
| BiBo (ours) | 0.0379 |

Table 6: Number of preferred motions or interactions in **User study**.

| Method | Count |
|---|---|
| DiP(CLoSD) (Tevet et al., 2024) | 20 |
| MotionLCM (Dai et al., 2024) | 53 |
| BiBo (ours) | **77** |

Table 7: Impact of different components in BiBo on motion quality, evaluated using HumanML3D. ↑ and ↓ indicate higher and smaller is preferred, respectively. Phys. shows support for physical plausibility. **Bold** indicates the best performance.

| Method | Phys. | FID ↓ | R.P.@$1 \sim 3$ | | ↑ |
|---|---|---|---|---|---|
| *Ground Truth* | ✗ | 0.001 | 0.514 | 0.706 | 0.800 |
| BiBo (ours, w/o LDM) | ✗ | 0.238 | 0.467 | 0.662 | 0.762 |
| BiBo (ours, w/o LDM, Phy.) | ✓ | 2.138 | 0.376 | 0.561 | 0.674 |
| BiBo (ours, w/o Causal) | ✗ | 0.101 | 0.526 | 0.716 | 0.801 |
| BiBo (ours, w/o Causal, Phy.) | ✓ | 2.377 | 0.376 | 0.571 | 0.679 |
| BiBo (ours, w/o Gen., Phy.) | ✓ | 2.312 | 0.382 | 0.577 | 0.689 |
| BiBo (ours, w/o Act., Phy.) | ✓ | 1.414 | 0.419 | 0.616 | 0.721 |
| BiBo (ours) | ✗ | **0.076** | **0.542** | **0.738** | **0.829** |
| BiBo (ours, Phy.) | ✓ | 1.883 | 0.411 | 0.604 | 0.716 |

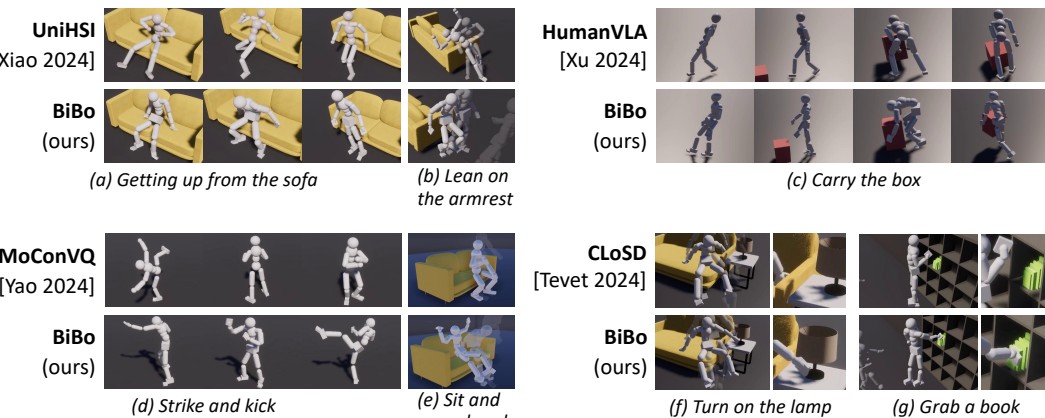

(a) Getting up from the sofa

(b) Lean on the armrest

(c) Carry the box

(d) Strike and kick

(e) Sit and wave hands

(f) Turn on the lamp

(g) Grab a book

Figure 5: Visualization of executing results of comparison methods. Compared with BiBo, UniHSI generates less natural motions, while HumanVLA requires stricter initial positioning for transportation. MoConVQ shows limited motion activity, and CLoSD struggles to achieve precise control.

by $63.8\%$ relatively, demonstrates comparable physical plausibility to CLoSD. As in Tab. 4, BiBo achieves the highest control precision. These results demonstrate that the proposed executor provides an effective medium for bridging general-purpose VLM with the physical world.

**Ablation Study.** We employ LDM and causal attention to mitigate the discontinuity between future and current motion, and use previous generated motion and the actual executed motion to promote smoothness and environmental awareness. We conduct ablation studies to demonstrate the contribution of these designs. Specifically, discontinuity can be manifested as an abrupt change in joint velocities. As a result, we quantify discontinuity by computing average joint acceleration $\bar{a} = \mathbb{E}_j(\|\boldsymbol{p}_j^{n+1} + \boldsymbol{p}_j^{n-1} - 2\boldsymbol{p}_j^n\|_2) / t^2$ at the initial frame $n$ of the future motion, where $\boldsymbol{p}_j^n \in \mathbb{R}^3$ is the position of joint $j$ at frame $n$, and $t$ is the frame interval in seconds.

According to the results in Tab. 4, 5 and 7, both LDM and causal attention improve motion quality. Incorporating previous generated motion effectively reduces discontinuity during motion transitions. Incorporating actual executed motion may affect generation quality, but enhance adaptability to environmental interactions as in Tab. 2.

**Qualitative Result and Case Study.** We conduct a user study to evaluate generation quality through questionnaires. The questionnaire consists of 5 pairs of motions and scene interactions generated by BiBo, MotionLCM, DiP(CLoSD) under the same prompts. Each pair of motions is displayed in the same row with the left–right order randomized, and participants are asked to select the one with higher generation quality. The selections from 30 volunteers are summarized in Tab. 6. The motions and scene interactions generated by BiBo are preferred.

We conduct visual evaluation with the comparison methods in Fig.5, including MoConVQ, Human-VLA, UniHSI, and CLoSD. In (a) and (b), UniHSI produces unnatural movements, whereas BiBo generates natural standing and leaning motion. In (c), when initial agent pose is not aligned with

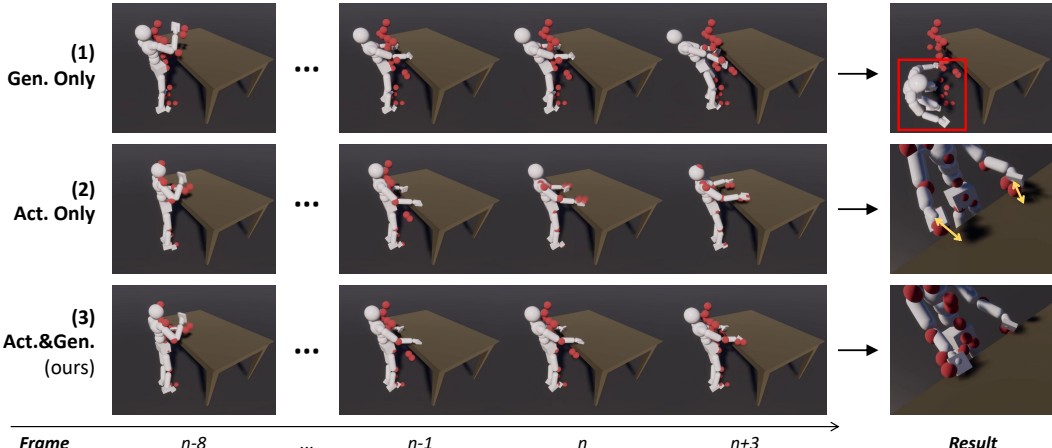

Figure 6: Visual comparison between the executed result of different motion generation method, where the red balls in the image represents the generated motion. Act. and Gen. denotes extending future motion from actual executed motion and previous generated motion, respectively. Extending only from the generated motion fails to account for physical feedback, which may lead to falls. In contrast, extending only from the executed motion introduces discontinuities, resulting in jitter. Our method addresses both issues by incorporating physical feedback while avoiding discontinuities.

the target, HumanVLA fails to pick up the box, while BiBo performs human-like locomotion and completes the task. In (d) and (e), BiBo accurately follows the text prompt to perform the strike and raise hand actions. For (f) and (g), BiBo achieves higher control precision compared with CLoSD.

We further visualize how three strategies for extending future motion respond to environmental feedback: (1) relying only on previously generated but not executed motion, (2) relying only on previously executed motion, and (3) integrating both by using LDM. In the experiment, we place a desk in front of the agent, and instruct it to raise a hand and then slap downward, which lead to a hand–desk collision.

As in Fig. 6, for (1), the generated future motion (represented by the red balls) ignores the desk collision, continuing to drive the hand downward and ultimately causing the agent to lose balance. For (2), the generated future trajectory (frame $\geq n$) exhibits discontinuous jumps relative to the preceding motion (frame $< n$), producing jitter that bounces the hand off the desk surface. By comparison, our method in (3) preserves motion continuity between frames $n-1$ and $n$, while adapting to physical collision by gradually redirecting the hand upon the desk surface at frame $n+3$. This smooth transition reduces jitter and allows the hands to maintain contact with the table.

## 5  CONCLUSION

We introduce BiBo, a framework that empowers off-the-shelf Vision-Language Models to control humanoid agents. Our key insight is that VLMs can control humanoid agents without costly data collection or task-specific training. To achieve this, BiBo comprises two novel components, an embodied instruction compiler that compiles high-level instructions into executable commands, and a diffusion motion executor that generates motions consistent with the physical environment. Experiments show that BiBo achieves high success rates across multiple task designs and maintains high text-to-motion fidelity while performing complex interactions.

**Limitations and Future Work.** First, our executor is trained on a text-to-motion dataset of limited size, which may restrict its generalization capability. With the availability of larger-scale motion datasets (Lin et al., 2023; Fan et al., 2025), there is potential to further enhance robustness and generalization. Second, while our model incorporates environmental feedback through motion execution results, explicitly modeling environmental geometry—such as height maps (Cen et al., 2024) or basis point set (Yi et al., 2024) features—remains an important direction for future exploration. Third, we focus on human–scene interactions in this paper, but there is potential to extend our framework to broader interaction modes, such as hand–object interactions (Chao et al., 2018) and human–human interactions (Liang et al., 2024). We leave these directions to future studies.

ACKNOWLEDGMENTS

This work was supported by the National Natural Science Foundation of China (92570101).

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

# Supplementary Materials for Endowing GPT-4 with a Humanoid Body: Building the Bridge Between Off-the-Shelf VLMs and the Physical World

CONTENTS

# A   SUPPLEMANTARY RELATED WORK

## A.1   MOTION DIFFUSION MODEL

Diffusion (Ho et al., 2020; Song et al., 2020) has emerged as a powerful generative framework for motion synthesis (Khani et al., 2025). It improves generation diversity compared to variational autoencoders (VAEs) (Petrovich et al., 2021a) and generative adversarial networks (GANs) (Shiobara & Murakami, 2021), while being more efficient than GPT-like next-token prediction (Zhang et al., 2023) and BERT-like masked modeling (Guo et al., 2024) methods. While pioneering works directly denoised Gaussian noise into motion sequences (Tevet et al., 2022; Liang et al., 2024; Yi et al., 2024), recent approaches extend diffusion with gradient guidance (Xie et al., 2023) and ControlNet (Gang, 2025) for controllability, latent diffusion models (LDMs) (Xiao et al., 2025; Hong et al., 2025) for enhanced motion quality, and flow matching (Consistency Models) (Dai et al., 2024; Jiang et al., 2025) for faster inference. In this work, we adopt LDM architecture with Classifier Free Guidance and few-step denoising, supporting real-time control while achieving high-fidelity.

## A.2   LARGE VISION-LANGUAGE-ACTION MODEL

Large Vision Language Models (VLMs) have demonstrated cross-domain generalization capability (Achiam et al., 2023; Team et al., 2023; Bai et al., 2023), enabling them to perform action planning for embodied agents conditioned on environmental context and user instructions (Yao et al., 2024; Xiao et al., 2023; Wang et al., 2024a). Among these methods, Vision-Language-Action (Ma et al., 2024; Zhong et al., 2025) attracts extensive research attention. They typically learn action token by finetuning VLM on collected action data (Kim et al., 2024; Black et al., 2024; 2025), thereby driving low-level action executor. Unlike low degree of freedom (DoF) platform (e.g. robot arms, vehicles) (Brohan et al., 2022; 2023; Zhou et al., 2025), humanoids process higher dimensional action space, requiring a strong executor and extensive finetuning data (Bjorck et al., 2025; Ding et al., 2025). BiBo explores an alternative approach to bypass finetuning for action tokens. It employs an embodied instruction compiler that guides an off-the-shelf VLM to output structured action commands. These commands drive an executor trained on open-source human motion dataset, thereby performing diverse scene interaction.

## A.3   HUMAN SCENE INTERACTION

### A.3.1   PHYSICAL AND NON-PHYSICAL HSI TASKS

Non-Physical HSI methods (Hassan et al., 2021; Yi et al., 2024; Cen et al., 2024; Jiang et al., 2024; Huang et al., 2023; Xu et al., 2023) focuses on generating motions that are consistent with the scene. During execution, they directly set joint positions or rotations without involving physical feedback. In this way, the generated and the executed motions are the same, but not physical plausible. The quantitative evaluation of non-physical HSI involves generation quality, control accuracy, and scene consistency. Physical HSI methods (Hassan et al., 2023; Pan et al., 2024; Wang et al., 2024a; Xiao et al., 2023; Tevet et al., 2024; Pan et al., 2025), focus on both motion-scene consistency and physical plausibility. During execution, they apply torques to the joints, and the final executed motion is influenced by physical environmental feedback, which is physical plausible may differ from the generated one. In addition to the non-physical evaluation, it also includes the task success rate. BiBo is a physical HSI system. According to the experiments in Sec. 4, BiBo achieved plausible results in both generation quality and task success rate.

### A.3.2   RL POLICY VS. DIFFUSION GENERATOR AS LLM'S EXECUTOR

RL Policy (Hassan et al., 2023; Xu et al., 2024; Pan et al., 2025) offers physically plausible sence interaction and high success rate on specific tasks. Existing methods typically prompt LLM to select from a fixed skill set (Xiao et al., 2023; Wang et al., 2024a) or rely on a predefined action template, which makes it difficult to scale to more diverse motions (e.g., dance, boxing). Diffusion generator can synthesize a wide variety of high-fidelity motions (Xu et al., 2023; Yi et al., 2024; Tevet et al., 2024). Beyond skill-set–based planning, recent approaches attempt to let the LLM generate motion VQ tokens (Yao et al., 2024) or textual motion caption (Cen et al., 2024; Zhou et al., 2024). However, few methods consider the surrounding scene or support physical interaction. BiBo uses a VLM to

guide a diffusion generator, and the generated motions are tracked by an RL policy. This method generates diverse and natural motions, while maintaining scene awareness and physical plausibility.

### A.4 Arbitrary Length Motion Generation

Arbitrary length motion generation (Chen et al., 2024; Han et al., 2024) involves producing motion sequences that can extend indefinitely, commonly applied in fields like robotics and computer animation. Arbitrary-length motion generation methods can be roughly divided into two types: segment composition and future extension.

Segment composition methods (Athanasiou et al., 2022; Shafir et al., 2023; Zhuo et al., 2025) first generate all key motion segments, and then connect them by generating transitions. However, compared with future extension, these approach may have limitations for motions exceeding the segment length limit, and cannot change control signals in real-time.

Future extension methods predict the next segment based on the previous one. They include extending from a static pose (Hassan et al., 2021; Yi et al., 2024), a fixed-length context (Shafir et al., 2023; Zhuo et al., 2025), and autoregressive generation (Xiao et al., 2025). Extending from a static pose begins with a single pose and generates future motion, but this introduces gaps due to the lack of dynamic context. Using a fixed-length context balances efficiency and continuity, but is limited by the context size. Autoregressive generation, which relies on previously generated frames, captures long-range context and ensures smooth continuity, though it is less efficient.

Arbitrary length physical motion (Tevet et al., 2024) generation introduces an additional challenge, as the generated and executed motions may not be consistent. Extending from generated motion cannot adapt to physical feedbacks, while extending from executed motion introduces a gap between the previous and current generated results. BiBo adopts a lightweight fixed-length context to ensure real-time performance. It extends from both generated and executed motion, enhancing continuity and physical adaptation.

## B  Datasets

### B.1  Random Generated Scene Dataset

#### B.1.1  Statistics

The scene dataset comprises 100 scenes spanning diverse room types and layouts, including 50 living rooms and 50 bedrooms. The floor areas range from 12.30 to $99.18\text{m}^2$, with an average of $49.12\text{m}^2$, featuring various floor plan geometries as illustrated in Fig. 7. The objects include diverse categories such as ornaments (e.g., plant containers, trinkets), containers (e.g., shelves, cabinets), tables (e.g., TV stands, desks), planar surfaces (e.g., monitors, wall panels), as well as furniture (e.g., sofas, beds) and miscellaneous items. The distribution of these categories is shown in Fig. 4.

Each scene contains 6–18 single-interaction tasks and 1–3 composite tasks, resulting in a total of 1,365 single tasks and 162 composite tasks. The single-interaction tasks include 297 sit, 150 sleep, 282 touch, 367 reach, 177 watch, and 294 lift tasks. Notably, reach tasks are also embedded within other interaction tasks to improve testing efficiency. The composite tasks comprise 68 simple, 52 medium, and 42 hard cases, with the length distributions shown in Fig. 4. Examples of tasks are shown in Tab. 7. The evaluation criteria for task success are described in Sec. 4.1. The comparison with existing datasets is shown in Tab. 8.

#### B.1.2  Scene Construction

The scenes are generated using InfiniGen[1] (Raistrick et al., 2023), which produces Blender-format `.blend` scene files. For each generated scene, all light sources and cameras are removed to ensure a consistent simulation environment. Each object in the scene, including walls and furniture, is then exported as an `.obj` mesh file representing its visual geometry, and decomposed into convex collision bodies using the VHACD algorithm (Mamou et al., 2016). The visual meshes and their corresponding collision bodies are subsequently organized into a unified `.urdf` asset file. These URDF

---

[1]https://infinigen.org/

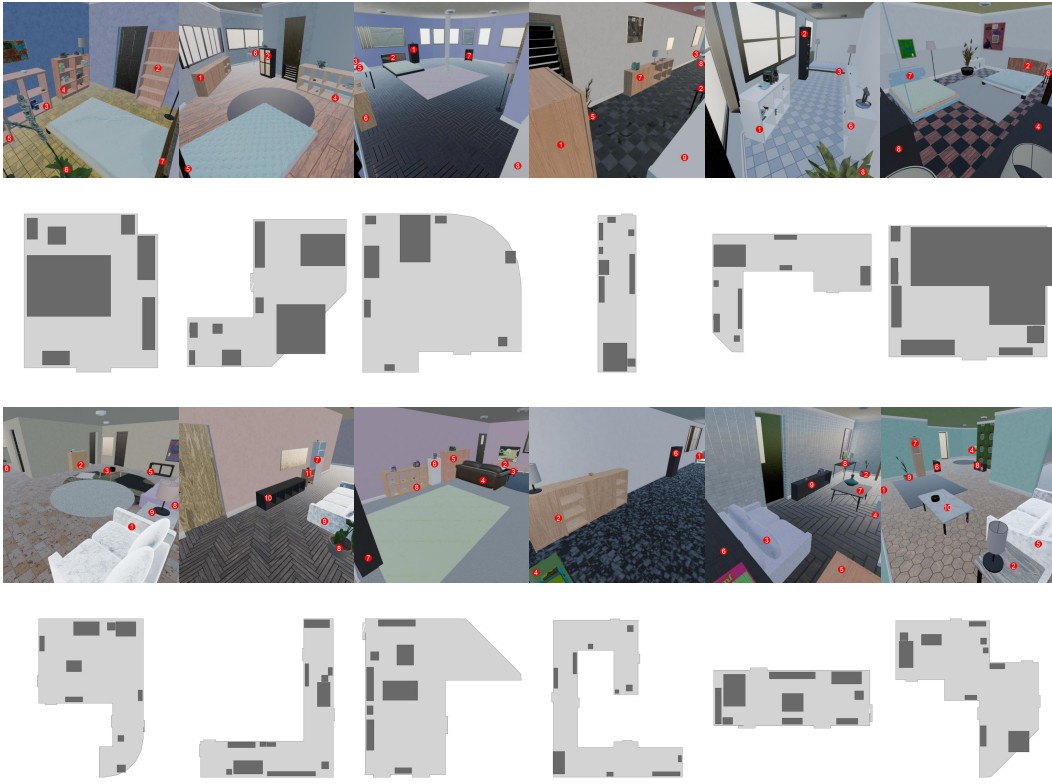

Figure 7: Visualization of the randomly generated scenes and corresponding floor plans. The top two rows show bedroom scenes, and the bottom two rows show living room scenes. In the scene images, each object is annotated with a red label. In the floor plans, light gray regions denote the floor, while dark gray regions represent the objects.

Table 8: Comparison of the evaluation dataset bewteen different HSI methods. We compare the scalability of Object (Obj.), Scene (Sec.), and Task, the variety and quantity of Object (O.), Interaction (I.), and Layout (L.), as well as the maximum step count and whether planning during evaluation (Plan.). The proposed evaluation method is diverse and scalable.

| Method | Dataset | Scalability ↑ | | | Diversity ↑ | | | | | | Difficulty ↑ | |
|---|---|---|---|---|---|---|---|---|---|---|---|---|
| | | Obj. | Sce. | Task | O. Type | Diff. O. | L. Type | Diff. L. | Diff. I. | I. Type | Max Step | Plan. |
| HumanVLA | HITR | | | ✓ | 23 | 76 | 4 | 63 | 63 | 1 | 1 | |
| UniHSI | ScanNet | | | | 20 | 448 | 10 | 10 | 100 | 4 | 8 | |
| CLoSD | Multitask | | | | 3 | 3 | 3 | 3 | 4 | 4 | 1 | |
| BiBo (ours) | Ran. Gen. | ✓ | ✓ | ✓ | **73** | **6146** | **100** | **100** | **1527** | **6** | **15** | ✓ |

files, along with metadata such as object positions and orientations, are imported into IsaacGym Preview 4[2] to construct the physical simulation environment. The initial position of the humanoid agent is dynamically determined at runtime to introduce spatial diversity and prevent initialization bias.

The scene and object generation process follows InfiniGen's default indoor generator, with minor modifications to facilitate task construction. Specifically, the footrests are removed from sofas and blankets are removed from beds. Object extraction is conducted through the Blender Python API 4.2.0[3], and all meshes are simplified to fewer than 10,000 faces to reduce computational overhead. The simplified meshes are subsequently processed using Trimesh[4] for vertex filtering and mesh repair, ensuring watertight topology and the removal of abnormal geometries.

During the import process, since the built-in VHACD and SDF collision modules in IsaacGym exhibit limitations in geometric approximation and compatibility, we employ PyVHACD[5] to manually

---

[2]https://developer.nvidia.com/isaac-gym

[3]https://docs.blender.org/api/current/index.html

[4]https://trimesh.org/

[5]https://github.com/thomwolf/pyVHACD

Listing 1: Example of a task JSON schema. The first object name does not contain an index, indicating that it can correspond to any *bookstack* or *bookcolumn* in the scene. It contains an (*), indicating that the target object of the last interaction can be the same as that of the first interaction. The two intermediate interaction objectives are placed in the same second-level array, representing that they must be achieved simultaneously.

```
{
  "prompt": "Grab a book, then sit on the couch while turning on the
      light, and finally place the book somewhere.",
  "mission": [
    [{"object":"book*","type":"touch"}],
    [{"object":"sofa1","type":"sit"}, {"object":"lamp1","type":"touch"}],
    [{"object":"book","type":"touch"}]
  ]
}
```

construct the collision bodies. Given an object mesh as input, VHACD decomposes it into up to 64 convex hulls that approximate the object's external geometry. Each convex hull is further abstracted by its axis-aligned bounding box (AABB), which is encoded into the URDF as a geometry box element to define the collision shape. Decorative items such as trinkets and plants are defined as dynamic assets, with a density of approximately 50kg/m³, comparable to wooden boxes. Large furniture such as shelves and walls are defined as static assets to stabilize the simulation and reduce computational cost.

To determine the humanoid agent's initial position during evaluation, Shapely[6] is used to construct 2D polygons representing the floor and scene objects. After applying a uniform padding operation to prevent boundary collisions, a random point located inside the floor polygon but outside all object polygons is sampled as the agent's starting position. This ensures valid initialization across diverse layouts while preventing overlap with scene geometry.

### B.1.3 TASK GENERATION

**Task Format.** All tasks are stored in JSON `.json` format, as shown in Lst. 1. It contains two fields: `prompt` and `mission`. The `prompt` field is a natural language instruction of the task, and the `mission` field is a two-level array of interaction objectives. The first level represents interactions that must be completed sequentially, and the second level represents interactions that can be completed simultaneously. Each interaction objective includes the target object name `target` and the interaction type `type`. There are five available interaction types, including *[watch, sit, sleep, touch, lift]*. Note that the *reach* task is implicitly included in these interaction types, since the agent should first navigate to the target object before performing the corresponding interaction.

The object name follows the format `category[index][*]`. The `category` component is required and specifies the category of the target object, wildcard matching all unvisited objects of that type in the scene. The `[index]` component is optional and designates a specific object instance. The `[*]` component is also optional and indicates that, after this interaction, the object will not be excluded from subsequent wildcard matching.

**Single Interaction.** We use a rule-based program to construct single-interaction tasks. For each interaction type, the program queries the scene for eligible interactive objects and generates corresponding prompts accordingly. Specifically, the target objects for the watch task are directional planar surfaces such as TV screens or wall paintings, with example prompts like *"watching TV1."* The sit task targets seating furniture such as sofas or beds, e.g., *"sitting on sofa1."* The sleep task targets beds. The touch task involves ornamental objects in the room, with example prompts such as *"grab a book from bookstack1"* or *"turn on lamp1."* The lift task targets large ornaments or containers, with prompts like *"lifting large plant container1."*

**Composite Task.** The composite tasks are constructed using a semi-automatic pipeline, where volunteers manually annotate 5 simple, 3 medium, and 3 hard task examples. The remaining tasks

---

[6]https://github.com/shapely/shapely

---

**Box 1. Prompt for Composite Task Generation**

**SYSTEM\*:** You are an intelligent task generator. Your goal is to create interaction tasks that a humanoid agent can perform within a given scene.

Each task should be in JSON format with two fields:
- prompt: natural language instruction of the task, can be both abstract and concrete
- mission: a two-level array of interaction objectives. The first level is sequential, while the second level is simultaneous.

Each interaction objective contains two fields:
- type: one of [watch, sit, sleep, touch, lift]
- target: name of target object "category[index][*]". "category" is required. [index] specifies a particular object (optional), omit to match all uninteracted objects of that category. [*] indicates this object should not be omitted in subsequent matching.

Task difficulty criteria:
- simple: contain $< 4$ steps (including navigating to another object between two interactions, e.g., "sit on sofa1 and then sit on sofa2" = 3 steps)
- medium: 4-10 steps, or contains dynamic object manipulation (e.g., lift, transport)
- hard: $> 10$ steps, or contains simultaneous interactions with multiple objects (e.g., sit on sofa and put a hand on the side table)

**USER\*:** Example of simple task:
[examples]
Example of medium task:
[examples]
Example of hard task:
[examples]

The multi-view scene images are provided, each image contain object labels, corresponding to:
- 1: sofa (-0.4, 5.1)
- 2: desk (-0.2, 3.8), containing: [nature shelf trinket1]
- 3: monitor (1.1, 5.3)

You are currently generating an simple task, with abstract prompt and 2 interactions. No simultaneous interaction. No dynamic object manipulation. Please analyze before outputting the final task, and enclose your final answer in $>>>$ and $<<<$.

---

are then generated by GPT-4o based on the provided examples and scene information. All generated tasks are manually reviewed to ensure that they are achievable. The task difficulty is categorized according to the criteria described in Sec. 4.1.

Composite tasks are composed of multiple single interactions that occur either sequentially or concurrently. Compared with single interactions, they feature more diverse interaction patterns, such as *"laying on the coffee table"*, *"leaning on the bookshelf"*, or *"sitting while turning on the lamp"*.

During task generation, the GPT-4o is provided with several components: example tasks, Blender-rendered multi-view scene images with object labels, and a list of objects with their corresponding coordinates and parent–child relationships. To ensure a balanced difficulty distribution, we use a random number generator instead of GPT-4o to determine task difficulty. Specifically, we explicitly specify the task difficulty to be generated and the corresponding criteria to be satisfied.

To capture multi-view scene images, we first compute the positions of all wall corners based on the floor polygon. Cameras are then placed along the bisectors of these corners at a height of 2 m, capturing images with a 30° downward tilt and a 75° field of view (FOV). For placing object labels, the point cloud of each object is first projected onto the camera plane to generate a mask. Then we

use OpenCV[7] to compute the distance from each pixel within the mask to its boundary, and the pixel with the maximum distance is selected as the center of the object label.

Based on these inputs, it outputs tasks in JSON format. The generated JSON file is then parsed and automatically validated by querying the scene to check (1) whether the target objects exist, (2) whether the interaction type belongs to the predefined interaction list, and (3) whether the coordinates of simultaneously contacted target objects are within a 1m distance. Finally, all generated tasks are manually reviewed to ensure correctness and executability. The prompt is shown in Box. 1.

### B.2 HUMANML3D

#### B.2.1 STATISTICS

We train the diffusion-based motion executor on the HumanML3D[8] dataset (Guo et al., 2022b), which comprises 14,616 human motion sequences paired with 44,970 natural language descriptions. HumanML3D covers everyday activities, spatial interactions, and complex body dynamics. The motion data are extracted from HumanAct12 (Guo et al., 2020) and AMASS (Mahmood et al., 2019), augmented through mirroring to double the dataset size, and normalized into a 263-dimensional relative-coordinate motion representation.

The dataset is divided into 24,843 motion sequences for training and validation and 4,382 for testing. After selecting motion sequences with lengths ranging from 40 to 200 frames, a total of 24,545 motion episodes paired with 66,633 captions are used for training and validation, while 4,646 episodes paired with 12,536 captions are reserved for testing.

#### B.2.2 MOTION REPRESENTATION

We follow Tevet et al. (2022), each motion episode is represented as a sequence of joint positions, rotations and velocities in 3D space, sampled at 20 frame per second (FPS). Each motion frame $x \in \mathbb{R}^F$ encodes a single pose and is defined as:

$$x = (\dot{r}_a, \dot{r}_x, \dot{r}_z, r_y, \mathbf{j}_p, \mathbf{j}_r, \mathbf{j}_v, \mathbf{f}),$$

where $\dot{r}_a$ is the root angular velocity around the Z-axis; $\dot{r}_x$ and $\dot{r}_z$ are root linear velocities in the XY-plane; $r_y$ is the root height. The joint features include local joint positions $\mathbf{j}_p \in \mathbb{R}^{3(J-1)}$, rotations $\mathbf{j}_r \in \mathbb{R}^{6(J-1)}$, and velocities $\mathbf{j}_v \in \mathbb{R}^{3J}$, all defined relative to the root. Additionally, $\mathbf{f} \in \mathbb{R}^4$ denotes binary foot contact indicators for four foot joints (two per leg).

#### B.2.3 METRICS DEFINITION

We evaluate the generative capability of BiBo across four dimensions: 1) motion quality, 2) physical plausibility, 3) generation efficiency, and 4) controllability. Specifically, motion quality is measured by Fréchet Inception Distance (FID), R Precision, and Diversity. Generation efficiency is assessed using Average Inference Time per Sequence (AITS). Physical plausibility is evaluated with Penetration, Float, and Skate. Controllability is measured by Mean Absolute Error (MAE). The generation quality evaluation is implemented based on the codebase of MotionLCM (Dai et al., 2024), while the physical evaluation is based on CLoSD (Tevet et al., 2024).

**R Precision.** R Precision measures the retrieval accuracy by evaluating whether the ground-truth motion is ranked within the top-$k$ retrieved motions. We compute R Precision over the entire dataset by accumulating embeddings from all batches. The dataset is then divided into groups of size $R = 32$ (the batch size for R Precision calculation). For each group $g$ containing $R$ text-motion pairs, we compute the Euclidean distance matrix $\mathbf{D}^{(g)} \in \mathbb{R}^{R \times R}$ between text embeddings $\mathbf{E}_t^{(g)} \in \mathbb{R}^{R \times d}$ and motion embeddings $\mathbf{E}_m^{(g)} \in \mathbb{R}^{R \times d}$:

$$\mathbf{D}_{i,j}^{(g)} = \|\mathbf{E}_{t,i}^{(g)} - \mathbf{E}_{m,j}^{(g)}\|_2, \tag{5}$$

where $\mathbf{E}_{t,i}^{(g)}$ and $\mathbf{E}_{m,j}^{(g)}$ denote the $i$-th text embedding and $j$-th motion embedding in group $g$, respectively. For each text query $i$ in group $g$, we rank all motions within the group by their distances and

---

[7]https://opencv.org/
[8]https://github.com/EricGuo5513/HumanML3D

compute the top-$k$ precision:

$$\text{R-Precision@}k^{(g)} = \frac{1}{R} \sum_{i=1}^{R} \mathbb{I}[\text{rank}(\mathbf{D}_{i,i}^{(g)}) \leq k], \tag{6}$$

where $\text{rank}(\mathbf{D}_{i,i}^{(g)})$ is the rank of the ground-truth motion $i$ for text query $i$ within group $g$, and $\mathbb{I}[\cdot]$ is the indicator function. The final R Precision is computed as the average over all groups:

$$\text{R-Precision@}k = \frac{1}{G} \sum_{g=1}^{G} \text{R-Precision@}k^{(g)}, \tag{7}$$

where $G$ is the total number of groups.

**FID.** Fréchet Inception Distance (FID) measures the distribution distance between generated and ground-truth motions in the embedding space. We compute the mean $\boldsymbol{\mu}_g, \boldsymbol{\mu}_{gt} \in \mathbb{R}^d$ and covariance matrices $\boldsymbol{\Sigma}_g, \boldsymbol{\Sigma}_{gt} \in \mathbb{R}^{d \times d}$ for generated and ground-truth motion embeddings, respectively. The FID is then defined as:

$$\text{FID} = \|\boldsymbol{\mu}_g - \boldsymbol{\mu}_{gt}\|_2^2 + \text{Tr}(\boldsymbol{\Sigma}_g + \boldsymbol{\Sigma}_{gt} - 2(\boldsymbol{\Sigma}_g \boldsymbol{\Sigma}_{gt})^{1/2}), \tag{8}$$

where $\text{Tr}(\cdot)$ denotes the matrix trace. Lower FID values indicate better motion quality.

**Diversity.** Diversity measures the variability of generated motions by computing the average pairwise distance between randomly sampled motion embeddings. Given $N$ generated motion embeddings $\{\mathbf{e}_i\}_{i=1}^N$, we randomly sample $T$ pairs of indices $(i_k, j_k)$ without replacement and compute:

$$\text{Diversity} = \frac{1}{T} \sum_{k=1}^{T} \|\mathbf{e}_{i_k} - \mathbf{e}_{j_k}\|_2, \tag{9}$$

where $T$ is the number of diversity samples (typically $T = 300$). Higher diversity values indicate more diverse motion generation.

**AITS.** Average Inference Time per Sequence (AITS) measures the generation efficiency by computing the average time required to generate a complete motion sequence. For each sequence $i$, we record the inference time $t_i$ from the start of the diffusion process to the completion of VAE decoding. The AITS is computed as:

$$\text{AITS} = \frac{1}{N} \sum_{i=1}^{N} t_i, \tag{10}$$

where $N$ is the total number of generated sequences. Lower AITS values indicate faster generation.

**Penetration.** Penetration measures the depth of ground penetration in generated motions. For each frame $t$ in sequence $i$, we compute the lowest joint height $h_{\min}^{(i,t)} = \min_j \mathbf{J}_{i,t,j,y}$, where $\mathbf{J}_{i,t,j,y}$ is the $y$-coordinate (height) of joint $j$ at frame $t$ in sequence $i$. The penetration depth for frame $t$ is:

$$p_{i,t} = \max(0, -(h_{\min}^{(i,t)} + \tau)) \times 1000, \tag{11}$$

where $\tau = 0.005$ m is a tolerance threshold, and the result is converted from meters to millimeters. The overall penetration metric is the mean over all frames:

$$\text{Penetration} = \frac{1}{\sum_i L_i} \sum_{i=1}^{N} \sum_{t=1}^{L_i} p_{i,t}, \tag{12}$$

where $L_i$ is the length of sequence $i$. Lower penetration values indicate better physical plausibility.

**Float.** Float measures the floating height of characters above the ground. Similar to penetration, we compute the lowest joint height $h_{\min}^{(i,t)}$ for each frame. The floating distance for frame $t$ is:

$$f_{i,t} = \max(0, h_{\min}^{(i,t)} - \tau) \times 1000, \tag{13}$$

where $\tau = 0.005$ m is the tolerance threshold. The overall float metric is:

$$\text{Float} = \frac{1}{\sum_i L_i} \sum_{i=1}^{N} \sum_{t=1}^{L_i} f_{i,t}. \qquad (14)$$

Lower float values indicate that characters are closer to the ground, which is more physically plausible.

**Skate.** Skate measures foot sliding artifacts during ground contact. For each consecutive frame pair $(t, t+1)$ in sequence $i$, we identify the contact joint $j_t^* = \arg\min_j \mathbf{J}_{i,t,j,y}$ at frame $t$ (the joint with the lowest height). If both frames $t$ and $t+1$ have the contact joint below the contact threshold $\tau = 0.005$ m, we compute the horizontal sliding distance:

$$s_{i,t} = \begin{cases} \|\mathbf{J}_{i,t+1,j_t^*,[x,z]} - \mathbf{J}_{i,t,j_t^*,[x,z]}\|_2 \times 1000 & \text{if } \mathbf{J}_{i,t,j_t^*,y} \leq \tau \text{ and } \mathbf{J}_{i,t+1,j_t^*,y} \leq \tau \\ 0 & \text{otherwise} \end{cases}, \qquad (15)$$

where $\mathbf{J}_{i,t,j_t^*,[x,z]}$ denotes the $x$ and $z$ coordinates of the contact joint, and the result is converted from meters to millimeters. The overall skate metric is the mean over all consecutive frame pairs:

$$\text{Skate} = \frac{1}{\sum_i (L_i - 1)} \sum_{i=1}^{N} \sum_{t=1}^{L_i - 1} s_{i,t}. \qquad (16)$$

Lower skate values indicate less foot sliding, which is more physically plausible.

**MAE.** Mean Absolute Error (MAE) measures the controllability by computing the average absolute error between predicted and target joint positions for controlled joints. For each controlled joint $j$ with active condition mask $\mathbf{m}_{i,j} \in \{0, 1\}$, we compute the absolute error:

$$\text{MAE}_j = \frac{1}{\sum_i \mathbf{m}_{i,j} L_i} \sum_{i=1}^{N} \mathbf{m}_{i,j} \sum_{t=1}^{L_i} \|\mathbf{J}_{i,t,j}^{\text{pred}} - \mathbf{J}_{i,t,j}^{\text{ref}}\|_1, \qquad (17)$$

where $\mathbf{J}_{i,t,j}^{\text{pred}}$ and $\mathbf{J}_{i,t,j}^{\text{ref}}$ are the predicted and reference joint positions, respectively, and $\|\cdot\|_1$ denotes the L1 norm. The overall MAE is computed as the mean over all controlled joints:

$$\text{MAE} = \frac{1}{|\mathcal{J}_c|} \sum_{j \in \mathcal{J}_c} \text{MAE}_j, \qquad (18)$$

where $\mathcal{J}_c$ is the set of controlled joints (e.g., head, wrists, feet, pelvis). Lower MAE values indicate better controllability.

### B.3 SCANNET SCENEPLAN

ScanNet ScenePlan (Xiao et al., 2023) consists of 10 scenes from ScanNet dataset, which are real-world scanned data. The benchmark includes 20 simple, 23 medium, and 57 hard task sequences, involving multiple interaction categories, such as sit, sleep, and touch. An interaction is considered successful when the character makes contact with the specified object part.

#### B.3.1 SCENE FORMAT

Each scene in ScanNet contains static scene meshes, geometry/instance segmentation, camera poses, and RGB/Depth images. Each geometry segmentation consists of a series of points representing a part of an object. Instance segmentation contains a series of geometry segmentations and object type labels, representing an object.

We extract object information from the scene based on the instance segmentation, and exclude non-interactive objects such as walls, ceilings, and floors. Each object is assigned a unique instance ID to specify interaction targets in tasks. Following UniHSI, we import the scene's mesh as the ground mesh in IsaacGym, and assign consistent physical parameters with UniHSI.

As the rendering quality of mesh is low, we use original RGB and Depth images during VLM inference. When navigating, we mark areas with a height greater than 0.1 as obstacle pixels.

### B.3.2 TASK FORMAT

Each task in the ScenePlan specifies a test scene and provides a list of objects and tasks. The object list annotates the interactive objects or waypoints in the scene. Each object contains part annotations (e.g., for a couch, the seat_surface, back, and left_arm) and the position for interaction. Waypoints can be seen as virtual objects, where the interaction position represents the coordinates of the waypoint.

The task list includes navigation or object interaction. Each task consists of a contact list representing simultaneous contact events. A contact event contains five keys: object name, part name, contact joint, whether the object is in contact, and the contact normal. For waypoints, the contact list length is 1, and all events except for the object name are set to "none", indicating that the agent should move towards the waypoint without interaction.

BiBo uses an off-the-shelf VLM to observe the scene and parse instructions. It can autonomously navigate and generate interaction positions and joint targets. Therefore, we only retain the interactive objects in the input, without using the ground truth waypoints or interaction pose annotations. Specifically, we assign each object a unique ID (e.g., couch1) when parsing the scene. During execution, we generate interaction instructions based on the object category (e.g., for couch1, the instruction would be "sit on couch1").

### B.4 AMASS

### B.4.1 STATISTICS

AMASS (Mahmood et al., 2019) constitutes a comprehensive and diverse database of human motion, which systematically unifies 25 distinct optical marker-based motion capture datasets through a unified framework and parameterization scheme. This consolidated resource provides immediate utility for multiple applications, including character animation, motion visualization, and the generation of training data for deep learning architectures.

The AMASS dataset represents a substantial advancement over existing human motion repositories, encompassing more than 62.87 hours of motion capture data. The database spans 501 unique subjects performing 18,253 distinct motion sequences, thereby providing unprecedented scale and diversity for human motion research and applications. To the best of our knowledge, we are the first to validate motion generation quality using the full AMASS dataset. The preprocessed AMASS dataset (without mirror augmentation) contains 7,582 motion sequences, totaling 40.0 hours.

### B.4.2 IMPLEMENTATION

We download all AMASS[9] subsets (including the latest subsets), prioritizing the SMPLH (Romero et al., 2022)[10] format, and using the SMPLX (Pavlakos et al., 2019)[11] format when SMPLH is unavailable. Following the preprocessing pipeline from HumanML3D, we first process all motion clips into joint coordinate sequences. Then, we remove prefixes and suffixes of motions (typically T-poses) based on their similarity to the initial/final frames. Finally, we segment long clips to below 200 frames for evaluation pipeline adaptation.

### B.5 BABEL

### B.5.1 STATISTICS

BABEL (Punnakkal et al., 2021)[12] is a large-scale motion capture dataset that bridges the gap between precise 3D human motion and semantic action understanding by providing hierarchical language labels for approximately 43 hours of AMASS mocap sequences. The dataset contains over 28,000 sequence-level labels and 63,000 frame-level labels across 250+ action categories, enabling research in action recognition, temporal localization, and motion synthesis.

---

[9]https://amass.is.tue.mpg.de/

[10]https://mano.is.tue.mpg.de/

[11]https://smpl-x.is.tue.mpg.de/

[12]https://babel.is.tue.mpg.de/

TEACH (Athanasiou et al., 2022)[13] processes this data and converts it into a motion generation dataset with frame-level annotations for both transitions and actions. The dataset is partitioned into 22,923 training motions (20.1 hours) and 8,343 evaluation motions (7.3 hours). The motion representation employs root translation combined with 6D rotations for each joint, encompassing 22 joints and resulting in a 135-dimensional feature space.

BiBo uses a 263-dimensional model, which cannot achieve lossless conversion with BABEL's 135 dimensions. Therefore, we start from the original motion data to create a 263-dimensional BABEL dataset.

Rather than extracting each transition-action pair into motion segment separately, we greedily select the longest segments from transition start to action end to maximally preserve shorter valid actions. In total, 13,877 text-motion pairs are extracted, amounting to 27.4 hours.

### B.5.2  IMPLEMENTATION

we first clean invalid motions such as T-poses before and after all motion sequence. Then, we segment motions into clips that start from a transition and end at an action. Clips shorter than 40 frames are dropped, while those longer than 200 frames are truncated.

The text annotations in BABEL are brief and homogeneous. We use additional templates to concatenate multiple action captions within the same segment, and employ various predefined templates to enhance text diversity. Finally, the words in the annotated text may not match the vocabulary of the text embedding in evaluation pipeline. We establish a mapping table from annotated text words to vocabulary words based on similarity search, and use an LLM to fix incorrect mappings.

## C  EMBODIED INSTRUCTION COMPILER

### C.1  ONLINE CONTROL LOOP

The embodied instruction compiler takes as input the user's instruction and the scene observation, and outputs the next structured action command for the executor to perform. To achieve this, Sec. 3.2 introduces a three-stage VQA pipeline designed to progressively fill in the structured action command. In implementation, the embodied instruction compiler can be further divided into three modules: (1) an action Planning Module, which includes the off-the-shelf VLM and the three-stage VQA process; (2) an humanoid agent state machine controlled by the Planning Module; and (3) a Navigation Module that provides path-planning services for the state machine. These three modules, together with the executor and the physical environment, form an online control loop for the humanoid agent.

Specifically, the Planning Module is invoked when a new user instruction arrives, when the previous interaction is completed, or when 150 frames have elapsed since the module was last invoked. It reads the current user instruction, scene information, and agent state (including all the already executed action commands and the ongoing action), and decides whether to skip, start a new action, or end the current one. When a new action starts, the system applies the three-stage VQA process to generate the action command, which is then passed to the state machine.

The state machine is consist of navigation and interaction states. Upon receiving an action command, it switches between navigation and interaction according to the interaction type, the target object, scene information, and the agent's current state (position and ongoing action). In the navigation state, it outputs action commands based on the Navigation Module's path-planning results; in the interaction state, it issues action commands that drive the executor to synthesize the next interaction.

During navigation, the Navigation Module first constructs a pixel obstacle map of the scene and then performs path planning using a modified A* algorithm. In our implementation, we introduce an additional repulsion term in the A* cost function to guide the trajectory away from obstacles and ensure safe navigation.

---

[13]https://github.com/atnikos/teach

---

**Box 2. User Instruction Refinement**

**SYSTEM\*:** You are a state-of-the-art intelligent humanoid robot. Given a vague command, you can interpret it into a sequence of clear, executable instructions according to the scene.

Definition of Clear Instruction:
- Imperative sentence.
- Have exactly one verb.
- If the verb denotes an interactive action, it must explicitly state the object. If not stated in the origin command, you should choose one from the object list.
- The object stated in the sentence must exists in the provided object list.
- Make sure every verb in the original command is included in the instruction sequence.
- Don't miss details like adjectives, directions, and numbers.

**USER\*:** Vague Command: Jump in place. Sit, and sleep.
Objects:
- 1: sofa1 (5.0 ,5.0)
- 2: simple bookcase1 (6.0, 4.5), containing [book column1, book column2]
Clear Instructions:

**ASSISTANT\*:** Jump. Sit on the sofa1. Sleep on bed1.

**USER:** <Multi-view Images>
Vague Command: <User Instruction>.
Objects: <Scene Description>
Clear Instructions:

---

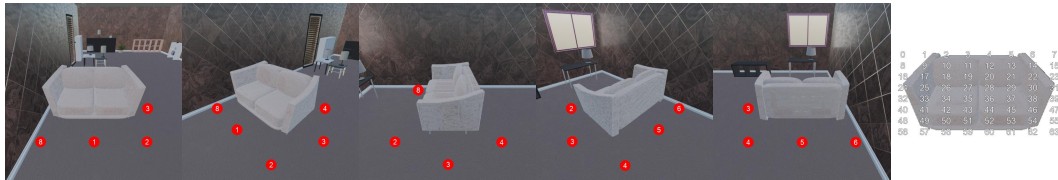

Figure 8: Visualization of the labeled images: the left side presents multi-view images with labels indicating directions relative to the object, while the right side shows the labeled image used for determining target position of key joints.

## C.2 VLM-BASED ACTION PLANNING MODULE

### C.2.1 OFF-THE-SHELF VISION LANGUAGE MODEL

BiBo employs GPT-4o, accessed via the official API[14] . In the Python environment, we utilize the OpenAI API library[15] and manually construct both the conversation-prompt framework and the chain-of-thought procedure. BiBo can also be integrated with other variants of VLMs (e.g., Qwen (Bai et al., 2023; 2025), Claude (Anthropic, 2024), and Gemini (Team et al., 2023)). In our experiments, we compare the scaling capabilities of VLMs of different sizes and examine how tailored prompt designs influence models of comparable capacity. Specifically, we employ Claude 3.5 Sonnet[16] , Qwen2.5-VL[17] , and GPT-4o mini as large-, medium-, and small-scale comparison methods, respectively.

---

[14]https://platform.openai.com/
[15]https://github.com/openai/openai-python
[16]https://www.claude.com/platform/api
[17]https://qwen.ai/apiplatform

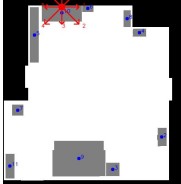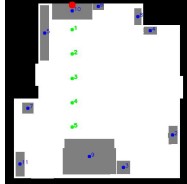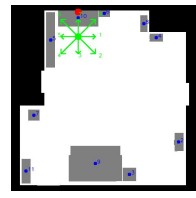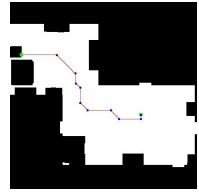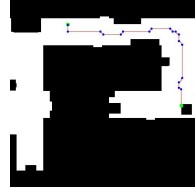

Figure 9: Visualization of agent pose reasoning process and the BEV image inputs for VLM.

Figure 10: Visualization of the navigation result of the modified A*.

### C.2.2 DETAILS OF BASIC ATTRIBUTE ANALYSIS

In Basic Attribute Analysis, the compiler takes as input the user instruction, scene information, and the agent's status. The VLM refines the user instruction based on these inputs as in Box. 2, and decides whether to `skip()`, `start()` a new action, or `end()` an ongoing action, using the prompt in Box. 3. When an action starts, the VLM first generates a brief summary of the action as the motion caption. Then, it analyzes the basic motion attributes according to the scene objects and the agent's current state, facilitating the subsequence reasoning process, using the prompt in Box. 4 and 5.

**Scene Information.** It includes multi-view images and a corresponding scene description. The multi-view images are captured in the same manner as described in Sec. B.1.3, by capturing images from all wall corners to maximize the coverage of the scene contents. Each image is labeled with object IDs. The scene description contains object IDs, coordinates, and parent–child relationships among objects. The parent–child relationships are computed using a union–find algorithm (Cormen et al., 2022) based on the containment relationships among the objects' 2D polygonal shapes, and the object with the largest (or similar) area and the lowest height is selected as the parent.

**Agent Status.** It includes all previous planning results and executed actions, the currently executing action, the agent's pose, nearby objects, and objects held by the agent. The previous planning results are provided through multi-round conversation history, while the currently executing action is identified by its motion caption, elapsed time, and execution result. All objects are specified by their names and corresponding IDs.

**Motion Attributes.** They include the related objects, the key joints involved, the interaction type, and whether IK should be enabled. The related objects are categorized into three types: *target*, *at*, and *by*. The *target* is the target object of interaction, *at* refers to the anchor used for agent localization, and *by* denotes the tool being used. For example, take a book *(target)* from the shelf *(at)* or turn on the TV *(target)* by the remote *(by)*. The *target* is chosen as the anchor if there are no *at* objects. The key joints include the *head*, *hands*, *feet*, and *pelvis* (the latter being controlled by the agent's position and facing direction). The interaction types are divided into four categories: *contact*, *non-contact*, *distal*, and *manipulate*. *contact* involves physical contact (e.g., sitting), *non-contact* refers to proximity without contact (e.g., hand dryer use), *distal* denotes remote interactions without approaching the target, and *manipulate* refers to interactions that change the object's state (e.g., carrying).

### C.2.3 DETAILS OF AGENT POSE REASONING

In Agent Pose Reasoning stage, the VLM take the rendered scene images with labels as input, and sequentially infers: (1) the agent's direction relative to the anchor object, (2) the agent's distance relative to the anchor object, and (3) the agent's facing direction. The success of this reasoning process depends on the choice of scene viewpoint as well as the placement and annotation of the labels within the rendered images.

**Agent's Direction Relative to Anchor.** We first capture multi-view rendered images of both the anchor and target objects. The images are rendered with an object-centered setup at a 30° depression angle, sampled at 45° intervals with a 75° field of view, and occluded views are excluded, as shown in Fig. 8.

---

**Box 3. Planning for The Next Action**

**SYSTEM\*:** You are a state-of-the-art intelligent humanoid robot.
You are currently in a scene with the following objects, and the layout of the scene is provided in the image:
<Multi-view Images>
<Scene Description>
You should perform a sequence of actions to fulfill the following instructions, step by step:
<User Instruction>

In each step, the sensor will provide you with your position, the objects you are holding, your current states and action elapsed time. You should decide your next action command accordingly, including:
1. skip(): Stay in the current state.
2. start(action): Start doing the specified action. Will add the action to your current state, which means you are doing the action.
3. stop(action): Stop doing the specified action. Will remove the action from your current state, which means you are no longer doing the action.

Example of actions:
1. Jumping
2. Leaning against wall1
3. Taking hammer1 from toolbox1
4. Standing on the right of table1 three meters away

Constraints:
- Each step you can only give one command.
- Each action should be indivisible, which containing no more than one verb.
- For interactive actions, you must explicitly specify the target object, as example 2, 3, 4.
- Multiple actions in current state means performing multiple actions at the same time.
- You are encouraged to directly output the command without any additional explanation.

**USER\*:** Scene: <Scene Description>
Position: origin (0, 0)
State: [("Standing", 3s, done)]
Holding: {left hand: none, right hand: none}

**ASSISTANT\*:** stop("Standing.")

**USER:** . . . (Multi-round Conversation)

---

Then, we employ Orient Anything[18] (Wang et al., 2024b), a model fine-tuned on DINOv2 (Oquab et al., 2023) that can predict object orientation and corresponding confidence scores in a zero-shot manner. For each view, Orient Anything outputs an estimated orientation, which we discretize into directional bins. The confidence scores of all views within each bin are summed, and the bin with the highest aggregated confidence is selected as the object's front direction. If the overall confidence is below a threshold, the object is considered to have no clear orientation.

Subsequently, eight labels are uniformly placed around the anchor object at 0.4 m intervals and 45° spacing, excluding non-traversable locations. These labels are projected onto the anchor's multi-view rendered images and provided to the VLM, together with textual descriptions indicating their spatial relations (if applicable) to the anchor and target, e.g., "Label 1 is in front of the monitor." The VLM then selects one label as the agent's relative direction to the anchor object. We use the prompt in Box.6. For non-distal interactions, the prompt explicitly states that the label represents a standing location rather than a direction to skip the next reasoning step, as non-distal interaction is expected to occur within a 0.4m range around the object.

---

[18]https://github.com/SpatialVision/Orient-Anything

---

**Box 4. Basic Motion Attribute Analysis (Object)**

**USER\*:** You are a state-of-the-art intelligent humanoid robot, <VLM Instance ID>. The following command describes an interaction with objects marked with "<" and ">". Please identify the roles of these objects.

The roles include "target", "by", and "at".
- "target" is the intented interaction target.
- "by" means the interaction is done by using these objects.
- "at" means the interaction is done at these objects.

Please answer in the following format:
   target: <target>
   by: [<object1>, <object2>, ...]
   at: [<object3>, <object4>, ...]

Example:
Question 1: Grab the <cloth1>from the <washing machine1>.
Answer 1:
   target: <cloth1>
   at: <washing machine1>
Question 2: Turn on the <light1>with the <switch1>.
Answer 2:
   target: <light1>
   by: <switch1>

Sentence: <Motion Caption>

---

**Agent's Distance from Anchor.** As shown in Fig. 9, we construct a simplified bird eye view (BEV) of the scene using the polygons of floorplan and objects, where walkable areas and obstacles are filled with different colors, and the positions of the agent and objects are indicated by specific labels. Along the previously predicted direction, a series of distance labels are placed starting from 0.5m and spaced at 1m intervals, excluding positions that are not reachable. The prompt template is shown in Box. 7, which provides each label's distance and direction relative to target objects (if target is distinct from the anchor). The VLM selects one label as the final standing location. For non-distal interactions, this step is skipped, and the distance is fixed at 0.4m.

**Agent's Facing Direction.** Using the BEV map, we place arrows around the standing location as candidate facing directions, as shown in Fig. 9. We use the prompt in Box. 8. In the prompt, for each candidate we provide (i) its angle (relative to a global reference) and (ii) the object it points toward, restricting objects to those relevant to the current interaction. The VLM then selects one candidate as the final facing direction. The BEV input is omitted at this stage for non-distal interactions.

### C.2.4 Details of Key Joint Generation

This process generates joint target positions relative to the object surfaces, as in Box 9. Given the agent's location relative to the object, we select the scene image rendered from the corresponding viewpoint and uniformly place an 8×8 grid of labels on it, shown in Fig. 8. Each label corresponding to a point on the object surface. For each key joint, the VLM selects one label as an anchor point, from which it infers the direction and distance between the joint and the anchor. These values collectively define the joint's target position.

For contact and manipulate types of interactions, we adopt a simplified strategy. For objects with a size smaller than $0.25 \times 0.25 \times 0.25$m, the key joint generation process is skipped, and the target position is directly set at the object center. For larger objects, the reasoning for direction and distance is omitted. Instead, the model determines whether the agent should exert force on the object. If not, the target position is placed on the object surface; if so, it is positioned toward the object center, guiding the agent to apply force to the object.

---

**Box 5. Basic Motion Attribute Analysis (Motion)**

**SYSTEM\*:** You are a state-of-the-art intelligent humanoid robot, <VLM Instance ID>. You can assess the details of an action to perform it accurately.

The details include:
- Interaction Type: one of [contact, non-contact, long-range, manipulation]. Contact refers to contact with the target object (if present), non-contact denotes proximity without contact, long-range indicates interact at a distance, and manipulation involves moving the object.
- Key Joints: the most important joints involved in the interaction, available options: ['pelvis', 'left_foot', 'right_foot', 'left_hand', 'right_hand', 'head']
- Use Inverse Kinematic: true or false, enable for precise or stable interactions (e.g., touching small objects, carrying), disable for high dynamic motions (e.g., dancing)

Constraints:
- No more than two contact points. If there may be more, choose the most important two.
- One hand cannot manipulate multiple objects.
- You are encouraged to directly output the details without any explanation.

**USER\*:** Interaction: Turn on <lamp1>on <table1>.
State: You are holding book1 in your right hand.
Details:

**ASSISTANT\*:**- Interaction Type: contact
- Key Joints: ["left_hand"]
- Use Inverse Kinematic: true

**USER:** Interaction: <Motion Caption>
State: You are <Agent State>.
Details:

---

**Box 6. Inferring Agent's Direction Relative to Anchor**

**USER\*:** You're the state-of-the-art intelligent humanoid robot. When you interact with an object, you can determine your own position relative to it.

The scene layout is provided in the images, which are photos of the <Anchor Object> taken from different perspectives. The photos contain red circular markers labeled with integer indices, each marker represents a direction relative to the <Anchor Object>. In different photos, the marker with the same index represents the same direction.
<Multi-view Images>

Marker Directions: <Direction Label Descriptions>
For example:
- Label 1 is in the front of monitor and desk
- Label 2 is in the front-left of monitor and desk
. . .
When performing <Motion Caption>, which direction should you stand in? Your answer should be a marker index enclosed in >>>and <<<.

---

### C.2.5 MULTIPLE ACTION MERGING

For actions occurring concurrently (e.g., when one action has not yet ended while another starts), the system first checks for potential conflicts after performing basic attribute analysis. Conflicts are defined under the following conditions:

---

**Box 7. Inferring Agent's Distance from Anchor**

**USER\*:** You're the state-of-the-art intelligent humanoid robot. When you interact with an object, you can determine your own position relative to it.

The provided image is a bird-eye-view map of the scene.
<BEV>

Red markers indicate the <Target Object>, while blue markers denote the IDs of scene objects. These objects are as follows:
<Scene Description>

The green markers in the image represent a set of candidate standing locations:
<Distance Label Descriptions>
For example:
There distance to the monitor is:
- Label 1 : 0.5m
- Label 2 : 1.5m
. . .

When you are performing <Motion Caption>, which position should you stand in? Your answer should be a marker index enclosed in >>>and <<<.

---

**Box 8. Inferring Agent's Facing Direction**

**USER\*:** You're the state-of-the-art intelligent humanoid robot. When you interact with an object, you can determine your facing direction.

The provided image is a bird-eye-view map of the scene.
<BEV>

Red markers indicate the <Target Object>, while blue markers denote the IDs of scene objects.
<Scene Description>

The green arrows in the image represent a set of candidate facing directions:
<Facing Label Descriptions>
For example:
- Arrow 1: 0°, facing directly to the monitor.
- Arrow 5: 180°, facing away from the monitor.
. . .

When you are performing <Motion Caption>, which direction should you face? Your answer should be an arrow index enclosed in >>>and <<<.

---

- The two actions involve the same key joint;

- Both are non-distal interactions and their target objects are more than 1m apart;

- Their anchor objects are more than 1m apart.

If no conflict is detected, the system proceeds to generate a new action command through the subsequent pose reasoning and joint generation processes, and updates the existing action command accordingly.

---

**Box 9. Inferring Target Position of Key Joints**

**SYSTEM\*:** You are a state-of-the-art intelligent humanoid robot. When interacting with an object, you can determine the target position of a specific joint. This process involves two steps: (1) locating a target point on the target object, and (2) specifying the joint's position relative to that target point (including its direction and distance).

Example 1: use the laptop
Joint: right_hand
Target Point: keyboard
Direction: up
Distance: 0

Example 2: use the hand dryer
Joint: right_hand
Target Point: outlet
Direction: down
Distance: 0.2 m

**USER\*:** The image shows a \<View Direction\> view of the \<Target Object\>. It contains an 8×8 grid of numbered labels (indexed from 1 to 64), with each label corresponding to a point on the surface of the \<Target Object\>.
\<Image\>

You are performing \<Motion Caption\>. Please identify the target position of your \<Joint Name\> in the following format:
- Target Point: a noun or noun phrase (can include adjectives and other modifiers to better describe its features) describing a part of the \<Target Object\> that your \<Joint Name\> refers to.
- Label: one label index in the image corresponding to the target point.
- Direction: the direction of your \<Joint Name\> relative to the target point. Available options: [up, down, left, right, directed into the image, directed out of the image, toward the object center, along the surface normal].
- Distance: the distance of your \<Joint Name\> from the target point, using meter (m).

---

Specifically, when generating a new action command, the system constructs a merged motion caption combining the ongoing and newly initiated motions (e.g., if *sit* has not ended and *turn on the lamp* starts, the merged caption becomes *sit and turn on the lamp*). During Agent Pose Reasoning, the rendered image centers on all anchor objects, and the merged motion caption is used in the VQA process of VLM. During Joint Pose Generation, the system regenerates the key joint target positions for the ongoing actions and overwrites them in the corresponding action commands.

All action commands are stored concurrently in the humanoid state machine's action command list. The state machine's final output command uses the merged motion caption, adopts the location and facing of the latest generated action command (since the reasoning already considers the merged caption), and directly combines the key joint positions of all action commands.

### C.2.6 RULE-BASED REFLECTION

To enhance the stability of the three-stage VQA process, we incorporate a reflection mechanism. Reflection is triggered in the following cases: (a) failure to follow the expected QA format; (b) conflicts with ongoing actions; and (c) implausible spatial relationships (e.g., a contact positioned too far from the anchor). When an error is detected, the system responds based on the error type: format violations result in a simple rollback, while logical inconsistencies trigger a rollback followed by the explicit insertion of the reflection result into the conversation history, as in Box. 10.

---

**Box 10. Example of Explicit Insertion of Reflection Results in Conversation History**

**ASSISTANT:** start("Sitting on sofa1.")

**USER:** Command execution failed.
Reason: You are performing "Sleeping on bed1.", and it is impossible to perform "Sleeping on bed1 and sitting on sofa1.", as bed1 and sofa1 are located too far apart.

Possible solutions:
- stop "Sleeping on bed1."
- try another action

---

### C.3  HUMANOID AGENT STATE MACHINE

The State Machine consists of two states, Navigation and Interaction, and maintains an action command list that stores all currently executing actions. During each frame update, the State Machine first converts the positions and directions in each action commands from local coordinate system, defined relative to its anchor object, into global coordinate system. Then, it employs the the method in Sec. C.2.5 to convert the command list into a merged action command. Next, the system evaluates the difference between the current agent pose (i.e., location and facing) and the target value.

When the distance to the target is less than 0.5m and the deviation in facing direction is within 45°, the State Machine transitions into the Interaction state and marks the corresponding action command as executing. Once all executing non-distal actions with target objects are completed, the system switches back to the Navigation state.

### C.3.1  NAVIGATION STATE

The State Machine first invokes the Navigation Module to generate a planned path. It identifies the point on this path closest to the agent's current position as the starting point, and then selects the next waypoint that is both nearest to the start and farther than 0.5m away.

The direction from the current position to this next point is used as the facing direction. If the angular deviation between the current and target facing directions exceeds 45°, the target location is fixed at the current position, and the motion caption is set to *"A person is slowly turning around in place."* Otherwise, the target location is set to the next waypoint, and the motion caption is set to *"A person is walking."* When the distance to the next path point exceeds 1.2m, the moving speed is set to 1m/s. When the distance falls below 1.2m, the speed decreases linearly until it reaches 0m/s at 0.2m.

The navigation command is further merged with concurrently executing distal, manipulation, or non-interactive actions (i.e., actions without target objects, which are excluded from the second stage of Basic Attribute Analysis and categorized as the Type-V action), allowing navigation and motion execution to proceed simultaneously.

### C.3.2  INTERACTION STATE

First, for non-distal motions, the target location is set to a position 0.3m away from the target object, guiding the agent to walk closer to the target. When the positional error drops below 0.2m, the interaction action command start executed. For contact actions involving force exertion, the joint target position is first assigned to the contact point on the object's surface. Once the agent is within 0.1m of this point, the target position is set to the object center.

After 60 frames of execution, the system evaluates whether each interaction is done based on pre-defined rules: a contact action is done if the relevant joint is applying force, is within 0.25m of the target position, and within 0.1m of the target surface; a non-contact action is done when the joint is within 0.1m of the target position; a distal action is done if its duration exceeds 120 frames; and a manipulation is done when the object's movement exceeds 0.1m while remaining within 0.1m of the hand. The execution duration and completion status of all actions are continuously fed back to the planner to support subsequent decision-making.

---

**Algorithm 1:** The Modified A* Path Planning with Repulsion Term

---

**Input:** Start point $p_{\text{start}}$, goal point $p_{\text{goal}}$, obstacle distance map $M$, repulsion ratio $\alpha$, repulsion
      distance $\beta$, scene-to-pixel scaling ratio $\gamma$
**Output:** Path $\mathcal{P} = [p_1, \ldots, p_T]$
$O \leftarrow (M == 0)$;
$M \leftarrow \max(\beta - \gamma M, 0) / \beta$;
$M \leftarrow (1 - \alpha)M + \alpha$;
Initialize open set with $p_{\text{start}}$ and set $g(p_{\text{start}}) = 0$ ;        `// g is current cost`
**while** *open set not empty* **do**
    $p_t \leftarrow \arg\min_p f(p)$ ;             `// f is estimated cost`
    **if** $p_t = p_{goal}$ **then break**;
    **foreach** *neighbor $p'$ of $p_t$* **do**
        **if** $O[p']$ *is true* **then continue**;
        $g(p') \leftarrow g(p_t) + M[p']$;
        $f(p') = g(p') + \alpha \cdot h(p')$ ;        `// h is heuristic function`
        **if** $g(p')$ *improves* **then** update open set and parent pointer;

Reconstruct path $\mathcal{P}$ from parent pointers;
**return** $\mathcal{P}$

---

### C.4 NAVIGATION MODULE

The Navigation Module performs path planning based on the input agent position, target position, scene floorplan, and all object information. First, it constructs a navigation map using the floorplan and the convex decomposition of all objects, while optimizing both the agent and target positions to ensure they are located outside obstacles. It then performs pathfinding using an modified A* algorithm that incorporates a repulsion term to encourage paths farther from obstacles, followed by a polyline simplification step that converts the dense pixel path into sparse waypoints.

#### C.4.1 MAPPING MODULE

For each scene object, the module applies the current position and rotation to transform the coordinates of the 64 convex AABBs obtained from decomposition (refer to Sec. B.1.2), and projects them onto the XY-plane. This calculation is GPU-accelerated with PyTorch[19]. The projected polygons are converted into Shapely geometries, padded by 0.1 to form the 2D polygonal outlines of the objects. The floorplan and object polygons are then processed with PyClipper[20] for clipping, and discretized into a pixel obstacle map. Pixels covered by the floorplan but not by any object represent navigable areas, while the rest correspond to obstacles. The map is discretized to a resolution of $512 \times 512$, scaled isotropically according to the longest axis of the floorplan.

If the agent position lies within an obstacle, the module searches in four directions (up, down, left, right) to find the nearest accessible area, marking all traversed pixels as navigable. If the target position lies within an obstacle, it is shifted outward according to the direction of the corresponding stand location relative to the object.

Finally, OpenCV is employed to compute an obstacle distance map, which records the distance from each navigable pixel to the nearest obstacle. Specifically, a morphological dilation operation is iteratively applied to the pixel obstacle map, where navigable pixels are assigned a value of 0 and obstacles 1. The number of dilation iterations before a pixel is removed corresponds to its distance to the nearest obstacle. The process terminates once all pixels become 1.

#### C.4.2 PATH PLANNING WITH MODIFIED A*

The modified A* algorithm utilizes this obstacle distance map for pathfinding. Specifically, a repulsion term is introduced into the cost function, increasing the step cost as the agent approaches

---

[19]https://pytorch.org/
[20]https://github.com/fonttools/pyclipper

obstacles. This design encourages the generation of paths that stay farther from walls, thereby lowering the overall path cost. The complete algorithm is presented in Alg. 1.

We use Shapely to simplify the path from dense pixels into sparse waypoints. We then iterate through each pair of waypoints. If the line connecting them maintains a minimum distance greater than 0.5m from the nearest obstacle (check by padding the line into a rectangle with a width of 0.5m and performing a polygon intersection), the intermediate points are bypassed. This process yields the final simplified path.

## D  DIFFUSION-BASED MOTION EXECUTOR

The executor generates future motion extending the currently performing one, condition on the structured motion command from the compiler. The core of the executor is a latent diffusion model, which includes a Variational Autoencoder (VAE) that transforms human motion into latent motion tokens, and a Latent Diffusion Model (LDM) that performs a diffusion process in the latent space.

### D.1  DIFFUSION PRELIMINARY

Our denoising network follows the DDIM (Song et al., 2020) framework with a fixed forward diffusion and a learned reverse denoising process. Let the future-motion latent be denoted by $\boldsymbol{S}_f \equiv \boldsymbol{S}_f^0$, and let $\{\beta_t\}_{t=1}^T$ be a variance schedule with $\alpha_t = 1 - \beta_t$ and $\bar{\alpha}_t = \prod_{i=1}^t \alpha_i$.

**Forward Process.** Starting from clean data $\boldsymbol{S}_f^0$, the forward (noising) Markov chain adds Gaussian noise over $T$ steps:

$$q\left(\boldsymbol{S}_f^t \mid \boldsymbol{S}_f^{t-1}\right) = \mathcal{N}\left(\boldsymbol{S}_f^t;\ \sqrt{1-\beta_t}\,\boldsymbol{S}_f^{t-1},\ \beta_t\,\mathbf{I}\right),$$

This can be written in closed form as:

$$q\left(\boldsymbol{S}_f^t \mid \boldsymbol{S}_f^0\right) = \mathcal{N}\left(\boldsymbol{S}_f^t;\ \sqrt{\bar{\alpha}_t}\,\boldsymbol{S}_f^0,\ (1-\bar{\alpha}_t)\,\mathbf{I}\right).$$

**Reverse Process.** At sampling time, we sample the terminal noise from a standard normal distribution and iterate backward:

$$\boldsymbol{S}_f^T \sim \mathcal{N}(0,1), \qquad \boldsymbol{S}_f^{t-1} = \frac{1}{\sqrt{\alpha_{t-1}}}\left(\boldsymbol{S}_f^t - \frac{\epsilon\left(\boldsymbol{S}_f^t,\ \boldsymbol{s}_m,\ \boldsymbol{s}_c\right)}{\sqrt{1-\alpha_{t-1}}}\right), \qquad \boldsymbol{S}_f = \boldsymbol{S}_f^0.$$

We employ Classifier-Free Guidance (CFG) (Ho & Salimans, 2022) to modulate the control strength, using the condition set $(\boldsymbol{s}_m, \boldsymbol{s}_c)$ within the noise predictor $\epsilon(\cdot)$.

**Training Objective.** The reverse model $\epsilon_\theta$ is trained to predict the injected noise at each timestep via the DDPM (Ho et al., 2020) objective:

$$L_{\text{simple}} = \mathbb{E}_{t,\,\boldsymbol{S}_f^0,\,\epsilon}\left[\left\|\epsilon - \epsilon_\theta\left(\boldsymbol{S}_f^t,\ t,\ \boldsymbol{s}_m,\ \boldsymbol{s}_c\right)\right\|^2\right],$$

where $\boldsymbol{S}_f^t$ is obtained from the forward process as above. After $T$ denoising steps, the procedure yields the clean future-motion latent $\boldsymbol{S}_f = \boldsymbol{S}_f^0$.

### D.2  DIFFUSION ARCHITECTURE

We adopt a 9-layer Transformer Encoder architecture with skip connections as the diffusion backbone. Each layer uses 4 self-attention heads, with a model hidden size of 256 and an feed-forward network (FFN, a linear layer) dimension of 1024. Starting by setting the future motion tokens $\boldsymbol{S}_f^T$ as Gaussian noise, where $T$ is the total denosing steps. The diffusion timestep $T$ is represented via a sinusoidal positional embedding passed through a small MLP. The diffusion denoiser $\mathcal{F}$ iteratively denoises $\boldsymbol{S}_f^t$ using DDIM scheduler, condition on command $\mathcal{C}$ and the latent of preceding actual executed motion $\boldsymbol{S}_a$. Specifically, we encode the motion captions using a pretrained text encoder (Devlin et al., 2019), where the [CLS] token is taken as the caption token $\boldsymbol{s}_m \in \mathbb{R}^H$. Other control parameters are encoded with an MLP, and their representations are summed to form the control token $\boldsymbol{s}_c$. $\boldsymbol{s}_m$ and $\boldsymbol{s}_c$ are concatenated with $\boldsymbol{S}_a$, together conditioning the denoising process $\boldsymbol{S}_f^{t-1} = \mathcal{F}(\boldsymbol{S}_f^t, [\boldsymbol{S}_a, \boldsymbol{s}_m, \boldsymbol{s}_c])$ by cross attention with $\boldsymbol{S}_f^t$.

### D.3 VAE Architecture

The proposed Variational Autoencoder (VAE) utilizes a shared 9-layer Skip-Transformer architecture for both encoder and decoder. Each layer employs 4-head multi-head attention with a model dimension of 256 and a 1024-dimensional feed-forward network. We use the GELU activation function and set the dropout rate to 0.1. To enforce temporal causality, we implement two key mechanisms. First, a causal mask is applied within the self-attention layers to the concatenated sequence of latent and frame tokens. Second, for cross-attention, we introduce an growing memory window that expands its size in correspondence with the target timestep, restricting access to only the relevant prefix of latent variables. These designs effectively prevent future information leakage, ensuring faithful and time-consistent sequence generation.

### D.4 Training Details

Our implementation uses PyTorch as the primary deep learning framework for training and inference. We leverage the Transformers[21] library for tokenization and text embedding. The diffusion process is implemented with the Diffusers[22] toolkit, including noise schedulers and sampling pipelines. All experiments run on a CUDA-enabled backend [23].

All training is conducted on the HumanML3D dataset (Guo et al., 2022a). Motion sequences are processed at 20 Hz. Each sequence is concatenated with a **stance sequence** at the beginning to serve as the initial state, enabling the model to learn to initiate a motion from scratch. When trianing with control conditions, we apply random conditioning by sampling from 60 predefined combinations of control signals including trajectory, heading, velocity, and key joint positions, enabling flexible spatial control during generation. During generation, following CLoSD (Tevet et al., 2024), we set the length of previous motion context to 20, and generation length to 40.

We use the AdamW optimizer with parameters $\beta_1 = 0.9$, $\beta_2 = 0.999$, weight decay=0, and $\epsilon = 10^{-8}$. The learning rate is initialized to $5 \times 10^{-4}$ and follows a cosine decay schedule with a linear warm-up over the first 2000 steps. We utilize a DDIM sampler, apply gradient clipping with a threshold of 1.0, and set the unconditional probability to 0.1. For inference, we use 5 DDIM steps and a guidance scale of 7.5. The model is trained for up to 3000 epochs with a batch size of 256. All experiments are run with a fixed random seed of 1234 for reproducibility.

### D.5 IK Post Optimization

To enable precise control of end effector joints, we introduce an Inverse Kinematics (IK) post-processing step. This step optimizes the posture of arms or legs when the compiler issues precise contact command. It operates on the generated pose for each frame, with the optimized motion serving as the reference for the tracking policy.

Our method employs a custom variant of the FABRIK (Aristidou & Lasenby, 2011) algorithm. Specifically, for the upper limb, we solve for a 3-DOF kinematic chain consisting of the left shoulder, elbow, and wrist, with the shoulder joint acting as a fixed root. For the lower limb, we solve for a 3-DOF kinematic chain consisting of the hip, knee, and ankle, with the hip joint serving as the fixed root. The joint lengths are derived from the initial pose and remain constant. Unlike standard FABRIK, our primary IK target is the wrist joint, not the hand end-effector. This design choice provides a stable base for subsequent hand orientation and grasping.

The iterative optimization process begins by placing the end joint (wrist for the upper-limb, ankle for the lower-limb) directly at the target position. Then, standard backward and forward passes are executed to satisfy the kinematic constraints of the limb segments. The position of the end effector is subsequently reconstructed, where it is placed at a fixed distance from the solved joint position and oriented along the vector of the corresponding segment (forearm for the upper-limb, thigh or shin for the lower-limb). For targets beyond the limb's reachable workspace, the chain is fully extended and pointed towards the target. The entire iterative refinement process is capped at a maximum of 50 iterations, with a tolerance of $10^{-3}$.

---

[21]https://huggingface.co/docs/transformers
[22]https://huggingface.co/docs/diffusers
[23]https://developer.nvidia.com/cuda-toolkit

### D.6 TRACKING POLICY

We utilize the PHC tracking policy (Luo et al., 2021), enabling humanoid agents to accurately track the generated joints in physical space. We initialize the policy network with weights from CLoSD (Tevet et al., 2024), which is trained using Isaac Gym simulator (Makoviychuk et al., 2021)[24].

## E COMPARISON METHODS

### E.1 TASK COMPLETION

#### E.1.1 UNIHSI

UniHSI (Xiao et al., 2023) represents each human–scene interaction task as a sequence of contact interactions. It employs AMP (Peng et al., 2021) achieve contact goals. Each contact contains a standing position, a contact flag (if not in contact, the agent navigates to the standing position), and a set of contact joint information. Each joint information contains a joint name, a contact normal and 200 surface points on the contact area (which, in the original task definition, correspond to the target object and a specific part index of that object). During task execution, the agent also needs to obtain nearby scene height information.

We use the official GitHub repository[25] and the released model checkpoints. During evaluation, we import scenes in IsaacGym using the same asset options as in the scene configuration, and additionally sample each object's point cloud using Trimesh to construct the contact surfaces and scene height map. To build the ground-truth task plan, we first procedurally generate contact surfaces and standing points, then create a navigation path based on the scene map, and finally convert it into the UniHSI task format.

As shown in Fig. 11, contact surfaces are generated according to object categories and interaction types. For sofas, we decomposed the backrest, armrests, and seat cushion by point-cloud height, take the $0.5 \times 0.5$m region at the center front of the cushion as the pelvis-joint contact surface for the Sit task, and set the standing point 0.5m in front of the cushion's front edge center. For beds, we identify the mattress, pillow, and headboard based on the height of the point cloud, with contact normals facing upward. For the Sleep task, the central $0.5 \times 0.5$m region of the mattress is used as the pelvis contact surface, and the pillow's upper surface point cloud as the head contact surface. For the Sit task, we use the $0.5 \times 0.5$m region at the center front of the mattress as the pelvis contact surface, with upward normals and the standing point 0.5m in front of the mattress's front edge center. For Touch tasks, where the target objects are small, we use the object's point cloud as the contact surface for the left or right hand. We uniformly sample 36 candidate standing points along a 0.5m-radius circle around the object and select the first reachable one as the standing position, with the contact normal pointing from the object to the standing point. Tasks of type watch, lift, and composite cannot be executed by UniHSI. Navigation is implemented using the same A* algorithm as in BiBo.

As in Fig. 5, the qualitative evaluation of UniHSI covers two interaction scenarios: (a) getting up from the sofa and (b) leaning on the armrest. In (a), we use the same configuration as the sitting task, placing the standing point 0.5 m in front of the cushion's front-edge center at a height of 0.86 m. In (b), UniHSI and BiBo use the same configuration, with the pelvis and hands set to make contact with the upper part of sofa's armrest.

#### E.1.2 HUMANVLA

HumanVLA (Xu et al., 2024) performs object transportation tasks based on visual observation and natural language descriptions. Its task format consists of a prompt embedding, the initial and target poses of scene objects, the target object name, and the waypoints before and after transportation. At runtime, the model takes as input the current and target poses of the target object, the next waypoint, the rendered scene image, and the BERT (Devlin et al., 2019) embedding of the prompt.

---

[24]https://developer.nvidia.com/isaac-gym
[25]https://xizaoqu.github.io/unihsi/

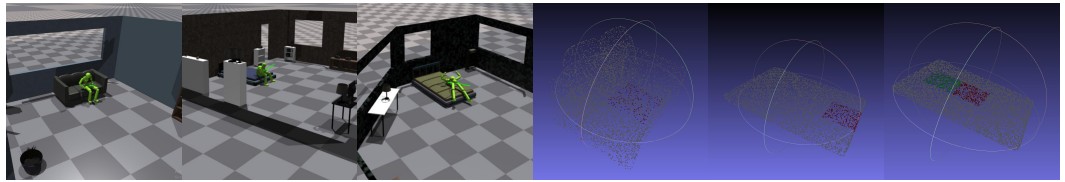

Figure 11: Visualization of task execution process in UniHSI and the ground truth contact surfaces in random generated scene.

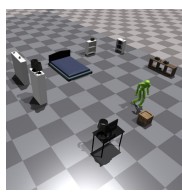 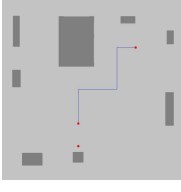 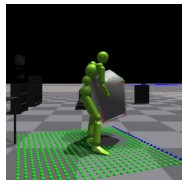 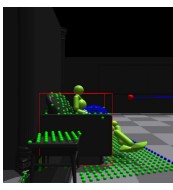 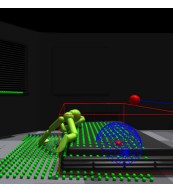

Figure 12: Visualization of Human-VLA and the path planning result.

Figure 13: Visualization of the task execution process in TokenHSI, illustrating the height map and target position.

We use the official repository[26] and the released *HumanVLA-Teacher* model checkpoint (the Student model struggles to complete the task). The randomly generated scenes and tasks are organized according to the HumanVLA task format. Walls are removed, and the asset import options and settings remain consistent with the original scene configuration. Since HumanVLA is sensitive to interaction poses, we modify the A* pathfinding by setting the destination 1.5 m in front of the target object, and adding an extra waypoint 0.5 m in front of the target to encourage a direct approach toward it. The task execution process and pathfinding result are shown in Fig. 12.

During the qualitative evaluation shown in Fig. 5, the agent is initialized at a position of (-3m, 2m) relative to the target, facing along the +y-axis. The waypoint is set at (-1m, 0m) relative to the target. The box has size of $0.36 \times 0.36 \times 0.5$m, with 0.5m representing its height.

### E.1.3 TOKENHSI

TokenHSI (Pan et al., 2025) achieves multi-interaction generalization and integration by fine-tuning task-specific tokenizers and action heads. Its task format consists of scene objects and a task plan, where each step in the plan is defined by an action type and its corresponding parameters. Specifically, the *traj* action includes a list of path waypoints, the *sit* action specifies the interaction point, seat bounding box, position, and orientation, and the *carry* action defines the target box, its bounding box, and the destination position. During execution, the model also samples a local height map around the agent as part of its observation to enhance environmental awareness.

We adopt the official repository[27] and the *Longterm 4 Basic Skills* checkpoint, constructing all tasks according to the TokenHSI format. The scene is first converted into an file structure compatible with TokenHSI, which includes each object's height map, mesh, bounding box, facing direction, up vector, and sit position. When importing scenes, the same asset options as those in the original scene configuration are applied. The constructed tasks follow the same format as in TokenHSI, encompassing object poses and the complete action plan. Trajectory waypoints for *traj* actions are generated using the same A* pathfinding algorithm as in BiBo, while interaction point generation follows the procedure used in UniHSI. The task execution process is illustrated in Fig. 13

### E.1.4 CLOSD

CLoSD (Tevet et al., 2024) use a motion diffusion to drive a tracking policy (Luo et al., 2023) to perform human-scene interaction in physical environment. It is controlled by a rule-based finite state machine (FSM). The FSM determines the next action based on the current state and a programmatic

---

[26]https://github.com/AllenXuuu/HumanVLA
[27]https://liangpan99.github.io/TokenHSI/

script, outputting the subsequent 60-frame action command that includes the motion caption, target positions of the feet, hands, and head, as well as the standing position and facing direction.

We employ the official implementation[28] and the *Multitask* model checkpoint. The state sequence is generated according to the scene objects and task. Specifically, for each interaction task, we define an update function and a transition rule. The update function outputs the next action command and checks whether a transition should occur. When all interactions in the current group meet their transition conditions, the FSM proceeds to the next group of interactions (where interactions within the same group occur simultaneously). The transition rules follow the same criteria of task success, as described in Sec. 4.1.

In the *reach* state update function, the A* algorithm is first applied to perform pathfinding. If the agent is oriented toward the next waypoint within a 45° cone, the target position is set to that waypoint (clipped to a maximum distance of 1.2m), and the motion caption prompt is *"a person is walking."* Otherwise, the facing direction is adjusted toward the next waypoint, and the prompt is set to *"a person is turning around in place."* The configurations for *sit*, *sleep*, and *touch* actions follow those defined in UniHSI. For the *lift* action, the target positions of both hands are initially placed 0.3m to the left and right sides of the object, then directed toward the object's center to facilitate grasping. Finally, the pelvis height is set to 0.86m (the normal standing height) to guide the agent to stand upright. A *reach* state is inserted between consecutive interactions to enable navigation across interaction points.

### E.1.5 ABLATION STUDY

In the ablation study presented in Sec. 4.1, we analyze the impact of several components in both the compiler and executor. In the compiler, we ablate the voting mechanism and the use of image labels, while in the executor, we ablate the use of actual executed motion input, previously generated motion input, and IK post-optimization.

To ablate the voting mechanism, we use a single VLM to analyze motion attributes instead of aggregating outputs from multiple VLM instances. The image label is ablated by prompting the VLM to directly predict image coordinates and facing angles. For the executor, we conduct ablations by extending future motion using only the previously generated motion or the actual executed motion. Specifically, the same motion sequence is provided as input to both the Diffusion Model and the VAE during inference. Finally, we ablate IK post-optimization by directly using the raw generated motion without applying inverse kinematics refinement.

### E.2 MOTION QUALITY

### E.2.1 MOTIONLCM

MotionLCM (Dai et al., 2024) adopts a Transformer-based Latent Diffusion architecture, incorporates Consistency training to enable few-step generation, and integrates control signals through a ControlNet module.

We use MotionLCM v2, adopting the official implementation[29] and following the provided training scripts and configurations to train the VAE, Consistency Model, and ControlNet. In addition to the original TM2T metrics, we introduce two additional evaluation pipelines to assess Physical Plausibility and Control Accuracy.

The Physical Plausibility evaluation follows the implementation provided in CLoSD, whereas the Control Accuracy evaluation is performed under the same experimental setting as BiBo. Specifically, a single joint is randomly selected from the head, hand, or foot, and the ground-truth joint position at frame 40 is used as the control signal. The mean absolute error (MAE) between the generated joint position and the control signal is then calculated to quantify the control error.

Training and evaluation is conducted on a workstation with 8× RTX 4090 GPUs, 256 GB DDR4 RAM, and an Intel(R) Xeon(R) Gold 6226R CPU @ 2.90GHz, running Ubuntu 24.04 with CUDA 11.8 and PyTorch 2.3.

---

[28]https://github.com/GuyTevet/CLoSD
[29]https://github.com/Dai-Wenxun/MotionLCM

### E.2.2 MotionStreamer

MotionStreamer (Xiao et al., 2025) adopts a latent diffusion architecture, where a Transformer-based diffusion model performs next-token prediction, and a CNN-based VAE decoder reconstructs the motion from tokens, enabling the generation of high-fidelity motions of arbitrary length.

We use the official implementation[30]. The original version employs a 272-dimensional motion representation[31], which is incompatible with our evaluation pipeline. Therefore, we modify it to use the 263-dimensional motion representation in HumanML3D, and train the model following the provided scripts and configurations. The generated motions are exported into the evaluation pipeline to assess motion quality and physical plausibility.

Training is conducted on a workstation equipped with 8× RTX A6000 GPUs, 256 GB DDR4 RAM, and an Intel(R) Xeon(R) Gold 6226R CPU @ 2.90 GHz, running Ubuntu 24.04 with CUDA 11.8 and PyTorch 2.3. Evaluation is performed on the same machine as MotionLCM.

### E.2.3 MoGenTS

MoGenTS (Yuan et al., 2024) is based on a masked modeling framework, consisting of a Motion Transformer and a CNN-based vector quantization VAE (VQ-VAE). It introduces a masking mechanism that leverages both spatial skeletal relationships and temporal dependencies, enabling high-quality motion generation. We use the official implementation[32] and the released checkpoints. The generated motions are then exported to the evaluation pipeline to assess motion quality and physical plausibility. The evaluation is conducted on the same machine as MotionLCM.

### E.2.4 MoConVQ

MoConVQ (Yao et al., 2024) is an end-to-end physical motion generator built upon a VQ-VAE architecture, which operates within its own simulation environment. We employ the official implementation[33] and released checkpoint for all experiments. For quantitative evaluation, we directly adopt the results reported in CLoSD to ensure consistency and comparability across methods. In the qualitative experiments illustrated in Fig. 5, we evaluate MoConVQ under two representative prompts: (d) *"A person is boxing with someone, and kicking at the air."* and (e) *"A person is sitting on a sofa and waving a hand above the head."* Since the simulation environment of MoConVQ does not natively support static mesh colliders like sofa, we follow the approach in this repository[34] , and use a rectangular block with the same height as the sofa cushion and very large mass as a substitute. The evaluation is conducted on the same machine as MotionLCM.

### E.2.5 DiP & CLoSD

DiP (Tevet et al., 2024) is a real-time motion diffusion model capable of generating motions of arbitrary length, built upon a Transformer architecture. CLoSD employs a tracking policy to follow the motions generated by DiP in physical environment. DiP extends future motion from the actual executed motion, thereby adapting physical feedback.

We use the official implementation[35]. In the text-to-motion experiments in Tab. 3, we adopt the released *t2m* checkpoint, while in the control accuracy experiments in Tab. 4, we use the *multi-target* checkpoint and adopt the same setting as other comparison methods. The motions generated by DiP and CLoSD are exported using the provided scripts and evaluated through a unified evaluation pipeline as other comparison methods. The evaluation is conducted on the same machine as MotionLCM, using IsaacGym Preview 4.

---

[30]https://github.com/zju3dv/MotionStreamer

[31]https://github.com/Li-xingXiao/272-dim-Motion-Representation

[32]https://github.com/weihaosky/mogents

[33]https://github.com/heyuanYao-pku/MoConVQ

[34]https://github.com/heyuanYao-pku/Control-VAE

[35]https://github.com/GuyTevet/CLoSD

Table 9: Comparison between different VAEs on motion reconstruction error, using HumanML3D. **Bold** and underline is the best and second best, respectively.

| Method | MotionLCM | MoGenTS | VAE | w/ Causal | w/ AE | w/ VQ |
|---|---|---|---|---|---|---|
| FID ↓ | 0.025 | **0.006** | 0.021 | 0.012 | 0.031 | 0.107 |
| MPJPE ↓ | 27.44 | 16.35 | 20.36 | **7.58** | 31.72 | 39.82 |

Table 10: Impact of latent dimension of the Causal VAE on motion reconstruction error.

| Latent Dim | 32 | 64 | 128 |
|---|---|---|---|
| FID ↓ | 0.015 | 0.012 | **0.010** |
| MPJPE ↓ | 8.04 | 7.58 | **6.76** |

### E.2.6 DOUBLE TAKE

Double Take (Shafir et al., 2023) is a diffusion inference strategy that enables a pretrained motion diffusion model to generate motions of arbitrary length. In the first take, it synchronously generates all segments of a long-sequence motion, using gradient guidance during inference to ensure coherent transitions at the connections between adjacent segments. In the second take, it uses masking to separately diffusion-optimize the transitions.

We use the official implementation of the Double Take[36] inference process, employing the released MDM pretrained on HumanML3D with 263-dim motion format. During inference, we follow the Double Take inference process to obtain motion sequences and adopt the HumanML3D evaluation pipeline.

### E.2.7 ABLATION STUDY

In Tab. 4, 5, and 7, we ablate the LDM by directly training the diffusion model on raw motion sequences, and ablate Causal Attention by removing the attention mask during both training and inference. The inputs of actual executed motion and previously generated motion are further ablated by using only one of them during inference, following the same setting described in Sec. E.1.5.

## F SUPPLEMENTARY EXPERIMENTS

### F.1 RENDERING

In the experiments, we use Blender Python API 4.2.0 EEVEE Next pipeline to render the scene images during the planning process.

For visualization, we adopt three approaches:

1. Headless server environments: We perform visualization rendering in Blender using the EEVEE Next pipeline, where humanoid body are represented as skeletal lines.

2. Debugging environments and Fig. 16, 17 and 18: We use the built-in visualization window provided by IsaacGym for rendering.

3. Case study in Fig. 5 and video demonstrations: We conduct simulations in IsaacGym, export the root states and meshes of all objects, and render them in Unity. The humanoid body is reconstructed in Unity[37] based on IsaacGym XML-based humanoid body file.

### F.2 VARIANTS OF VAE

**Setting.** We experimented with several types of VAEs to convert motion sequences into latent tokens for higher reconstruction quality. Specifically, we tested:

1. The original VAE architecture with a Skip-Transformer encoder–decoder structure.

2. Adding causal attention, as described in Sec. 3.3.

3. Introducing causal autoregressive next-token prediction decoding — specifically, like the behavior of GPT. Instead of feeding $N$ zero tokens at once and decoding all $N$ motion frames simultaneously, we iteratively append one zero token at a time and decode $N$ frames across $N$ steps. All other hyperparameters remain identical with the original setting.

---

[36]https://github.com/priorMDM/priorMDM
[37]https://unity.com/

Table 11: Comparison between different VLM variants on task success rate in random generated scene. **Bold** and underline represent the best and second best performance, respectively. Gray color denotes using ground-truth action plan, which is excluded in the comparison.

| Method (%) | Single Interaction | | | | | | Composite Task | | |
|---|---|---|---|---|---|---|---|---|---|
| | Reach ↑ | Watch ↑ | Sit ↑ | Sleep ↑ | Touch ↑ | Lift ↑ | Simple ↑ | Medium ↑ | Hard ↑ |
| BiBo (ours, w/ Sonnet 3.5, $\sim 175B$) | **99.18** | 98.31 | 87.54 | 91.33 | 70.57 | 59.52 | 50.00 | 34.62 | 19.05 |
| BiBo (ours, w/ Qwen2.5-VL, $\sim 72B$) | 98.09 | 98.87 | 72.73 | 78.67 | 68.44 | 55.10 | 25.00 | 11.54 | 14.29 |
| BiBo (ours, w/ GPT-4o mini, $\sim 8B$) | 91.28 | 92.66 | 48.15 | 16.00 | 56.74 | 52.38 | 19.12 | 13.46 | 9.52 |
| BiBo (ours, w/ GPT-4o, $\sim 200B$) | **99.18** | **99.62** | **95.84** | **94.89** | **86.05** | **65.42** | **58.82** | **36.54** | **27.78** |
| BiBo (ours, w/ GT plan) | 98.91 | 99.06 | 96.75 | 93.33 | 87.23 | 70.41 | 61.76 | 44.23 | 42.86 |

---

**Box 11. Prompt Example for Visual Grounding (JSON Coordinates)**

**SYSTEM\*:** You are a precise visual grounding model that outputs JSON coordinates only.

**USER\*:** You are given an image. Your task is to locate the <Object Category> in the image. Please output the approximate **center coordinates** of this object.

The coordinate system follows the image pixel convention:
- The origin (0, 0) is at the top-left corner.
- The x-axis increases to the right.
- The y-axis increases downward.
Return coordinates in pixels relative to the image resolution.

Return only a JSON object, for example:

```
{"x": 180, "y": 240}
```

---

4. Incorporating vector-quantized VAE (VQ-VAE), implemented using an residual-quantization VAE (RQ-VAE) with 1,024 code entries, 6 residual quantizers, and an exponential moving average (EMA) decay factor of $\mu = 0.99$.

We train these VAE models on the train split of HumanML3D, and evaluate FID and Mean Per Joint Position Error (MPJPE) on the test split. FID evaluates the similarity between the distribution of the original and reconstructed dataset, and MPJPE evaluates the reconstruction error.

**Results.** According to the experimental results in Tab 9, the (2) Causal Attention configuration achieves the best performance, with a MPJPE of $7.58$mm. We further test models with different latent dimensions under the Causal Attention setting, and the results in Tab. 10 show that larger latent dimensions further improve performance. To balance computational efficiency and reconstruction accuracy, we choose a latent dimension of 64.

## F.3 VARIANTS OF OFF-THE-SHELF VLMS

**Setting.** We evaluate BiBo's scaling capability across VLMs of different sizes, and assess the generalizability of prompt design. Specifically, we use the large-scale Claude Sonnet 3.5 ($\sim 175$B), the medium-scale Qwen2.5-VL ($\sim 72$B), and the small-scale GPT-4o mini ($\sim 8$B). The parameter size of Qwen2.5-VL is publicly available (Bai et al., 2025), while the sizes of Claude Sonnet 3.5 and GPT-4o mini are referenced from Abacha et al. (2024). In implementation, we directly replace the API endpoint of GPT-4o with that of each comparison VLMs. Each task is evaluated once under a randomly initialized pose.

**Results.** The results in Tab. 11 show that when using the prompt originally designed for GPT-4o, BiBo achieves a comparable task success rate on the similarly scaled Claude Sonnet 3.5, demonstrating generalization capability. As the size of the VLM increases, the task success rate improves, reflecting scaling ability.

---

**Box 12. Prompt Example for Visual Grounding (Numerical Labels)**

**SYSTEM\*:** You are a precise visual grounding model that identifies numerical labels in an image and outputs only structured results.

**USER\*:** You are given an image that contains 64 numerical labels (each labeled with a number from 1 to 64).

Your task:
- Select the label that lies on the visible surface of the <Object Category>.
- Avoid choosing any number that lies on occluding or overlapping objects that block the target.
- Then, output only one label number (an integer between 1 and 64) that best fits the requirement.

Output format:
- Wrap your final answer between >>> and <<< for easy parsing.

---

### F.4 Accuracy of the Three-Stage VQA

**Setting.** For basic motion attribute analysis, we incorporate a voting mechanism to enhance robustness. In the agent pose reasoning process, a series of textual label descriptions (e.g., object orientations predicted by Orient Anything) are introduced to improve the VLM's spatial understanding. During key joint position generation, we overlay a grid of numerical labels on the image to facilitate visual grounding.

We conduct ablation studies on these components to evaluate their individual contributions. The voting mechanism is ablated by using only a single VLM instance for analysis, and the textual label descriptions are ablated by using only the raw image labels without any additional contextual information in the prompt. The effect of the label grid is examined by comparing the joint generation accuracy with and without the grid, using the prompts shown in Box 11 and Box 12, respectively.

The experimental designs are specified as follows:

1. **Basic Attribute Analysis.** We randomly sample 100 test cases during the task evaluation process and manually annotate the corresponding ground-truth plans. Each test case includes a motion caption, an agent state, a scene observation, and a ground-truth plan. The VLM is prompted as described in Box 4 and Box 5. The output of the VLM is considered correct if it matches the ground-truth annotation for each motion attribute.

2. **Agent Pose Reasoning.** Following the same evaluation process as in (1), we select 100 interaction cases involving the agent's position and facing direction. For the *watch* task, we exclude the evaluation of agent–anchor distance since multiple plausible spatial configurations may exist.

3. **Key Joint Generation.** This is formulated as a visual grounding task. We construct test cases using the COCO 2017 dataset (Lin et al., 2014)[38] and evaluate whether the coordinates predicted by the VLM fall within the ground-truth instance boundaries, allowing a tolerance of 10 pixels. Only images that contain a category with exactly one instance are used, where the instance area should occupy 4%–25% of the entire image. Based on these criteria, we filter 500 samples from the 5,000-image validation split and perform the evaluation using GPT-4o mini for cost efficiency.

**Results.** According to the experimental results in Tab. 12, 13 and 14, the voting mechanism achieved a relative improvement of 10.3% in basic attribute analysis accuracy. The combination of Orient Anything and the rule-based textual label description yielded a 31.9% relative improvement in pose reasoning accuracy—an enhancement primarily observed in the model's understanding of object orientation. Meanwhile, the label grid led to a 55.9% relative improvement in localization performance,

---

[38]https://cocodataset.org/

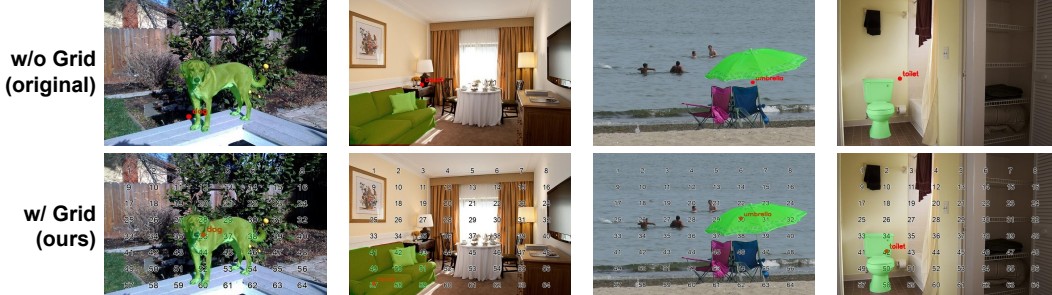

Figure 14: Impact of the label grid on visual grounding of VLM. The result shows that label grid effectively enhance the grounding accuracy.

Table 12: Impact of voting mechanism on motion attribute analyzing accuracy.

| Method | Accuracy↑ |
|---|---|
| Analyzing (w/o Voting) | 78% |
| Analyzing | **86%** |

Table 13: Impact of label description on agent pose reasoning accuracy.

| Method | Accuracy↑ |
|---|---|
| Pose (w/o Description) | 69% |
| Pose | **91%** |

Table 14: Impact of label grid on key joint position generation accuracy.

| Method | Accuracy↑ |
|---|---|
| Joint (w/o Label Grid) | 42.2% |
| Joint | **65.8%** |

which aligns with the conclusions of Yang et al. (2023) and Chae et al. (2024). The visualization of the grounding results are shown in Fig. 14.

### F.5 SUPPLEMENTARY ABLATION STUDY

**Setting.** For the compiler, we employ a VLM for planning, thereby incorporating visual information. During the reasoning process, a rule-based reflection mechanism (refer to Sec. C.2.6) is integrated for error detection and self-correction. For the executor, an additional moving-speed control signal is introduced. We evaluate the contributions of these components through ablation and test the task success rate after each ablation. Each task runs once in the ablation groups.

The visual information ablation is performed by removing all image inputs in the first two VQA stages while retaining the label descriptions (e.g., the relative distance and orientation of each label with respect to the anchor). Since key joint generation is based on image, we keep visual information in the last VQA stage. To ablate the reflection mechanism, all rule-based detections are removed. For the moving-speed ablation, the corresponding condition mask in the motion diffusion is set to zero, and all speed-related descriptions (e.g., walking slowly, turning slowly, refer to Sec. C.3.1) are removed from the compiler.

**Results.** The results in Tab. 15 show that visual information and reflection both contribute to improving the planning quality of the compiler, while the moving speed control effectively enhances locomotion in the scene.

### F.6 CONTROL ACCURACY

**Setting.** First, we evaluate the control accuracy of BiBo on HumanML3D. The control conditions include the facing direction measured in $rad$, the moving speed measured in $m/s$, and the joint positions measured in $m$. The control error is represented using the Mean Absolute Error (MAE). Second, a sine-cosine encoding is adopted for the facing direction during training. We compare the training convergence curves of raw facing angle and the sine-cosine representation.

**Results.** The results in Tab. 16 demonstrate that the control parameters provide effective spatial guidance for motion generation. As shown in Fig. 15, employing sine–cosine facing direction encoding improves both the training stability and convergence.

Table 15: The impact of visual information, reflection and speed control on task success rate. **Bold** and underline represent the best and second best performance.

| Method (%) | Single Interaction | | | | | | Composite Task | | |
|---|---|---|---|---|---|---|---|---|---|
| | Reach ↑ | Watch ↑ | Sit ↑ | Sleep ↑ | Touch ↑ | Lift ↑ | Simple ↑ | Medium ↑ | Hard ↑ |
| BiBo (ours, w/o Visual Info.) | 96.19 | 89.83 | 84.18 | 90.67 | 76.24 | 62.93 | 54.41 | 30.77 | 21.43 |
| BiBo (ours, w/o Reflection) | 96.46 | 95.48 | 86.87 | 81.33 | 81.56 | 61.56 | 50.00 | 32.69 | 23.81 |
| BiBo (ours, w/o Speed Control) | 73.57 | 84.18 | 72.73 | 69.33 | 51.77 | 27.89 | 38.24 | 15.38 | 9.52 |
| BiBo (ours) | **99.18** | **99.62** | **95.84** | **94.89** | **86.05** | **65.42** | **58.82** | **36.54** | **27.78** |

Table 16: The control error (MAE) of BiBo on HumanML3D dataset, including facing direction (Rot.), moving speed (Vel.) and joint positions.

| Cond | Rot. (rad) | Vel. (m/s) | Joint Position (m) | | | | | |
|---|---|---|---|---|---|---|---|---|
| | | | Pelvis | Head | L Hand | R Hand | L Foot | R Foot |
| MAE ↓ | 0.177207 | 0.015499 | 0.024649 | 0.033833 | 0.054909 | 0.057240 | 0.032403 | 0.032687 |

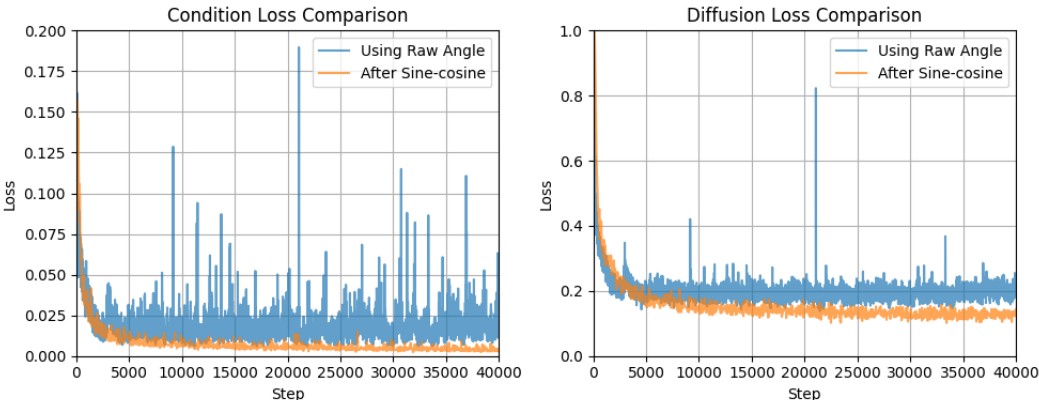

Figure 15: Visualization of the training loss variations with and without using sine–cosine encoding of the facing direction. For both the Diffusion Loss and Condition Loss, the sine–cosine encoding yields more stable training and faster convergence.

## F.7 HYPERPARAMETER SELECTION

**Setting.** We investigate the effects of different learning strategies and model sizes on motion generation. For the learning rate, we experiment with 5e-4, 1e-3, and 2e-3. For the model size, we select hidden dimensions of 128, 256, and 512, with the feed-forward layer dimension scaled proportionally. For the 512-dimensional model, we use 8 attention heads. All other parameters remain consistent with the current configuration.

We evaluate both computational efficiency and generation quality. Computational efficiency is measured by the number of parameters and AITS, while generation quality is assessed using FID and Top-1 ∼ 3 R-Precision.

**Results.** As shown in Tab. 17, the current configuration (256 hidden dimension, 1024 feed-forward dimension, 4 attention heads, and a 5e-4 learning rate) achieves the best generation quality, while maintaining computational efficiency comparable to smaller models. Due to the limited dataset size, larger models significantly increase computational cost without yielding better performance.

## F.8 MOTION PREFIX

**Setting.** We prepend a prefix to all motion sequences in the dataset. This strategy improves data utilization efficiency and enhances the model's ability to initiate motions from scratch or dynamically transfer to new motions. We explore three different prefix augmentation strategies:

1. Adding an all-zero prefix.
2. Extracting a stance motion frame from the dataset and replicating it for 20 frames.

Table 17: Impact of hyperparameter combinations on generation efficiency and motion quality, using HumanML3D. **Bold** and underline denotes the best and second best performance.

| Hid. Dim. | F.F. Dim. | Head | LR | Efficiency | | Motion Quality | | | |
|---|---|---|---|---|---|---|---|---|---|
| | | | | Params (M) ↓ | AITS↓ | FID↓ | R.P.1↑ | R.P.2↑ | R.P.3↑ |
| 256 | 1024 | 4 | 5e-4 | 28.541 | 0.047 | **0.076** | 0.542 | **0.738** | **0.829** |
| 256 | 1024 | 4 | 1e-3 | 28.541 | 0.047 | 0.094 | 0.536 | 0.729 | 0.822 |
| 256 | 1024 | 4 | 2e-3 | 28.541 | 0.047 | 0.091 | **0.543** | 0.735 | 0.826 |
| 128 | 512 | 4 | 5e-4 | **20.584** | **0.044** | 0.127 | 0.505 | 0.703 | 0.804 |
| 512 | 2048 | 8 | 5e-4 | 59.495 | 0.061 | 0.085 | 0.534 | 0.732 | 0.827 |

Table 18: Comparison of different prefix strategies under two preset conditions.

| Method | Ground-truth Motion Preset | | | | Prefix Motion Preset | | | |
|---|---|---|---|---|---|---|---|---|
| | FID↓ | R.P.1↑ | R.P.2↑ | R.P.3↑ | FID↓ | R.P.1↑ | R.P.2↑ | R.P.3↑ |
| No Prefix | **0.068** | **0.552** | **0.747** | **0.834** | 0.156 | 0.492 | 0.682 | 0.786 |
| All-zero Prefix | 0.086 | 0.511 | 0.713 | 0.809 | 0.106 | 0.501 | 0.703 | 0.799 |
| Learnable Prefix Token | 0.123 | 0.523 | 0.721 | 0.818 | 0.148 | 0.504 | 0.704 | 0.803 |
| Stance Motion Prefix | 0.076 | 0.542 | 0.738 | 0.829 | **0.087** | **0.525** | **0.721** | **0.816** |

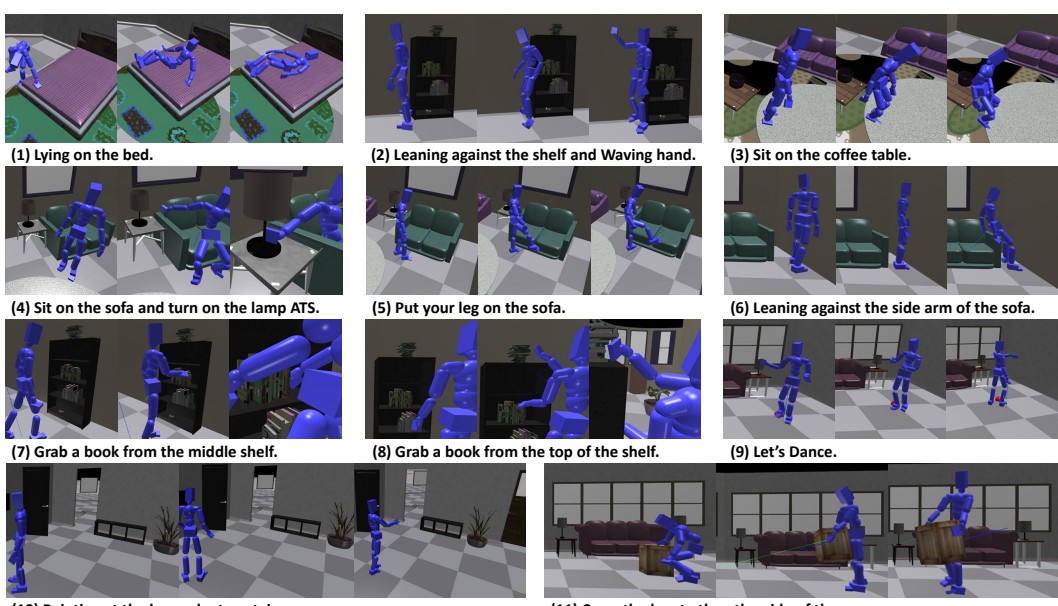

(1) Lying on the bed.  (2) Leaning against the shelf and Waving hand.  (3) Sit on the coffee table.

(4) Sit on the sofa and turn on the lamp ATS.  (5) Put your leg on the sofa.  (6) Leaning against the side arm of the sofa.

(7) Grab a book from the middle shelf.  (8) Grab a book from the top of the shelf.  (9) Let's Dance.

(10) Pointing at the large plant container.  (11) Carry the box to the otherside of the room.

Figure 16: Visualization of BiBo in IsaacGym. BiBo can perform one type of interaction across multiple object categories (in 3, 4), and execute multiple types of interactions on the same object (in 4, 5, 6). It can generate diverse motions conditioned on text (in 9), and supports precise control (in 7, 8), composite action (in 2, 4), long-range interactions (in 10) and manipulation (in 11).

3. Introducing a learnable prefix token.

We evaluate the generation quality on the HumanML3D dataset. During testing, we preset an initial motion segment and let the model to iteratively extend the future motion, after which the preset portion is trimmed off to compute FID and R-Precision. Two preset motion conditions are tested: (1) using the first 20 frames of ground-truth motion (as in Tevet et al. (2024)), and (2) using the newly added prefix.

**Results.** Experimental results in Tab. 18 show that introducing a prefix enhances the model's ability to generate motion from scratch, rather than relying on a preset motion history. Among the tested strategies, the stance-motion prefix yields the best performance.

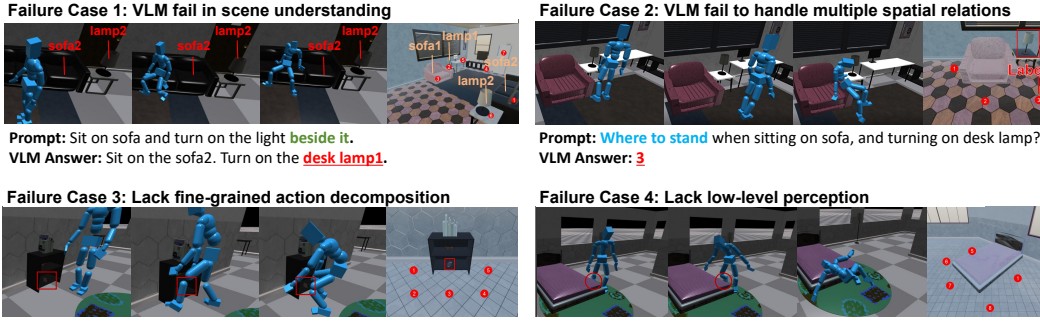

Figure 17: Visualization of failure cases of BiBo. Case 1 and 2 are related to the limitation of VLM, and Case 3 and 4 are related to the lack of explicit scene geometry modeling (e.g. scene voxel, point cloud) in executor.

## F.9 CASE STUDY

**Basic Interactions.** We evaluate BiBo's ability to perform various types of interactions, including basic interactions (e.g., sit, sleep), text-driven motion generation (e.g., dance), long-range interactions (e.g., pointing at), composite actions, and manipulation tasks (e.g., transport).

The prompts used for testing and the corresponding execution results in the simulator are shown in Fig. 16. According to the results, BiBo demonstrates diversity in interaction targets (in 3, 4), interaction modes (in 4, 5, 6), and control modalities (in 9). It also supports precise control (in 7, 8), composite actions (in 2, 4), long-range interactions (in 10), and manipulation tasks (in 11).

For more visual demonstration, please visit our Hugging Face repository [39].

**Failure Cases.** We visualize representative failure cases during task execution. The results are shown in Fig. 17. In case (1), the VLM misinterprets the scene layout (or mismatches textual labels and scene objects), leading to the selection of an incorrect interaction target. In case (2), the VLM focuses only on the agent's position relative to the desk lamp while ignoring the sofa's orientation. In cases (3) and (4), because the executor passively receives physical feedback rather than actively parsing the geometric structure of the scene, it shows limitations in fine-grained action decomposition and proactive obstacle avoidance.

Our method still has limitations, but we believe that as the multimodal capabilities of off-the-shelf VLMs and 3D encoding technologies continue to advance, these problems are expected to be resolved in the near future.

**Ablation of Speed Control.** During experiment, we observe that the moving speed not only enhances controllability, but also serves as a signal indicating whether to continue the motion or to stop the current one, which enhance motion coherence.

As shown in Fig. 18, in (1), the agent with speed control precisely reaches the target location, while the agent without speed control overshoots the target and then turns back. In (2), the agent with speed control keeps walking status between successive walking motions, while the agent without speed control tends to pause between successive walking motions, as it cannot determine whether to stop or keep moving. The agent with speed control demonstrates better controllability and motion coherence.

## F.10 DESIGN OF USER STUDY

We design a User Study questionnaire to assess the perceived quality of the generated results. Each questionnaire consists of 5 groups of motions and interactions, and each group contains three samples, generated by BiBo, MotionLCM, and DiP (CLoSD) using the same prompts, respectively.

The prompts are randomly selected from the HumanML3D dataset, and each questionnaire is unique. The three samples are randomly arranged from left to right and displayed using a con-

---

[39]https://huggingface.co/Behavia/BEHAVIA

**(1) Controllability**

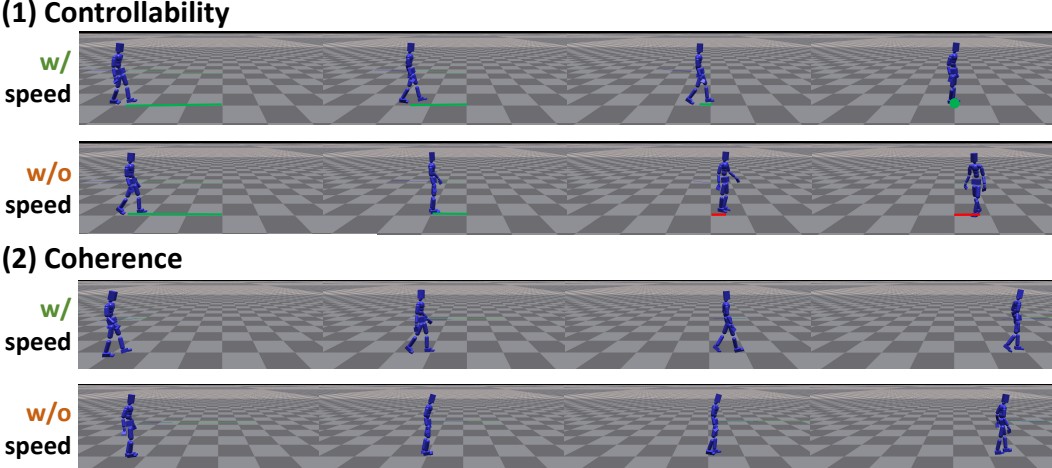

**(2) Coherence**

Figure 18: Visualization of locomotion with and without moving-speed control. In (1), the agent with speed control precisely reaches the target location, enhancing controllability. In (2), the agent without speed control tends to stand still between successive walking motions, as it cannot determine whether to stop or keep moving, lacking coherence.

Table 19: Comparison of BiBo and TESMO on motion quality (FID, R-Precision, Diversity), physical plausibility (Skating) and control accuracy (Goal Reaching Error and Orientation Error). BiBo demonstrates enhanced perfomance compared with TESMO.

| Method | FID ↓ | R-Precision ↑ | Diversity → | Foot Skating ↓ | Goal Reaching Error ↓ | Orientation Error ↓ |
|---|---|---|---|---|---|---|
| Ground Truth | 0 | 0.514 | 9.503 | 0.21 | 0 | 0 |
| TESMO | 20.465 | 0.376 | 6.415 | 0.56 | 0.1445 | 0.2410 |
| BiBo (ours) | **0.076** | **0.542** | **9.606** | 0.74 | **0.0246** | **0.1772** |
| BiBo (ours, Phys) | 1.883 | 0.411 | 8.298 | **0.01** | - | - |

sistent skeletal structure with line segments. The range of motion on the XZ-axes is represented by a gray ground. To prevent bias, the samples are labeled only with their prompts, without indicating the generation method.

Volunteers are asked to select the best-generated sample within each group. The corresponding method receives one point for each selection. A total of 30 participants evaluate 450 samples (150 per method), with a maximum of 150 points available. The questionnaire is hosted on the Google Docs [40] platform, with an example provided in the Fig. 19. The results are shown in Tab. 6

### F.11 ADDITIONAL COMPARISON METHODS

#### F.11.1 NON-PHYSICAL HUMAN SCENE INTERACTION

We compare the generation quality, physical plausibility, and controllability with non-physical human-scene interaction methods. We use TESMO (Yi et al., 2024) as the comparison method, with the data sourced from the original paper.

The results in Tab. 19 demonstrates that BiBo effectively enhances motion quality (20.465→0.076) and control accuracy (0.1445→0.0246) compared with TESMO.

#### F.11.2 RL-BASED METHODS

RL-based methods like Wang et al. (2024a) based on a fixed skill set, which may be limited in diversity and motion naturalness. BiBo uses a VLM to guide a diffusion generator, and the generated motions are tracked by an RL policy. This method generates diverse and natural motions, while maintaining scene awareness and physical plausibility.

---

[40]https://docs.google.com/forms/

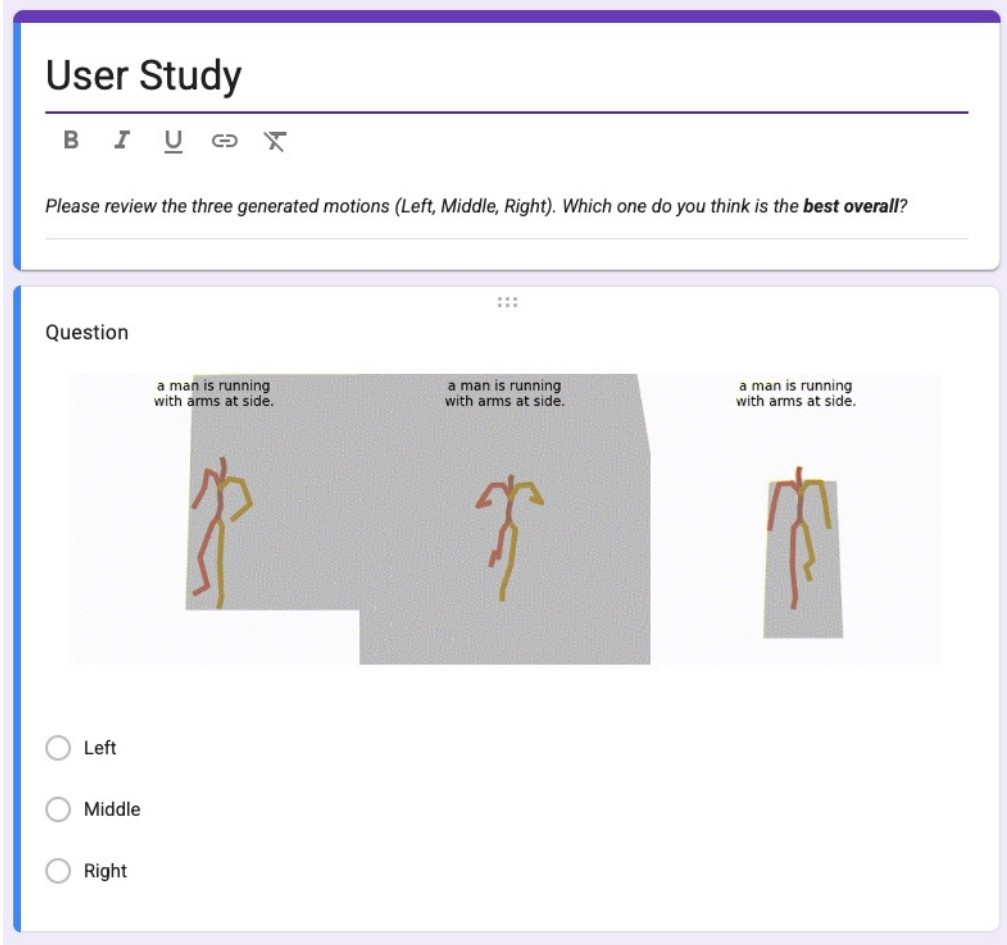

Figure 19: An example of the User Study questionnaire on Google Docs. Each questionnaire consists of 5 groups of generated samples. Each group contains 3 samples generated by different methods, as shown in the figure. Volunteers are asked to select the sample with the best quality to vote for the corresponding method.

Table 20: Comparison of executable skills between BiBo and RL-based methods. BiBo is capable of performing diverse motions.

| Method | Idle, Walk | Jump, Jog, Crawl | Sit, Lie | Punch, Kick | Lift, Carry | Watch, Pointing at |
|---|---|---|---|---|---|---|
| HumanVLA (Xu et al., 2024) | ✓ | ✗ | ✗ | ✗ | ✓ | ✗ |
| UniHSI (Xiao et al., 2023) | ✓ | ✗ | ✓ | ✗ | ✗ | ✗ |
| SIMS (Wang et al., 2024a) | ✓ | ✗ | ✓ | ✗ | ✓ | ✗ |
| BiBo (ours) | ✓ | ✓ | ✓ | ✓ | ✓ | ✓ |

**Skill.** We first qualitatively compare the capabilities of the methods. As shown in the Tab. 20, RL-based methods typically have a limited skill set, while BiBo can generalize to a broader range of interaction skills.

**Motion Quality.** We further compare the motion quality with RL-based methods. Specifically, we place a single interactive object in the scene and initialize the humanoid agent at a random position in front of it, facing the object. The control signals are generated by the program. We convert the collected motion into the HumanML3D format and calculate its FID as an indicator of motion quality.

We choose InterPhys, UniHSI, and SIMS as comparison methods, with the data sourced from SIMS. The results in Tab. 21 demonstrate that BiBo's design improves the naturalness of motion for LLM-based humanoid agents.

Table 21: Comparison of motion quality for Sit, Lie, and Carry actions.

| Method | Sit ↓ | Lie ↓ | Carry ↓ |
|---|---|---|---|
| InterPhys (Hassan et al., 2023) | - | - | 81.0 |
| UniHSI (Xiao et al., 2023) | 153.84 | 211.22 | - |
| SIMS (Wang et al., 2024a) | 125.66 | 171.24 | 65.14 |
| BiBo (ours) | **68.78** | **134.46** | **34.72** |

Table 22: Impact of motion context lengths on Top-1 R-Precision.

| Motion Context Length | Top-1 R-Precision |
|---|---|
| 1 Frame | 0.423 |
| 20 Frames | 0.542 |
| (20 + 40 + 80) Frames | **0.564** |

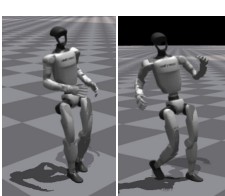 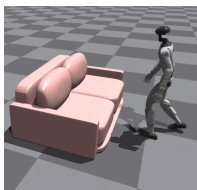 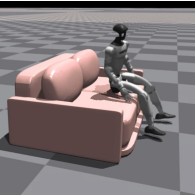 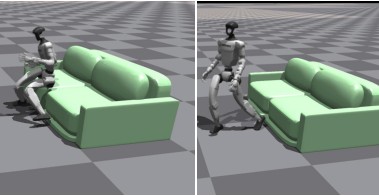

Start dancing         Sit on the sofa         Stand up from the sofa

Figure 20: Visualization of the execution results of BiBo on the Unitree G1 robot platform. The robot can perform diverse motions and is capable of performing physical scene interaction.

## F.12 LENGTH OF THE PREVIOUS MOTION CONTEXT

**Setting.** BiBo extends future motion using a 20-frame prefix composed of both generated and executed motion, which ensures motion continuity. However, we are also aware that a 20-frame prefix may be insufficient for maintaining long-horizon continuity.

Increasing the prefix length is a promising direction for future work. We provide some preliminary experiments using longer context. Instead of simply concatenating all past motion, we follow the idea of FramePack (Zhang & Agrawala, 2025) and construct motion context at multiple frame rates to improve efficiency. The evaluation is conducted on the HumanML3D dataset, comparing the generated results in terms of Top 1 R-Precision.

The results in Tab. 22 show that longer context introduce better text consistency (+4.1%), which also indicates better long-term continuity.

## F.13 EVALUATION ON ROBOTIC PLATFORM

Our proposed method focuses on physical Human Scene Interaction (HSI) modeling, aiming to leverage off-the-shelf VLMs for humanoid agents. The potential downstream applications of this approach extend to both computer animation and real-world humanoid robot control. While our method's motivation differs from traditional robot operating systems, real-world validation can provide valuable insights into its downstream application potential.

Therefore, we tested BiBo on the robot body. First, we attempted to transfer the model to the robot body. Then, we performed real-world deployment. Deploying to a real-world robot requires additional processes such as SLAM and low-level joint motor development, which require significant time, resources, and close collaboration with the robot manufacturer. Due to resource constraints, we only deployed the Embodied Instruction Compiler.

### F.13.1 MIGRATING TO HUMANOID ROBOT BODY

**Implementation.** We keep the Embodied Instruction Compiler and motion diffusion unchanged, while changing the Tracking Policy from the SMPL body to the robot body. Specifically, we use the Unitree G1 humanoid robot platform [41]. The Unitree G1 has 29 degrees of freedom (DOF), including 12 leg joints and 17 upper body joints, but no head joints.

---

[41]https://www.unitree.com/g1

Table 23: Task success rates of BiBo on Unitree G1 platform. BiBo on robotic platform shows comparable performance with methods on virtual characters.

| Method | Body | Reach ↑ | Sit ↑ | Stand ↑ |
|---|---|---|---|---|
| CLoSD (Tevet et al., 2024) | SMPL | **100%** | 88% | **98%** |
| BiBo (ours, on robot) | Unitree G1 | **100%** | **94%** | 86% |

Table 24: Comparison of task success rate in real-world environment.

| Task | Reach ↑ | Watch ↑ | Sit ↑ |
|---|---|---|---|
| LeVERB (Xue et al., 2025) | 20 / 20 | - / - | 1 / 20 |
| HumanoidVLA (Ding et al., 2025) | 9 / 10 | 10 / 10 | - / - |
| BiBo (ours) | 10 / 10 | 10 / 10 | 8 / 10 |

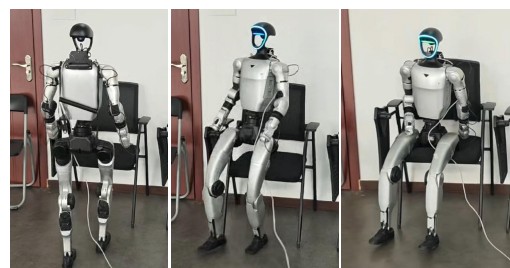

Sit on the chair in the middle

Figure 21: Visualization of the results of Embodied Instruction Compiler in real-world scene.

We train the Tracking Policy using the Unitree G1 training repository provided by PHC (Luo et al., 2023)[42], and refer to the official Unitree RL Gym[43]. Following CLoSD (Tevet et al., 2024), we train the Tracking Policy on the generated motions and basic tasks. The proposed Motion Diffusion generates a 263-dimensional joint feature in HumanML3D format. We first convert the joint features into 22 joint coordinates, and then use Smplify (Bogo et al., 2016)[44] to fit the SMPL (Loper et al., 2015)[45] skeleton. The SMPL skeleton includes 24 joints, and the features include the bone rotations. Specifically, we use Smplify3D[46]. By using Retarget2Humanoid[47], we redirect the obtained SMPL skeleton to the G1 robot body and iteratively optimize the rotations for the G1's 29 joints. Note that the original PHC uses a 39 DOF Unitree G1 body with 19 rotatable joints in the motion dataset.

**Results.** We test the robot's motion execution and scene interaction capabilities using IsaacGym. Specifically, we evaluate reach, sit and stand. Each task is tested 50 times under different initial position conditions. The task success rate criteria remain consistent with CLoSD.

Fig. 20 shows the visualization of the robot's execution results. The results in Tab. 23 indicate that while the robot exhibits motion diversity, it is also adaptable to the physical scene interaction.

### F.13.2 REAL-WORLD DEPLOYMENT OF EMBODIED INSTRUCTION COMPILER

**Implementation.** Deploying the Embodied Instruction Compiler onto real hardware requires odometry, visual sensors, and motion control.

The MID360 Lidar SLAM service provided by the manufacturer is unavailable on our robot. As an alternative, we use a plane-matching algorithm to match wall surfaces based on Lidar point clouds, enabling rough localization.

The Unitree G1 is equipped with a RealSense D435i RGBD depth camera on its head. However, the head cannot rotate, and the downward angle (48°) is too low for scene image capturing. Therefore, we attaches a stationary RealSense D435i camera to observe the scene. The camera is calibrated using visual features.

The experimental environment is relatively narrow, and some RL-policy models only take the target location as input, which leads to low controllability and increases the risk of equipment damage. For safety consideration, we use the motion controller from the manufacturer. The controller includes basic locomotion and sit actions, which are invoked via the RPC interface to control the robot's speed or movements.

---

[42]https://github.com/ZhengyiLuo/PHC

[43]https://github.com/unitreerobotics/unitree_rl_gym

[44]https://smplify.is.tue.mpg.de/

[45]https://smpl.is.tue.mpg.de/

[46]https://github.com/GuyTevet/motion-diffusion-model

[47]https://github.com/song-siqi/retarget2humanoid

Table 25: Comparison of task success rate of UniHSI and BiBo on ScanNet ScenePlan dataset. The results show that BiBo achieves plausible generalization on real-world scanned scene.

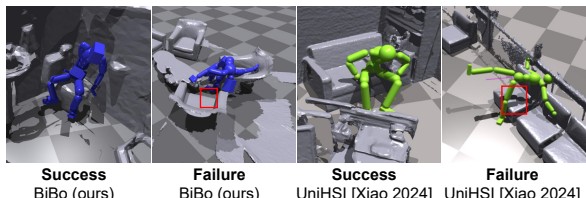

| Method | Simple ↑ | Middle ↑ | Hard ↑ |
|---|---|---|---|
| UniHSI | 73.2 | 43.1 | 22.3 |
| BiBo (ours) | **81.7** | **65.2** | **41.5** |

Figure 22: Success and failure cases of BiBo and UniHSI on ScenePlan.

Table 26: Comparison of interaction error between different methods. BiBo demonstrates precise interaction capability even without specifically RL-learning for contact tasks. IK is essential for precise interaction.

| Method | Pelvis ↓ | Left Hand ↓ | Right Hand ↓ |
|---|---|---|---|
| UniHSI | 0.126 | 0.181 | 0.094 |
| BiBo (ours) | 0.109 | **0.079** | **0.080** |
| BiBo (ours, w/o IK) | **0.107** | 0.101 | 0.099 |

Table 27: Comparison of R-precision for scene interactions across different methods. Higher R-precision indicates better alignment with the instructions, demonstrating stronger coherence. According to the results, BiBo exhibits instruction coherence during interaction.

| Method | Sit ↑ | Lie ↑ | Carry ↑ |
|---|---|---|---|
| UniHSI | 17.12% | 9.59% | - |
| BiBo (ours) | **25.34%** | **17.81%** | **21.23%** |

Unlike in virtual scene, the real-world environment does not have object instance annotations. We additionally use the TinySAM (Shu et al., 2025)[48] model for real-time zero-shot image segmentation. VLM annotates a SAM prompt (a point) on the label grid-enhanced image, thereby extracting the mask for the target instance. The segmented images are then fed into Orient Anything for orientation prediction. Path planning is performed by generating a height map from Lidar point clouds, which is then thresholded to create an obstacle map.

**Results.** We test the robot's success rates on various tasks in real-world scenarios, as in Fig. 21. The move and watch tasks validate the compiler's understanding of the scene and object attributes, while the sit task validates the robot's ability to interact with the scene. We use chairs with different shapes as target object, and specify them by natural language description (e.g., the chair without tray, the chair in the middle). Due to the limitations of the motion controller, sitting down requires a combined action of moving backward and sitting. As a result, we placed the chair against the wall.

The results in Tab. 24 show that the proposed compiler achieves plausible results in scene understanding and interaction. The robot succeeds in all the walk and watch tasks, and completes the sit task with a high success rate. The failure cases in the sit task are mostly due to inaccurate localization (as we use a simple algorithm for odometry). The robot's stance deviated from the front of the chair, preventing it from completing the sitting action.

## F.14 Additional Scene Interaction Evaluation

### F.14.1 Additional Benchmark

**Setting.** We conduct experiments on ScanNet ScenePlan benchmark from UniHSI. We evaluate each task three times under different initial conditions. For more details of ScenePlan dataset, please refer to Sec. B.3.

**Results.** The experimental results in Tab. 25 show that BiBo achieves plausible success rates across tasks of varying difficulty, and generalizes well to real-world scene data from ScanNet.

**Failure Cases.** We analyze the failure cases of BiBo and UniHSI. As shown in Fig. 22, most of BiBo's failures come from collisions with narrow scene areas, causing the agent to fall. We attribute this to BiBo's executor passively accepting environmental feedback rather than actively observing the environment, which is a direction worth exploring in the future. On the other hand, while UniHSI actively perceives the environment through a height map, it is trained on PartNet with specific object

---

[48]https://github.com/xinghaochen/TinySAM

shapes and relies on manual annotations. As a result, it struggles with generalizing across different scenes and objects in ScanNet, resulting in abnormal movements.

### F.14.2 INTERACTION ACCURACY

**Setting.** While the Contact Error proposed by UniHSI provides valuable insights, it remains limited in accurately reflecting interaction accuracy. Specifically, Contact Error is defined as follows:

$$\text{Contact Error} = \sum_{i, c_i \neq 0} e_i / \sum_{i, c_i \neq 0} 1, \qquad e_i = \begin{cases} ||\mathbf{d}_{k_i}||, & c_i = \text{contact} \\ \min(0.3 - ||\mathbf{d}_{k_i}||, 0). & c_i = \text{not contact} \end{cases} \tag{19}$$

$c_i$ represents an action, and $\mathbf{d}_{k_i}$ denotes the shortest distance from the joint to the surface point cloud of the target object. $c_i$ includes navigation waypoint, whose errors $e_i$ are predominantly 0. When the number of navigation waypoints increases, the contact error decreases, despite no actual improvement in interaction accuracy.

Therefore, we propose Interaction Error, which is more closely related to the interaction, excluding the interference of navigation waypoints:

$$\text{Interaction Error} = \sum_{i, c_i = \text{contact}} e_i / \sum_{i, c_i = \text{contact}} 1, \qquad e_i = ||\mathbf{d}_{k_i}|| \tag{20}$$

We test the Interaction Error of three frequently used joints — pelvis, left hand, and right hand — during contact interactions. Non-contact interaction accuracy evaluation can be referred in Sec. F.6, which achieves high control accuracy.

The experiments are conducted on interactions related to the specified joints. Pelvis is tested with sit and sleep task, while hands are tested with touch task.

**Results.** According to the experimental results in Tab. 26, BiBo outperforms UniHSI, which is specifically designed for contact tasks, in interaction accuracy. There are two reasons:

1. IK effectively improves the interaction accuracy of end effectors.
2. BiBo uses a Tracking Policy, which dynamically adjusts actions to achieve higher precision in interactions.

We also observe that the RL-learning in UniHSI led to lower interaction accuracy for the left hand compared to the right hand. BiBo uses motion diffusion trained on mirror augmented data, achieving high interaction accuracy for both the left and right hands.

### F.14.3 INTERACTION NATURALNESS

We evaluate the motion quality of different HSI methods. For more details, please refer to Sec. F.11.2 and Tab. 21.

### F.14.4 INTERACTION COHERENCE

**Setting.** Task Coherence is assessed via R-Precision. R-Precision quantifies the probability of retrieving the correct motion within a data batch, given a textual motion caption. For interaction motions, R-Precision indicates whether the execution conforms to the task description; for example, Carry should involve holding the object between both hands rather than pushing it.

To evaluate R-Precision, we collect motions from physics-based interaction processes. Since R-Precision computation requires batches containing diverse motion categories, we randomly substitute one sample in each real-world data batch with an interaction motion, and perform retrieval using the corresponding task instruction.

**Results.** According to the results in Tab. 27, BiBo generates interaction motions consistent with instructions. This demonstrates that the proposed method possesses high task coherence.

Table 28: Comparison of generation quality (FID) and diversity on the AMASS dataset. → indicates that closer to ground truth is better. The results show that BiBo generates motions with plausible FID without fine-tuning, and approaches the real world motion distribution.

Table 30: Comparison of generation quality (FID) across each subset of AMASS (Part 2). The results show that BiBo demonstrates advantages on every subset, and datasets containing more real motions exhibit greater similarity to the motion distribution generated by BiBo.

| Dataset | G.T. | BiBo (ours) | DiP |
|---|---|---|---|
| Overall FID ↓ | 0.0000 | **4.7996** | 8.9864 |
| Overall Diversity → | 7.9887 | **8.0127** | 8.1430 |

Table 29: Comparison of generation quality (FID) across each subset of AMASS (Part 1).

| Dataset | BiBo | DiP | Motions |
|---|---|---|---|
| ACCAD | **4.5684** | 10.2789 | 263 |
| BMLhandball | **4.8302** | 9.0686 | 885 |
| BMLmovi | **4.7213** | 10.7736 | 1910 |
| BioMotionLab_NTroje | **4.5839** | 9.2667 | 4323 |
| CMU | **4.7682** | 8.8602 | 4129 |
| CNRS | **10.4036** | 10.6496 | 79 |
| DFaust_67 | **4.9707** | 8.5679 | 138 |
| DanceDB | **5.6532** | 7.2548 | 1266 |

| Dataset | BiBo | DiP | Motions |
|---|---|---|---|
| EKUT | **3.5976** | 9.5299 | 332 |
| Eyes_Japan_Dataset | **5.4948** | 7.8679 | 2311 |
| GRAB | **4.6874** | 9.9674 | 1840 |
| HUMAN4D | **5.2315** | 7.9081 | 468 |
| HumanEva | **7.2032** | 10.7560 | 59 |
| KIT | **4.9822** | 10.2390 | 5545 |
| MOYO | **4.6147** | 7.8556 | 1171 |
| MPI_HDM05 | **5.7478** | 8.1308 | 945 |
| MPI_Limits | **5.5547** | 8.9683 | 139 |
| MPI_mosh | **5.0530** | 13.0224 | 125 |
| SFU | **6.0829** | 9.0587 | 113 |
| SOMA | **4.5367** | 9.6309 | 143 |
| SSM_synced | **11.2186** | 13.5802 | 24 |
| TCD_handMocap | **7.0114** | 10.4370 | 53 |
| TotalCapture | **4.7316** | 7.4846 | 259 |
| Transitions_mocap | **4.3569** | 10.8853 | 125 |
| WEIZMANN | **4.9802** | 9.0565 | 3435 |

Table 31: Motion generation quality comparison of different methods on BABEL dataset with 263-dim motion representation. According to the results, BiBo demonstrates advantages in generation quality and diversity.

| Method | Continuity | FID ↓ | M. Score ↓ | R.P.3 ↑ | R.P.5 ↑ | R.P.10 ↑ | Diversity → |
|---|---|---|---|---|---|---|---|
| Ground Truth | - | 0 | 5.6281 | 0.3188 | 0.4467 | 0.6474 | 8.0907 |
| Double Take (MDM) | ✓ | 13.6735 | 8.4137 | 0.1721 | 0.2691 | 0.4483 | 6.8614 |
| MotionLCM v2 | ✗ | 7.8812 | 5.1991 | 0.3739 | 0.5003 | 0.6970 | **7.9657** |
| CLoSD | ✓ | 11.2371 | 7.1783 | 0.1931 | 0.2908 | 0.4900 | 7.4086 |
| BiBo (ours) | ✓ | **5.7789** | **5.0965** | **0.3971** | **0.5269** | **0.7229** | 8.5351 |

## F.15 ADDITIONAL GENERATION QUALITY EVALUATION

### F.15.1 GENERALIZATION TO AMASS

**Setting.** To validate the generalization of our method, we test the motion generation quality on the AMASS dataset. To better demonstrate the generalization capability, we use the model trained on HumanML3D, and directly test it on AMASS without fine-tuning. Since the AMASS dataset lacks text annotations, we primarily evaluate FID and Diversity, representing motion quality and diversity.

**Results.** The results in the Tab. 28 demonstrate that BiBo exhibits generalization capability on the AMASS dataset. The generated motions and diversity are more aligned with the real distribution compared to DiP ↓ (FID 9.0→4.8).

We further test BiBo's FID on each subset of AMASS. The results in Tab. 29 and 30 show that BiBo demonstrates advantages across all subsets. Notably, the larger the dataset scale (containing richer real motions), the better the distribution similarity achieved by BiBo. These results demonstrate that BiBo is capable of generating motions with high realism and possess strong generalization ability.

### F.15.2 GENERALIZATION TO BABEL

**Setting.** We evaluate the text-to-motion generation performance of the proposed motion diffusion on 263-dim BABEL, using the model parameters trained on HumanML3D. During inference, upon reaching the starting frame of a transition, we switch the control signal to the next motion's prompt, and immediately generate the next motion segment.

**Comparison Methods.** Double Take is a segment composition-based method. DiP is a future prediction-based method. MotionLCM supports only fixed-length generation. Note that we use the officially released 263-dim MDM variant of Double Take. All comparison methods use the official released checkpoints trained on HumanML3D.

Table 32: Computational overhead during inference on personal laptop and workstation. The results show that the proposed model has low overhead with high inference speed.

| | GPU UTL ↓ | VRAM ↓ | CPU UTL ↓ | RAM ↓ | AITS ↓ |
|---|---|---|---|---|---|
| Personal Laptop | 83% | 2.5GB | 12.8% | 2.7GB | 0.066s |
| Workstation | 24% | 2.1GB | 10.4% | 1.3GB | 0.048s |

**Results.** The results in Tab. 31 show that BiBo achieves enhanced FID ↓ (7.9→5.8) and R-precision ↑ (0.37→0.39) on BABEL dataset. These results demonstrate that BiBo is capable of generating realistic human motions over long horizon.

### F.16    REAL-TIME PERFORMANCE

**Setting.** We additionally deploy and test our model on two platforms:

1. A consumer-grade personal laptop. The laptop is equipped with a 2.3GHz i7-11800H CPU, 32GB of RAM, and a 6GB RTX 3060 GPU, running on Windows 11.

2. A graphics workstation. The workstation is equipped an 3.2GHz i9 14900K CPU, 128GB of RAM, and a 24GB RTX 4090 GPU, running on Ubuntu 24.04.

We load the proposed motion diffusion model at float32 precision and perform 10,000 inference runs, averaging the performance metrics. We measure GPU utilization, GPU memory utilization, CPU utilization, RAM utilization, and inference time.

**Results.** The results in Tab. 11 show that the proposed method occupies only 2.1 GB VRAM and 1.3 GB RAM during inference. It runs at an AITS of 0.048s on RTX4090, while maintaining an AITS of 0.066s on a personal laptop. This indicates that BiBo can perform real-time inference with low resource consumption.

