# OpenReview forum: "Endowing GPT-4 with a Humanoid Body: Building the Bridge Between Off-the-Shelf VLMs and the Physical World"
_ICLR.cc/2026/Conference — ICLR 2026 Poster_

### Official Review · Reviewer_hAat · 2025-10-27

**Soundness:** 2
**Presentation:** 3
**Contribution:** 2
**Rating:** 4
**Confidence:** 4

**Summary:**

This paper proposes the BiBo framework, which aims to leverage off-the-shelf Vision-Language Models (VLMs) to control humanoid agents for interaction in open physical environments. The paper designs an embodied instruction compiler and a diffusion model-based motion executor to realize the translation from high-level instructions to low-level actions, and verifies their effectiveness on multiple tasks.

**Strengths:**

See Summary.

**Weaknesses:**

1.The core innovations of this study are "reducing data dependency via off-the-shelf VLMs" and "solving motion continuity issues using LDM (Latent Diffusion Model) + VAE (Variational Autoencoder)". However, the generalization ability in real physical environments has not been fully verified. BiBo’s interaction capability has not been validated in real-world scenarios (non-InfiniGen-generated simulated environments), making it impossible to prove its effectiveness under complex real conditions such as lighting changes and irregular objects.
2.The dataset used in this study has limitations, and its generalization is questionable. Experiments are only conducted on the HumanML3D dataset and InfiniGen-generated scenes, without validation on other public datasets (e.g., BABEL, AMASS). This makes it difficult to demonstrate the model’s generalization ability across a wider range of action types and scenes.
3.The evaluation metric system is incomplete. Although FID and R-Precision are used to evaluate motion quality, quantitative analysis of key dimensions in interaction tasks—such as "interaction naturalness" and "task coherence"—is omitted. It is recommended to supplement quantitative analysis of metrics like Interaction Accuracy and Task Coherence.
4.There is a lack of real-time performance analysis. Although the paper claims that BiBo supports real-time control (>20 Hz), it does not provide analysis of specific memory usage or GPU memory consumption. Performance benchmarks on typical hardware platforms should be provided to verify its deployment feasibility.
5.There is no systematic analysis of the method’s limitations. The paper does not discuss BiBo’s performance in extreme scenarios (e.g., occlusion, lighting changes) nor analyze the impact of VLM reasoning errors on the entire system. Such discussions should be supplemented in the main text or appendix.
6.In the ablation study (Table 2), control groups for "without Pose Reasoning (w/o Pose Reasoning)" and "without Joint Generation (w/o Joint Generation)" need to be added to quantify the impact of each stage on task success rates (e.g., positioning accuracy for sitting tasks, joint control accuracy for touching tasks) and clarify the necessity of the three-stage process.
7.It is recommended to supplement failure case analysis in the main text or appendix, including the category of failed tasks, root causes (e.g., pose recognition errors, insufficient collision handling), and potential improvement directions, to enhance the transparency of the research.

**Questions:**

See Weakness.

---

> ### Author Response · Authors · 2025-11-27
> **Ⅰ. Real-world Scenarios Experiment**
>
> Thank you for recognizing the "**effectiveness**" of our method across multiple tasks. Your comprehensive suggestions also greatly contribute to improve the quality of our paper.
> ## Ⅰ. Real-world Scenarios Experiment
> Our proposed method (BiBo) focuses on physical Human Scene Interaction (HSI) modeling, aiming to leverage off-the-shelf VLMs for humanoid agents. Following the **routine experimental setup [1~5]**, we conduct experiments on task completion and generation quality.
>
> The potential **downstream applications** of this approach extend to both computer animation and real-world humanoid robot control. While our motivation differs from robot operating systems, we agree that validating on real-world data or in real-world environments can offer valuable insights into the potential applications of the system.
> ### 1.1 Real-world Scaned Benchmark
> - **(1) Setting.** We additionally validate our method on real-world scanning scene, instead of those generated. Specifically, we use ScanNet[6]. ScanNet is a **real-world** scanned dataset containing static scene meshes, camera poses, and RGB/Depth images.
>
>     The evaluation is conducted on ScanNet ScenePlan[1], which  contains **simple, medium, and hard** interaction task sequences in ScanNet scenes, involving **multiple interaction categories**, such as sit, sleep, and touch. An interaction is considered successful when the character makes contact with the specified object part.
>
>     Specifically, we test each task three times. The object instances are extracted based on the segment labels in ScanNet. For each task, we extract its target objects from the original task JSON, and specify them in the prompt explicitly. Due to the low quality of mesh rendering, we use the original RGB/Depth images from ScanNet as VLM input.
> - **(2) Results.** The experimental results are in Tab. 1. BiBo shows improvement in success rates (avg **+49.7**) across tasks of varying difficulty, and it **generalizes well to real-world scene** data from ScanNet.
>
>     **Table 1.** Comparison of task success rate of UniHSI and BiBo on ScanNet ScenePlan dataset. The results show that BiBo achieves **plausible success rate** on real-world scaned scene.
>   |Method|Simple ↑|Middle ↑|Hard ↑|
>   -|-|-|-
>   UniHSI [ICLR.24]|73.2|43.1|22.3
>   BiBo (ours)|**81.7**|**65.2**|**41.5**
> For more details about the implementation, dataset distribution, and failure case analysis, please refer to Sec. B.3 and Sec. F.14.
> ### 1.2 Deploying to Real-world Robot
> - **(1) Challenge.** Deploying to a real-world robot requires additional processes such as SLAM and low-level joint motor development, which require significant time, resources, and close collaboration with the robot manufacturer. Unfortunately, due to **resource and time limitations**, it is challenging for us to deploy the full system in a real-world robotics scenario during the rebuttal period.
> - **(2) Setting.** We still additionally conduct **validations on the Unitree G1** platform, as well as **real-world experiment of our Embodied Instruction Compiler** with a physical humanoid robot. For further details, please refer to the supplementary material[7] and Sec. F.13 of the manuscript.
>
>     **Table 2.** Task success rates of BiBo on Unitree G1 platform. BiBo on robotic platform shows **comparable performance** with methods on virtual characters.
>   |Method|Body|Reach ↑|Sit ↑|Stand ↑|
>   -|-|-|-|-
>   CLoSD [ICLR.25]|SMPL[8] (*simpler*)|**100%**|88%| **98%**
>   BiBo (ours)|Unitree G1[9] (*complex*)|**100%**|**94%**|86%
>
>     **Table 3.** Success rate of the Unitree G1 using the proposed Embodied Instruction Compiler and the built-in motion controller in real-world scene interaction tasks. The results validate the **real-world generalization of our proposed Compiler**.
>   |Task|Reach ↑|Watch ↑|Sit ↑|
>   -|-|-|-
>   |LeVERB[10]* (2025)|20/20 | -/- |1/20
>   |HumanoidVLA[11]* (2025)|9/10|10/10|-/-
>   |BiBo (ours)|10/10|10/10|8/10
>
>     *The data are sourced from the original paper
> - **(3) Results.** The results in Tab. 2 show that, by replacing the portable tracking policy with that of the Unitree G1, the off-the-shelf VLM is able to control a humanoid robot to perform **diverse motion styles and scene interactions**.
>
>     Additionally, with the proposed Embodied Instruction Compiler, the off-the-shelf VLM demonstrates **plausible real-world generalization**, as shown in Tab. 3. It drives the robot to autonomously understand its surroundings, and perform action planning in real-world scenarios.
> - **(4) Potential.** We also closely monitor recent advances in imitation learning[12~14], which have shown **promising results in real-world tracking policies** environments. BiBo is **compatible** with these techniques.
>
>     With ongoing developments in humanoid robot manufacturing and low-level motion control techniques, we are confident that BiBo will demonstrate promising results in real-world robotics in the near future. These potentials are left for future work.

---

> ### Author Response · Authors · 2025-11-27
> **Ⅱ. Validation on a Broader Range of Publicly Available Datasets (Part 1)**
>
> ## Ⅱ. Validation on a Broader Range of Publicly Available Datasets
>
> ### 2.1 Evaluation on AMASS
>
> - **(1) Setting.** To validate the generalization of our method, we test the motion generation quality on the AMASS[15] dataset. To better demonstrate the generalization capability, we use the model trained on HumanML3D, and directly test it on AMASS **without fine-tuning**.
>
>     To the best of our knowledge, we are **the first** to validate motion generation quality using the **full AMASS dataset**. We download all AMASS subsets (including the latest subsets), prioritizing the SMPLH[16] format, and using the SMPLX[17] format when SMPLH is unavailable. Following the HumanML3D preprocessing pipeline, we first process all motion clips into joint coordinate sequences. Then, we remove prefixes and suffixes of motions (typically T-poses) based on their similarity to the initial/final frames. Finally, we segment long clips for evaluation pipeline adaptation.
>
>     Since the AMASS dataset lacks text annotations, we primarily evaluate *FID* and *Diversity*, representing motion quality and diversity.
>
> - **(2) Results.** The results in the Tab. 4 demonstrate that BiBo exhibits **generalization** capability on the AMASS dataset. The generated motions and diversity are more aligned with the real distribution compared to DiP (**FID ↓ 9.0->4.8**).
>
>     We further test BiBo's FID on each subset of AMASS. The results in Tab. 5 show that BiBo demonstrates advantages across all subsets. Notably, the larger the dataset scale (containing richer real motions), the better the distribution similarity achieved by BiBo. These results demonstrate that BiBo is capable of generating motions with **high realism** and possess **strong generalization** ability.
>
>     **Table 4.** Comparison of generation quality (FID) and diversity on the AMASS dataset. '→' indicates that closer to ground truth is better. The results show that BiBo generates motions with plausible FID without fine-tuning, and approaches the real world motion distribution.
>
>     | Dataset      | *Ground Truth**                 | BiBo (ours)   | DiP [ICLR 2025]
>     |---------------|----------------|----------------|----------------|
>     | Overall FID ↓    |     0.0000     | **4.7996**     | 8.9864     |
>     | Overall Diversity →  |   7.9887         | **8.0127**     | 8.1430     |
>
>     **Table 5.** Comparison of generation quality (FID) across each subset of AMASS. For reference, the HumanML3D test set contains 3,955 motions. The results show that BiBo demonstrates **advantages on every subset**, and datasets containing more real motions exhibit greater similarity to the motion distribution generated by BiBo.
>
>     | Dataset                       | BiBo (ours)   | DiP [ICLR.25 Spotlight]   | # Num of Motions  |
>     |-------------------------------|----------------|-------------------|------------|
>     | ACCAD| **4.5684**         | 10.2789           | 263        |
>     | BMLhandball| **4.8302**         | 9.0686            | 885        |
>     | BMLmovi| **4.7213**         | 10.7736           | 1910       |
>     | BioMotionLab_NTroje| **4.5839**         | 9.2667            | 4323       |
>     | CMU| **4.7682**         | 8.8602            | 4129       |
>     | CNRS| **10.4036**        | 10.6496           | 79         |
>     | DFaust_67| **4.9707**         | 8.5679            | 138        |
>     | DanceDB| **5.6532**         | 7.2548            | 1266       |
>     | EKUT| **3.5976**     | 9.5299            | 332        |
>     | Eyes_Japan_Dataset            | **5.4948**         | 7.8679            | 2311       |
>     | GRAB| **4.6874**         | 9.9674            | 1840       |
>     | HUMAN4D| **5.2315**         | 7.9081            | 468        |
>     | HumanEva| **7.2032**         | 10.7560           | 59         |
>     | KIT| **4.9822**         | 10.2390           | 5545       |
>     | MOYO| **4.6147**         | 7.8556            | 1171       |
>     | MPI_HDM05| **5.7478**         | 8.1308            | 945        |
>     | MPI_Limits| **5.5547**         | 8.9683            | 139        |
>     | MPI_mosh| **5.0530**         | 13.0224       | 125        |
>     | SFU| **6.0829**         | 9.0587            | 113        |
>     | SOMA| **4.5367**         | 9.6309            | 143        |
>     | SSM_synced| **11.2186**        | 13.5802       | 24         |
>     | TCD_handMocap| **7.0114**         | 10.4370           | 53         |
>     | TotalCapture| **4.7316**         | 7.4846            | 259        |
>     | Transitions_mocap| **4.3569**         | 10.8853           | 125        |
>     | WEIZMANN| **4.9802**         | 9.0565            | 3435       |

---

> ### Author Response · Authors · 2025-11-27
> **Ⅱ. Validation on a Broader Range of Publicly Available Datasets (Part 2)**
>
> ### 2.2 Evaluation on BABEL
>
> - **(1) Preliminaries.** Arbitrary-length motion generation methods can be roughly divided into two types:
>     - ① *Segment Composition*. They first generate all key motion segments, then connect them by generating transitions. However, this approach may have limitations for motions exceeding the segment length limit, and cannot change control signals in real-time. It is commonly evaluated on the BABEL[18~19] dataset using **135-dim** motion format.
>
>     - ② *Future Extension*. They predict the next segment based on the previous one, repeating this process to ensure continuous motion of arbitrary length, with the ability to dynamically change control conditions. It is more suitable for building humanoid agent, and commonly evaluated on the HumanML3D[20] dataset using **263-dim** motion format.
>
>     Our method belongs to the second category. Due to the **different motion representation**, we **cannot directly test** on the existing BABEL benchmark.
>
> - **(2) Setting.** To perform evaluation on BABEL, we follow the segmentation method in TEACH, and refer to HumanML3D's preprocessing scripts. We construct a **20 fps 263-dim BABEL** dataset. Specifically, we first clean invalid motions such as T-poses before and after all motion sequence. Then, we segment motions into transition-action clips following [21]. Clips shorter than 40 frames are dropped, while those longer than 200 frames are truncated.
>
>     The text annotations in BABEL are brief and homogeneous. We use additional templates to enhance the text diversity. Finally, the words in the annotated text may not match the vocabulary of the text embedding in evaluation pipeline. We establish a mapping table from annotated text words to vocabulary words based on similarity search, and use an LLM to fix incorrect mappings. The R-precision is calculated using text embeddings computed with the mapped words.
>
>     During inference, upon reaching the starting frame of a transition, we switch the control signal to the next motion's prompt, and immediately generate the next motion segment.
>
> - **(3) Comparison Methods.** Double Take[21] is a segment composition-based method. DiP[2] is a future prediction-based method. MotionLCM[22] supports only fixed-length generation. Note that we use the officially released 263-dim MDM variant of Double Take.
> All comparison methods use the official released checkpoints trained on HumanML3D.
>
>
>     **Table 6.** Motion generation quality comparison of different methods on BABEL dataset with 263-dim motion representation. According to the results, BiBo **demonstrates advantages** in generation quality and diversity.
>
>     Method |Continuity | FID ↓| Matching Score ↓ | R.P.3 ↑ | R.P.5 ↑ | R.P.10 ↑ | Diversity →
>     -|-|-|-|-|-|-|-
>     *Ground Truth** | -| 0 | 5.6281 |0.3188|  0.4467 |  0.6474 | 8.0907
>     Double Take (MDM variant)[ICLR.24]  | √| 13.6735 | 8.4137 | 0.1721 | 0.2691 | 0.4483  |6.8614
>     MotionLCM v2 [2024.12]|× | 7.8812 | 5.1991  |0.3739 | 0.5003 | 0.6970 | **7.9657**
>     CLoSD [ICLR.25]|√ | 11.2371 | 7.1783 | 0.1931 | 0.2908 | 0.4900 | 7.4086
>     BiBo (ours)|√ | **5.7789** | **5.0965** | **0.3971** | **0.5269** | **0.7229** | 8.5351
>
>
> - **(4) Results.** The results show that BiBo achieves enhanced **FID ↓ (7.9->5.8)** and **R-precision ↑ (0.37->0.39)** on BABEL dataset. These results demonstrate that BiBo is capable of generating realistic human motions over long horizon.

---

> ### Author Response · Authors · 2025-11-27
> **Ⅲ. Evaluation on Additional Metrics (Part 1)**
>
> ## Ⅲ. Evaluation on Additional Metrics (e.g., Interaction Naturalness)
>
> We agree that using more comprehensive quantitative evaluations could further demonstrate the effectiveness of our method, and we have tested interaction accuracy following your suggestion. However, after extensively examining relevant literature [1\~5, 23\~30], we were unable to find clear mathematical definitions for "interaction naturalness" and "task coherence."
>
> From our understanding, naturalness can be measured by the *distribution difference* (*FID*) with real-world motions, while task coherence can be evaluated using *task success rate* and *text recall precision* (*R-Precision*). We should further evaluate these metrics in **scene interaction**, instead of standalone motion generation If there are any inaccuracies in our understanding, we welcome further clarification.
>
> ### 3.1 Interaction Accuracy
>
> - **(1) Setting.** While the *Contact Error* proposed by UniHSI provides valuable insights, it remains **limited in accurately reflecting interaction accuracy**. Specifically, Contact Error is defined as follows:
>
>     $$
>     \\mathrm{Contact\\ Error} = \\sum\_{i,c_i\neq 0}e_i/\sum_{i,c_i\neq 0}1, \qquad
>     $$
>     $$
>     e\_i =
>     \\begin{cases}
>         || \\mathbf{d}\_{k\_i} ||,  & c\_i = \\mathrm{contact} \\\\
>         \\mathrm{min}(0.3-|| \\mathbf{d}\_{k\_i} ||,0).  & c\_i = \\mathrm{not\\ contact}
>     \\end{cases}
>     $$
>
>     $c_i$ represents an action, and $\mathbf{d}_{k_i}$ denotes the shortest distance from the joint to the surface point cloud of the target object. $c_i$ includes navigation waypoint, whose errors $e_i$ are predominantly 0. When the number of navigation waypoints increases, the contact error decreases, despite no actual improvement in interaction accuracy.
>
>     Therefore, we propose **Interaction Error**, which is more closely related to the interaction, excluding the interference of navigation waypoints:
>
>     $$
>     \mathrm{Contact\ Error} = \sum_{i, c_i = \mathrm{contact}} e_i / \sum_{i, c_i = \mathrm{contact}} 1, \qquad
>     e_i = || \mathbf{d}_{k_i} ||
>     $$
>
>     We test the *Interaction Error* of three frequently used joints — pelvis, left hand, and right hand — during contact interactions. Non-contact interaction accuracy evaluation can be referred in Sec. F.6, which achieves high control accuracy.
>
>     The experiments are conducted on interactions related to the specified joints. Pelvis is tested with *sit* and *sleep* task, while hands are tested with *touch* task.
>
>     **Table 7.** Comparison of interaction error between different methods. BiBo demonstrates **precise interaction capability** even without specifically RL-learning for contact tasks.
>
>     | Method | Pelvis ↓ | Left Hand ↓ | Right Hand ↓ |
>     |- |-|-|-|
>     |UniHSI[ICLR.24]|0.126 | 0.181 | 0.094 |
>     |BiBo (ours) | 0.109 | **0.079** | **0.080** |
>     |BiBo (ours, w/o IK)| **0.107** | 0.101 | 0.099
>
> - **(2) Results.** According to the experimental results in Tab. 8, BiBo outperforms UniHSI, which is specifically designed for contact tasks, in interaction accuracy. There are two reasons:
>     1. IK effectively improves the interaction accuracy of end effectors.
>
>     2. BiBo uses a Tracking Policy, which dynamically adjusts actions to achieve higher precision in interactions.
>
>     We also observe that the RL-learning in UniHSI led to lower interaction accuracy for the left hand compared to the right hand. BiBo uses motion diffusion trained on mirror augmented data, achieving high interaction accuracy for **both the left and right hands**.

---

> ### Author Response · Authors · 2025-11-27
> **Ⅲ. Evaluation on Additional Metrics (Part 2)**
>
> ## Ⅲ. Evaluation on Additional Metrics (e.g., Interaction Naturalness)
> ### 3.2 Interaction Naturalness
> - **(1) Setting.** *FID* quantifies the similarity between the generated motion distribution and the real-world motion distribution, thereby capturing the naturalness of interactions. Following the setting in SIMS, we assess the naturalness across three interaction categories: Sit, Lie, and Carry. In contrast to direct motion generation approaches, we acquire these motion samples through **physics-based interactions** in simulated scenes, where object initial positions are randomly sampled and interaction plans are generated according to predefined rules. Only successfully executed interaction sequences are retained for evaluation.
>
>     **Table 8.** Comparison of motion naturalness (FID) between different methods. The performance of comparison methods is reported in [23].
>     The results show that BiBo's design **effectively enhances motion naturalness**.
>
>     | Method | Sit ↓ | Lie ↓ | Carry ↓ |
>     | -| -| -|-
>     | InterPhys[Sig.23] | - | - | 81.00|
>     | UniHSI[ICLR.24] | 153.84 | 211.22 | -
>     | SIMS[CVPR.25] | 125.66 | 171.24 | 65.14 |
>     | BiBo (ours)| **68.78** | **134.46** | **34.72**
>
>
> - **(2) Results.** The results indicate that BiBo achieves lower **FID ↓** (avg **-37.8%**), demonstrating higher similarity to real-world motion. This improvement can be attributed to the proposed motion diffusion model. The diffusion is trained on real-world data, which effectively enhances interaction naturalness.
>
>     For more details, please refer to Sec. F.11.2.
>
>
> ### 3.3 Interaction Coherence
> - **(1) Setting.** Task Coherence is assessed via *success rate* and text *R-Precision*.
>
>     - ① *Success rate* validates whether the agent follows instructions to interact with the specified object, and accomplish the task objective (e.g., *whether the carried object reaches the target coordinates*).
>
>         The implementation of task success rate evaluation is detailed in Sec. 4.1. We evaluate average success rates across 6 interaction types. For methods incapable of completing all interaction categories, we report the mean success rate over their supported categories.
>
>     - ② *R-Precision* quantifies the probability of retrieving the correct motion within a data batch, given a textual motion caption. For interaction motions, *R-Precision* indicates whether the execution conforms to the task description; for example, Carry should involve holding the object between both hands rather than pushing it.
>
>         To evaluate *R-Precision*, we collect motions from physics-based interaction processes. Since *R-Precision* computation requires batches containing diverse motion categories, we randomly substitute one sample in each real-world data batch with an interaction motion, and perform retrieval using the corresponding task instruction.
>
>     **Table 9.** Comparison of interaction task success rates across different methods. The success rate reflects the task coherence of each method. According to the results, BiBo **demonstrates task coherence**.
>     | Method | HumanVLA | UniHSI | TokenHSI | CLoSD | BiBo (ours) |
>     -|-|-|-|-|-
>     Num of Task Type ↑| 2 | 4 | 4  | **6** | **6**
>     Success Rate ↑| 50.74%| 82.26%|  62.26% | 55.92% | **90.17%** |
>
>     **Table 10.** Comparison of R-precision for scene interactions across different methods. Higher R-precision indicates better alignment with the instructions, demonstrating stronger coherence. According to the results, BiBo **exhibits instruction coherence during interaction**.
>     | Method | Sit ↑ | Lie ↑ | Carry ↑ |
>     | - | - | - | - |
>     | UniHSI | 17.12% | 9.59% |  - |
>     | BiBo (ours) | **25.34%** | **17.81%** | **21.23%** |
>
>
> - **(2) Results.** According to the results in Tab. 9 and Tab. 10, BiBo achieves high task success rates while generating interaction motions consistent with instructions. This demonstrates that the proposed method possesses **high task coherence**.

---

> ### Author Response · Authors · 2025-11-27
> **Ⅳ & Ⅴ. Real-time Performance Analysis & Failure Case Analysis**
>
> ## Ⅳ. Detailed Real-time Performance Analysis (e.g., GPU Memory Consumption)
>
> - **(1) Setting.** We additionally deploy and test our model on two platforms:
>     1. A *consumer-grade personal laptop*. The laptop is equipped with a 2.3GHz i7-11800H CPU, 32GB of RAM, and a 6GB RTX 3060 GPU, running on Windows 11.
>
>     2. A *graphics workstation*. The workstation is equipped an 3.2GHz i9 14900K CPU, 128GB of RAM, and a 24GB RTX 4090 GPU, running on Ubuntu 24.04.
>
>     We load the proposed motion diffusion model at *float32* precision and perform 10,000 inference runs, averaging the performance metrics. We measure GPU utilization, GPU memory utilization, CPU utilization, RAM utilization, and inference time.
>
>     **Table 11.** Computational overhead during inference on personal laptop and workstation. The results show that the proposed model has **low overhead** with **high inference speed**.
>     |  | GPU UTL ↓ | VRAM ↓ | CPU UTL ↓ | RAM ↓ | AITS ↓ |
>     -|-|-|-|-|-
>     Personal Laptop| 83% | 2.5GB | 12.8% | 2.7GB | 0.066s
>     Workstation    | 24% | 2.1GB | 10.4% |  1.3GB | 0.048s
>
>
> - **(2) Results.** The results in Tab. 11 show that the proposed method occupies only **2.1 GB VRAM** and 1.3 GB RAM during inference. It runs at an AITS of **0.048s** on RTX4090, while maintaining an AITS of 0.066s on a personal laptop. This indicates that BiBo can perform **real-time inference with low resource consumption**.
>
> ## Ⅴ. Systematic Failure Case and Limitation  Analysis & Future Work
>
> ### 5.1 Failure Case Analysis
>
> We provide an in-depth failure case analysis in the Sec. F.9 and Sec. F.14, including multiple representative failure cases. Here is a brief summary:
>
> - **(1)** For **Embodied Instruction Compiler**, existing VLMs (whether off-the-shelf or fine-tuned on scene data) suffer from incorrect understanding of multi-object spatial relationships or **hallucination** phenomena.
>
>     **Table 12.** Impact of VLM capability on BiBo's scene interaction ability, evaluated by task success rate in random generated scenes. The results show that larger and more capable models **effectively enhances** BiBo.
>     | Method | GPT-4o | GPT-4o mini | Qwen2.5-VL | Sonnet 3.5 |
>     -|-|-|-|-
>     Scale ↑ | **~200B** | ~8B | ~72B | ~175B|
>     Task Success Rate ↑ | **90.17%** | 59.54% | 78.65% | 84.41% |
>
>     Nonetheless, experimental results in Tab. 12 demonstrate that the error level effectively decreases with improvements in VLM capabilities. BiBo can integrate with most of the state-of-the-art or even future VLM platforms. We are positive that BiBo will possess **stronger capabilities** in the foreseeable future.
>
> - **(2)** **Motion Executor** currently relies on passively receiving physical feedback, **lacking active environmental perception**. The generated motions exhibit high consistency with instructions, but insufficient consistency with the scene, showing limitations in generating interaction motions for complex geometries and agile obstacle avoidance.
>
>     Currently, techniques such as height maps[26] and basis point set[4] are widely applied. These techniques have the **potential to be integrated** with BiBo, thereby enhancing the autonomy of low-level control.
>
> ### 5.2 Limitations and Future Work
>
> We have added Limitations and Future Work in Sec. 5 in the main text.
>
>     - First, our executor is trained on a text-to-motion dataset of limited size, which may restrict its generalization capability. With the availability of larger-scale motion datasets, there is potential to further enhance robustness and generalization.
>
>     - Second, while our model incorporates environmental feedback through motion execution results, explicitly modeling environmental geometry—such as height maps or basis point set features—remains an important direction for future exploration.
>
>     - Third, we focus on human–scene interactions in this paper, but there is potential to extend our framework to broader interaction modes, such as hand–object interactions and human–human interactions.
>
>     We leave these directions to future studies.

---

> ### Author Response · Authors · 2025-11-27
> **Ⅵ. Additional Ablation for Clarifing the Necessity of Three-Stage Process (Part 1)**
>
> ## Ⅵ. Additional Ablation for Clarifing the Necessity of Three-Stage Process
>
> To improves planning robustness and mitigates VLM hallucinations, BiBo divides the planning process into three steps (attribute analyzing, pose reasoning, joint generation).
>
> We design ablation studies on these steps ot validate the effectiveness of the three-stage design. Specifically, we evaluate the overall task success rate of the system, and the planning accuracy of the compiler.
>
> ### 6.1 Impact of Three-Stage Process on Task Success Rate
>
> For ablation, we remove voting in the first stage, and prompt the VLM to directly output standing location, facing direction and pixel coordinate in the second and third stage.
>
> **Table 13.** Impact of attribute analysis (A.A.), pose reasoning (P.R.) and joint generation (J.G.) on the task success rate. Evaluated under random generated scenes. The results show that all three components effectively **improve task success rate**.
> | Method | BiBo (ours) | BiBo (ours, w/o A.A.) |  BiBo (ours, w/o P.R.) |  BiBo (ours, w/o J.G.) |
> -|- |-|-|-|
> | Task Success Rate ↑ | **90.17%** | 87.23% | 32.75% | 79.95% |
>
> The results in Tab. 13 show that attribute analysis and joint generation effectively improve planning accuracy (**+3.0%** and **+10.2%**, respectively). Pose reasoning employs discretized preset coordinates and step-by-step reasoning to simplify coordinate output, which plays a crucial role in the VLM scene localization (**+57.4%**).
>
> ### 6.2 Relation between Three-stage Process and Interaction Types
>
> We further investigate the impact of Pose Reasoning stage and Joint Generation stage on each task.
>
> **Table 14.** Impact of Pose Reasoning (P.R.) and Joint Generation (J.G.) on each interaction type. The results show that pose reasoning with label enhancement is **essential** for all tasks, while joint generation affects sit, sleep, and lift tasks that rely on it.
> |Method|Reach ↑|Watch ↑|Sit ↑|Sleep ↑|Touch ↑|Lift ↑|
> |-|-|-|-|-|-|-
> |BiBo (ours, w/o P.R.)|53.13%(-46.05%)|35.03%(-64.59%)|28.61%(**-67.23%**)|32.67%(-62.22%)|36.88%(-49.17%)|10.20%(-55.22)|
> |BiBo (ours, w/o J.G.)|98.91%|98.87%|84.18%(-11.66%)|62.67%(**-32.22%**)|84.75%|50.34%(-15.08%)
> |BiBo (ours)|99.18%|99.62%|95.84%|94.89%|86.05%|65.42%|
>
> The results in Tab. 14 show that Pose Reasoning is **critical for all interaction types**, especially for *watch* and *carry* which require facing the object, and *sit* and *sleep* which require standing in front of the sofa/bed while facing away. Joint Generation primarily affects motions that require *joint-level control* (especially *head & pelvis for sleep*). Note that touch mainly interacts with small objects and the joints can be directly localized through object coordinates, without requiring the VLM to select anchor points on object surfaces.

---

> ### Author Response · Authors · 2025-11-27
> **Ⅵ. Additional Ablation for Clarifing the Necessity of Three-Stage Process (Part 2)**
>
> ## Ⅵ. Additional Ablation for Clarifing the Necessity of Three-Stage Process
>
> ### 6.2 Relation between Three-stage Process and Interaction Types
>
> We further investigate the impact of Pose Reasoning stage and Joint Generation stage on each task.
>
> **Table 14.** Impact of Pose Reasoning (P.R.) and Joint Generation (J.G.) on each interaction type. The results show that pose reasoning with label enhancement is **essential** for all tasks, while joint generation affects sit, sleep, and lift tasks that rely on it.
> |Method|Reach ↑|Watch ↑|Sit ↑|Sleep ↑|Touch ↑|Lift ↑|
> |-|-|-|-|-|-|-
> |BiBo (ours, w/o P.R.)|53.13%(-46.05%)|35.03%(-64.59%)|28.61%(**-67.23%**)|32.67%(-62.22%)|36.88%(-49.17%)|10.20%(-55.22)|
> |BiBo (ours, w/o J.G.)|98.91%|98.87%|84.18%(-11.66%)|62.67%(**-32.22%**)|84.75%|50.34%(-15.08%)
> |BiBo (ours)|99.18%|99.62%|95.84%|94.89%|86.05%|65.42%|
>
> The results in Tab. 14 show that Pose Reasoning is **critical for all interaction types**, especially for *watch* and *carry* which require facing the object, and *sit* and *sleep* which require standing in front of the sofa/bed while facing away. Joint Generation primarily affects motions that require *joint-level control* (especially *head & pelvis for sleep*). Note that touch mainly interacts with small objects and the joints can be directly localized through object coordinates, without requiring the VLM to select anchor points on object surfaces.
>
> - **(3) Planning Accuracy.**  We annotate 100 ground truth planning samples based on the scene dataset, thereby independently evaluating the compiler's capability. In the ablation experiments, we systematically remove: 1) voting mechanism from the Attribute Analyzing, 2) labels and associated text descriptions from Pose Reasoning, and 3) label grid from Joint Generation.
>
>     Since images in the scene dataset are relatively idealized, we additionally introduce complex real-world data to test the effectiveness of the label grid. Specifically, we use the MS-COCO real-world image dataset with instance segmentation annotations. A planning is considered correct if the label selected by VLM falls within the designated segment. We use GPT-4o mini in this experiment.
>
>     **Table 15.** Impact of voting, label, and grid enhancement on three-stage planning accuracy. Attribute Analyzing and Pose Reasoning are evaluated on 100 manually annotated plans, while Joint Generation is evaluated on MS-COCO. The results **demonstrate the effectiveness** of these designs.
>     | Stage | Attribute Analyzing ↑ | Pose Reasoning ↑  | Joint ↑  Generation |
>     -|-|-|-
>     w/o | 78% | 69% | 42.2% |
>     w/ | **86%** | **91%** | **65.8%** |
>
>     As shown in Tab. 15, Voting and Label enhance planning accuracy by 8% and 22% respectively, whereas Label Grid boosts the large model's accuracy by 23.6% on real-world data. These results validate the **effectiveness of our proposed three-stage design**.
>
> ---
>
> We hope these responses adequately address your concerns. Please don't hesitate to contact us if you have any further questions or need additional clarification.
>
> ---
>
> [1] Xiao et al. UniHSI. ICLR 2024
>
> [2] Tevet et al. CLoSD. ICLR 2025
>
> [3] Liang et al. TokenHSI. CVPR 2025
>
> [4] Xu et al. HumanVLA. NIPS 2024
>
> [5] Hassan et al. InterPhys. Siggraph 2023
>
> [6] Dai et al. ScanNet. CVPR 2017
>
> [7] https://huggingface.co/Behavia/BEHAVIA
>
> [8] Loper et al. SMPL. TOG 2015
>
> [9] https://www.unitree.com/g1/
>
> [10] Xue et al. LeVERB. 2025
>
> [11] Ding et al. HumanoidVLA. 2025
>
> [12] Yin et al. Unitracker. 2025.07
>
> [13] Zhao et al. ResMimic. 2025.10
>
> [14] Chen et al. GMT. 2025.06
>
> [15] Mahmood et al. AMASS. ICCV 2019
>
> [16] Romero et al. MANO. Siggraph 2019
>
> [17] Pavlakos et al. SMPLX. CVPR 2019
>
> [18] Punnakkal et al. BABEL. CVPR 2021
>
> [19] Athanasiou et al. TEACH. 3DV 2022
>
> [20] Guo et al. HumanML3D. CVPR 2022
>
> [21] Tevet et al. PriorMDM. ICLR 2024
>
> [22] Dai et al. MotionLCM. ECCV 2024
>
> [23] Wang et al. SIMS. CVPR 2025
>
> [24] Yi et al. TESMO. ECCV 2024
>
> [25] Hassan et al. SAMP. ICCV 2021
>
> [26] Cen et al. G3DT. CVPR 2024
>
> [27] Li et al. ZeroHSI. 2024
>
> [28] Jiang el al. Trumans. CVPR 2024
>
> [29] Hassan et al. POSA. CVPR 2021
>
> [30] Li et al. GenZI, CVPR 2024

---

### Official Review · Reviewer_V4s2 · 2025-10-31

**Soundness:** 4
**Presentation:** 4
**Contribution:** 3
**Rating:** 6
**Confidence:** 5

**Summary:**

This paper introduces BiBo (Building humanoId agent By Off-the-shelf VLMs), a novel framework that leverages off-the-shelf Vision-Language Models (VLMs, such as GPT-4) to control humanoid agents. The core idea is to reduce data collection and training costs by combining pre-trained VLMs with a tailored embodied system for humanoid control. BiBo consists of two main components:

- Embodied Instruction Compiler: Converts high-level natural language commands into low-level structured motor commands by reasoning over scene context.
- Diffusion-based Motion Executor: Generates smooth, human-like motion trajectories while dynamically adapting to environmental feedback using a combination of Latent Diffusion Models (LDMs) and Inverse Kinematics (IK) optimization.
The authors highlight BiBo's ability to perform diverse and complex physical interactions in dynamic environments, achieving a task success rate of 90.2% and improving motion execution precision by 16.3% compared to prior methods.

**Strengths:**

- VLM Agent Workflow for Complex Task Understanding

The Embodied Instruction Compiler is well-designed, using a structured three-step reasoning process (attribute analysis, pose reasoning, and joint generation) to translate high-level commands into low-level motor instructions. This design allows BiBo to accurately interpret user intent and adapt to complex tasks in dynamic physical environments, such as sitting, lifting objects, or interacting with multiple scene elements. The use of voting mechanisms and multi-view representations further improves the system's robustness in understanding intricate tasks.

- Novel Integration of CLoSD + IK for Diffusion Motion Updates

The combination of CLoSD (a physics-based motion-tracking policy) and Inverse Kinematics for refining humanoid motion is inspiring. The framework dynamically corrects motion trajectories by incorporating physical feedback from the environment, ensuring smooth and continuous motion even in challenging scenarios (e.g., collisions, external forces). This joint optimization approach enhances both the adaptability and precision of motion generation, particularly for tasks requiring fine-grained control (e.g., grasping or touching objects). The use of Latent Diffusion Models (LDMs) further enables the generation of high-fidelity motions while maintaining computational efficiency. This part could be considered the most inspiring in this paper.

**Weaknesses:**

- Unclear Execution of Motion with CLoSD for Dynamic Objects

The paper lacks clarity on how the generated motion trajectories are passed to CLoSD for execution. For instance, when an object moves unpredictably, does the system rely on CLoSD alone for tracking, or does it dynamically update the motion plan using feedback? What if the dynamic object encounters collision with hands? While the authors mention incorporating physical feedback into motion updates, the explanation of how BiBo handles motion retargeting or re-planning in the presence of dynamic objects is insufficient. This aspect deserves more detailed discussion and evaluation.


- Limited Discussion and Citation of Related Work

The paper does not sufficiently discuss its approach to prior work in key areas, such as LLM planning, long-term task completion, or the use of diffusion models in HSI motion generation. For instance:

[1] SIMS: Simulating Stylized Human-Scene Interactions with Retrieval-Augmented Script Generation. ICCV2025

[2] Synthesizing Physically Plausible Human Motions in 3D Scenes. 3DV2024

[3] Generating Human Interaction Motions in Scenes with Text Control. ECCV2024

It is clear that TESMO[3] is exactly the type of previous approach this paper aims to compare against: it introduces discontinuity by conditioning on past generated rather than executed results.
At the very least, providing sufficient discussion and citations would improve my impression of this paper.

- Not complicated enough environments as claimed

Most environments in demo videos feature a single piece of furniture on flat ground, not challenging enough as claimed.

**Questions:**

Please see the weakness

---

> ### Author Response · Authors · 2025-11-25
> **Ⅰ & Ⅱ Design of Dynamic Object Interaction & Additional Related Works (Part 1)**
>
> Thank you for acknowledging our proposed compiler as "**well-designed**" and the executor as "**novel**". Your suggestions are also valuable in improving this paper.
>
> ## Ⅰ. Detailed Design of Dynamic Object Interaction
>
> Adapting to dynamic interaction is a key focus in the design of BiBo. We have incorporated this consideration into the design of the Planner, Motion Modeling, and Executor components. For more implementation details, please refer to Sec. C and Sec. D.
>
>  - **Embodied Compiler.** Unlike previous methods (e.g., UniHSI[1]) that plan before execution, BiBo performs **planning during execution** based on real-time scene states. This enables BiBo's planning results to dynamically adapt to object state changes.
>
>  - **Action Modeling.** We model each action **relative to an anchor object**. In each frame, BiBo calculates action control signals (e.g., position, rotation) based on the real-time transformation of the anchor, thereby adapting to the object's dynamic movement.
>
>  - **Motion Generator.** First, the proposed diffusion motion generator extends future motion from **on-the-fly action command and current environmental feedback**, enabling the generation of future motion in response to changes in the target object's state. Second, we apply **inverse kinematics at each frame**, guiding the end effectors (hands, legs) to continuously track the dynamic object. The results in Table 1 show that these designs (**IK+58.62, Diffusion+37.08**) effectively enable dynamic manipulation.
>
>     **Table 1.** Impact of diffusion motion generator design and inverse kinematics (IK) on dynamic object manipulation (using lift task) of BiBo. The results show that dynamic object manipulation depends on on-the-fly motion generation and IK optimization.
>     | Method | BiBo (ours)| BiBo (ours, w/o Diffusion Design)| BiBo (ours, w/o IK)|
>     -|-|-|-
>     Dynamic Interaction Success Rate ↑ | **65.42%** | 28.34% | 6.80% |
>
>
>
> ## Ⅱ. Discussion of Additional Related Works
>
> Thank you for recommending these references. We have added discussions and citations accordingly. For more details, please refer to Sec. A.
>
> We discuss these HSI methods from two perspectives: **tasks** and **methodologies**.
>
> ### 2.1 Physical and Non-Physical HSI Tasks
>
>
>
> - **Non-Physical HSI Methods (e.g. TESMO[2])** focuses on generating motions that are consistent with the scene. During execution, they directly set joint positions or rotations **without involving physical feedback**. In this way, the generated and the executed motions are the same, but **not physical plausible**. The quantitative evaluation of non-physical HSI involves generation quality, control accuracy, and scene consistency.
>
> - **Physical HSI Methods (e.g., SIMS[3], InterPhys[4], Ours)** focus on both motion-scene consistency and physical plausibility. During execution, they apply torques to the joi nts, and the final executed motion is **influenced by physical environmental feedback**, which is **physical plausible** may differ from the generated one.
> In addition to the non-physical evaluation, it also includes the task success rate.
>
> BiBo is a **physical** HSI system, while TESMO is **non-physical** method. We compare BiBo and TESMO on motion quality, physical plausibility, and control accuracy.
>
> **Table 2.** Comparison of BiBo and TESMO on motion quality (FID, R-Precision, Diversity), physical plausibility (Skating) and control accuracy (Goal Reaching Error and Orientation Error). BiBo demonstrates enhanced perfomance compared with TESMO.
> | Method | FID ↓| R-Precision ↑| Diversity →| Foot Skating ↓| Goal Reaching Error ↓| Orientation Error ↓
> -|-|-|-|- |-|-|
> Ground Truth | 0 | 0.514 | 9.503 | 0.21 | 0 | 0
> TESMO[2] | 20.465 | 0.376 | 6.415 | 0.56 | 0.1445 | 0.2410
> BiBo (ours) | **0.076** | **0.542** | **9.606** | 0.74 | **0.0246** | **0.1772**
> BiBo (ours, Phys) | 1.883 | 0.411 | 8.298 | **0.01** | - | - |
>
> The results in Tab. 2 demonstrates that BiBo effectively enhances motion quality (**20.465->0.076**) and control accuracy (**0.1445->0.0246**) comparied with TESMO.

---

> ### Author Response · Authors · 2025-11-25
> **Ⅱ. Discussion of Additional Related Works (Part 2)**
>
> ## Ⅱ. Discussion of Additional Related Works
>
>
> ### 2.2 Methods to Extend Future Motion
>
> - **Extending from static pose vs. dynamic motion.**  It is worth clarifying that the discontinuity in TESMO does not come from extending from the generated result (since TESMO is non-physical, meaning its generated and executed motions are actually identical).
>
>     TESMO introduces discontinuity because it inpaints future motion using **only 1-frame static pose**, lacking dynamic motion context such as style or velocity. In contrast, BiBo extends future motion from a **20-frames dynamic motion**, ensuring continuity in joint movement and overall motion style.
>
>     The results in Tab. 2 and Tab. 4 further show that dynamic context significantly improves motion quality compared with using only a static start pose.
>
> - **Extending from generated motion vs. executed motion.**  For non-physical generation, the generated motion and the executed motion are identical, so extending from the generated sequence is sufficient to ensure continuity.
>
>     For physical methods, however, the executed motion may deviate from the generated one due to physical feedback. Extending only from the generated motion loses environmental adaptability. Extending only from the executed motion introduces discontinuities between the previous and current generated results (as observed in CLoSD[5] and illustrated in Fig. 6).
>
>     As shown in Tab. 3, BiBo extends **both*** generated and executed motion, improving both motion continuity and environmental adaptability. For more details, please refer to Sec. 4.
>
>     **Table 3.** Impact of extending from generated and executed motion on task success rate, motion continuity, and motion quality. The results show that extending from executed motion provides stronger environmental adaptability, whereas extending from generated motion yields higher motion quality.
>     | Method | from Gen| from Exe| from Gen & Exe|
>     |-|-|-|-
>     | Task Success Rate ↑ | 77.51% | *88.05%* | **90.17%** |
>     | Motion Discontinuity ↓| **0.0370** | 0.0698 | *0.0379*|
>     | Motion Quality (FID ↓) | **1.414** | 2.312| *1.883*
>
> ### 2.3 RL Policy vs. Diffusion as Motion Executor for LLMs
>
> - **LLM for RL Policy Planning** offers **physically plausible** sence interaction and high success rate on specific tasks. Existing methods typically prompt LLM to select from a fixed skill set (e.g., SIMS[3]) or rely on a predefined action template (e.g., UniHSI[1]), which makes it **difficult to scale to more diverse motions** (e.g., dance, boxing).
>
> - **LLM for Diffusion Generator Planning** can synthesize a **wide variety** of high-fidelity motions. Beyond skill-set–based planning, recent approaches attempt to let the LLM generate motion VQ tokens (e.g., MoConVQ[6]) or textual motion caption (e.g., AvatarGPT[7]). However, **few methods consider the surrounding scene or support physical interaction**.
>
> SIMS uses LLM for RL Policy Planning based on a fixed skill set, which may be limited in **diversity and motion naturalness**.
>
> BiBo uses a VLM to guide a diffusion generator, and the generated motions are tracked by an RL policy. This method generates **diverse and natural** motions, while maintaining **scene awareness and physical plausibility**.
>
> **Table 4.** Comparison of executable skills between BiBo and SIMS. BiBo is capable of performing diverse motions.
>
> Method | Idle, Walk | Jump, Jog, Crawl | Sit, Lie | Punch, Kick | Lift, Carry | Watch, Pointing at |
> -|-|-|-|-|-|-|
> SIMS[3] | √| | √| | √| |
> BiBo (ours) | √| √| √| √| √| √|
>
> **Table 5.** Comparison of motion naturalness between SIMS and BiBo, which uses FID following SIMS. The performance of SIMS is reported in the original paper.
> The results show that BiBo's design effectively enhances motion naturalness.
>
> | Method | Sit ↓ | Lie ↓ | Carry ↓ |
> | -| -| -|-
> | SIMS (LLM→RL) | 125.66 | 171.24 | 65.14 |
> | BiBo (LLM→Diffusion→RL)| **68.78** | **134.46** | **34.72**
>
> As shown in Tab. 4, BiBo possesses a variety of interaction skills. The results in Tab. 5 further demonstrate that BiBo's design improves the naturalness of motion for LLM-based humanoid agents. Details can be found in Sec. F.11.2.

---

> ### Author Response · Authors · 2025-11-25
> **Ⅱ & Ⅲ. Additional Related Works (Part 3) & Evaluation Diversity and Complexity (Part1)**
>
> ## Ⅱ. Discussion of Additional Related Works
>
> ### 2.4 Limitations and Future Work
>
> BiBo extends future motion using a 20-frame prefix composed of both generated and executed motion, which ensures motion continuity. However, we are also aware that a 20-frame prefix may be **insufficient** for maintaining long-horizon continuity.
>
>  Increasing the prefix length is a promising direction for future work. We provide some preliminary experiments using longer context. Instead of simply concatenating all past motion, we follow the idea of FramePack and construct motion context at multiple frame rates to improve efficiency. Details can be found in Sec. F.12.
>
> **Table 6.** Impact of context length on the text alignment of the generated results, evaluated using R-Precision on HumanML3D. The results show that longer context improves text alignment, which also indicates better long-term continuity.
>  Motion Context Length | 1 frame | 20 frame | (20 + 40 + 80) frame  |
>  -|- |-| -|
>  Top-1 R-Precision ↑ | 0.423 | 0.542 | **0.564**
>
> The results in Tab. 4 show that longer context introduce better text consistency (**+4.1%**), which also indicates better long-term continuity.
>
> ## Ⅲ. The Diversity and Complexity of Randomly Generated Scenes and Tasks.
>
> The proposed evaluation pipeline employs InfiniGen to construct randomized objects and scenes, as well as introducing LLM-generated long-horizon tasks. This method offers greater task diversity and difficulty. Tab. 7 presents a comparison with the evaluation datasets used by previous methods. For more details or visualizations on the styles and difficulty distributions, please refer to Sec. 4.1 and Sec. B.1.
>
> **Table 7.** Comparison of the evaluation dataset* bewteen different HSI methods. The proposed evaluation method is diverse and scalable.
>
> | Method | Dataset | Scalablility |  |   |  Quantity ↑ |  |  | | | Difficulty ↑  || |
> -|-|-|-| -|-|-|-| - |- | -|-|-|
> |  | | **Object**|**Scene** |**Task** | **Obj. Type** | **Diff. Obj.** | **Layout Type**|**Diff. Scene** | **Diff. Task** | **Max Step** | **Inter. Type** | **Planning during Evaluation**|
> HumanVLA[8]| HITR |  |  | √ | 23|76 | 4 | 63 | 63 | 1 | 1 |
> UniHSI[1] | ScanNet[9] Subset | | | |  20 | 448 | 10 | 10 | 100 |  8* | 4*
> CLoSD[5] | Multitask| | || 3 | 3 | 3 |3| 4 | 1 | 4
> BiBo (ours) |Random Generated |  **√**| **√**| √| **73**| **6146**| **100** |**100**| **1527** | **15**| **6** |  **√** |
>
> *Train dataset is not included in the statistics. BiBo's scene interactions do not require a training scene dataset.
>
> **contact_pairs* for navigation is not included
>
> ### 3.1 Various Object Categories and Shapes
>
> - **Diversity.** The randomly generated scene contains **73 object categories**, each with **dozens of randomizable parameters** for style, appearance, and material (*e.g., sofa: footrest/armrest presence, cushion count, dimensions, color, and material*). Compared to using a fixed model library, this method provides more diverse object variations.
>
> - **Complexity.** For successful task execution, the Planner is required to identify diverse object poses and structures, and **generate accurate control parameters**, such as interaction positions and joint targets. The diversity in object categories and forms further requires the Executor to **generalize across object and interaction types**, presenting greater challenges than single-task benchmarks.
>
> ### 3.2 Random Scene Scale and Layout
>
> - **Diversity.** The size, shape, and layout of the scene are controlled by random parameters, which can generate different exterior wall **shapes** (*e.g., L-shaped, semi-circular*), object **placements** (cabinet object arrangements, sofa orientations), and **traversable area** (*wide or narrow*, see Sec. B.1.2). This method provides a more diverse and scalable benchmark compared to fixed scene layouts.
>
> - **Complexity.** The Planning system needs to handle **diverse object spatial relationships** and robustly perform safe path planning and control parameter generation (*e.g., if the left side of the bed is blocked, it needs to get on the bed from the right side*). The executor needs to handle **different interaction variants** (*e.g., touch targets at different heights*) and possess **physical scene adaptation** for long-distance locomotion in large complex scenes and multi-task interactions.

---

> ### Author Response · Authors · 2025-11-25
> **Ⅲ. The Diversity and Complexity of Randomly Generated Scenes and Tasks (Part 2)**
>
> ## Ⅲ. The Diversity and Complexity of Randomly Generated Scenes and Tasks
>
> ### 3.3 LLM Task Generation with Difficulty Levels
>
> - **Diversity.** The proposed evaluation tasks are constructed using **both rule-based programs and LLMs**, encompassing **diverse text instructions, target objects, and interaction modalities**. The constructed tasks include different interactions with the same object (sit/sleep on bed), the same type of interaction with different objects (sit on sofa/table), dynamic object interactions, simultaneous multi-object interactions, and various other forms.
> Details can be found in Sec. B.1.3.
>
> - **Complexity.** Previous methods typically provided detailed action command lists during evaluation. The proposed method **only provides natural language instructions**, requiring the planner to formulate specific action commands according to the scene. The tasks contain up to **15 steps** and **6 interaction types**, making them demanding for the executor's multi-task generalization and long-sequence interaction capabilities. The example task is shown in Lst. 1.
>
> ---
>
> Hope these responses could address your concerns. Should you have any additional questions or require further clarification, please feel free to reach out.
>
> ---
>
> [1] Xiao et al. Unified human-scene interaction via prompted chain-of-contacts. ICLR 2024.
>
> [2] Yi et al. Generating human interaction motions in scenes with text control. ECCV 2024.
>
> [3] Wang et al. Sims: Simulating stylized human-scene interactions with retrieval-augmented script generation. CVPR 2025.
>
> [4] Hassan et al. Synthesizing physical character-scene interactions. Siggraph 2023.
>
> [5] Tevet el al. Closd: Closing the loop between simulation and diffusion for multi-task character control. ICLR 2025.
>
> [6] Yao et al. Moconvq: Unified physics-based motion control via scalable discrete representations. TOG 2024.
>
> [7] Zhou et al. Avatargpt: All-in-one framework for motion understanding planning generation and beyond. CVPR 2024.
>
> [8] Xu et al. Humanvla: Towards vision-language directed object rearrangement by physical humanoid. NIPS 2024.
>
> [9] Dai et al. Scannet: Richly-annotated 3d reconstructions of indoor scenes. CVPR 2017.

---

### Official Review · Reviewer_U7bF · 2025-10-31

**Soundness:** 4
**Presentation:** 3
**Contribution:** 3
**Rating:** 6
**Confidence:** 4

**Summary:**

This paper proposes BiBo (Building humanoId agent By Off-the-shelf VLMs) — a framework that connects general-purpose Vision-Language Models (VLMs) like GPT-4 to humanoid control. The key idea is to use powerful pre-trained multimodal models to bypass costly humanoid-specific data collection and training. BiBo has two major components: 1. Embodied Instruction Compiler – translates high-level natural language instructions (e.g., “have a rest”) into structured, low-level control commands (e.g., sitting location, facing direction, joint targets) through a three-stage visual Q&A process. 2. Diffusion-based Motion Executor – a latent diffusion model (LDM) that generates continuous, physically-plausible humanoid motion conditioned on those commands and on real-time physical feedback.

**Strengths:**

1.	The idea of directly plugging an off-the-shelf VLM (GPT-4o) into a humanoid control pipeline is innovative. Avoids re-training large models by adding a lightweight compiler layer.
2.	The compiler–assembler analogy is clear and intuitive: the VLM acts like a “compiler” converting high-level language into structured commands, while the motion diffusion module serves as an “assembler” for physical actuation.
3.	The Latent Diffusion Model with joint decoding of executed and generated latents ensures temporal continuity and environmental awareness — addressing a major weakness in previous motion diffusion frameworks.
4.	The experiment is also comprehensive

**Weaknesses:**

1.	The motions shown in the video do not fully comply with physical laws. During interactions with objects, there are visible cases of hovering and penetration, which make it appear that the human keypoints are rule-based attached to the objects rather than physically constrained. The interactivity seems weaker compared with methods such as UniHSI.
2.	The motion generation in the video appears to heavily depend on the VLM’s outputs. However, the VLM tends to exhibit strong hallucination problems during grounding. It is unclear how the authors constrain the frame-to-frame consistency — both in the diffusion process and within the Embodied Instruction Compiler.
3.	During locomotion, the agent sometimes floats or teleports. Yet, in Table 3, BiBo is reported to outperform others on the skating and floating metrics. Providing a more detailed definition and evaluation standard for these metrics would make the results more convincing.

**Questions:**

See Weakness

---

> ### Author Response · Authors · 2025-11-24
> **Ⅰ. Compliance with Physical Laws in Generated Results**
>
> Thank you for recognizing our method as "**innovative**" and experiments as "**comprehensive**". Your suggestions are also valuable for improving this work.
>
> ## Ⅰ. Compliance with Physical Laws in Generated Results
>
> We appreciate your careful review of our supplementary material. We have also noticed some artifacts. Based on our investigation, this is caused by the **gap between rendering and simulation**.
>
> ### 1.1 Physical Law Adherence
>  The physical laws depend on the simulator (IsaacGym). We use the same simulator as UniHSI[1], and the same humanoid body configuration as in CLoSD[2]. This ensures that BiBo's **physical plausibility is consistent** with methods like UniHSI and CLoSD. Details can be found in Sec. E.
>
> **Table 1.** Physical plausibility comparison of methods using different simulators on HumanML3D. The results show that methods using the same simulator achieve **comparable physical plausibility**.
>
> Method | Simulator | Penetration ↓| Float ↓| Skate ↓
> -|-|-|-|-
> MoConVQ | Their own simulator | 0.25 | 32.0 | 0.29
> CLoSD | IsaacGym | 0.30 | **20.2** | **0.01**
> BiBo (ours) | IsaacGym | **0.19** | 20.6 | **0.01**
>
> We conduct comparison experiments with methods using simpler simulators (MoConVQ[3]). The results in Tab. 1 show that methods using the same simulator achieve comparable performance, while simpler simulators decrease physical plausibility.
>
> ### 1.2 Causes of Visual Artifacts in Video
>
> - **Collision Body Generation.** The collision bodies are generated based on the VHACD algorithm, which produces **collision bodies that slightly differ from the visual shapes**. This may cause visual penetration artifacts.
>
>     For more details, please refer to Sec. B.1.2. We have uploaded the example scene data, computed collision bodies, and collision body generation scripts.
>
> - **Visualization Platform.** We simulate in IsaacGym and export body positions and rotations to **re-render** the animations in Unity. The rendering process may introduce differences (e.g., floor height, motion blur) from the original simulation.
>
>     For more details, please refer to Sec. F.1. We have updated the repository with the raw exported motion data and visualization scripts.
>
> ### 1.3 Future Work to Improve Physical Plausibility
>
> To enhance physical law adherence, future directions include improving the precision of collision body generation or adopting a more rigorous physics simulation engine. We are actively working on migrating our approach to NVIDIA's latest IsaacSim[4] platform, which will enable broader platform compatibility and enhanced physical plausibility.

---

> ### Author Response · Authors · 2025-11-24
> **Ⅱ. Mitigating Hallucinations in VLM (Part 1)**
>
> ## Ⅱ. Mitigating Hallucinations in VLM (Part1)
>
> ### 2.1 Methods to Mitigate VLM Hallucinations
>
> - **Step-by-step Reasoning.** BiBo mitigates hallucinations by **dividing the planning process into three steps** (attribute analyzing, pose reasoning, joint generation). We further improve planning accuracy by employing enhancement techniques (e.g., Voting). Details can be found in Sec. 4.1, Sec. C.2 and Sec. F.4.
>
>     1. *Success Rate under Simulation Environment.* We design ablation studies to evaluate the impact of the three stage reasoning. Specifically, we remove voting in the first stage, and prompt the VLM to directly output standing location, facing direction and pixel coordinate in the second and third stage.
>
>
>         **Table 2.** Impact of attribute analysis (A.A.), pose reasoning (P.R.) and joint generation (J.G.) on the task success rate. Evaluated under random generated scenes. The results show that all three components effectively **improve task success rate**.
>         | Method | BiBo (ours) | BiBo (ours, w/o A.A.) |  BiBo (ours, w/o P.R.) |  BiBo (ours, w/o J.G.) |
>         -|- |-|-|-|
>         | Task Success Rate ↑ | **90.17%** | 87.23% | 32.75% | 79.95% |
>
>         The results in Tab. 2 show that attribute analysis and joint generation effectively improve planning accuracy (**+3.0%** and **+10.2%**, respectively). Pose reasoning employs discretized preset coordinates and step-by-step reasoning to simplify coordinate output, which plays a crucial role in the VLM scene localization (**+57.4%**).
>
>     2. *VLM Task Planning.* We additionally annotate 100 ground truth tasks to specifically test the impact of voting on planning accuracy.
>
>         **Table 3.** Impact of voting on planning success rate on 100 manual annotated task plans. The results show that voting mechanism reduces model hallucination.
>         | Method | w/ Voting | w/o Voting |
>         -|-|-
>         | Success Rate ↑ | **86%** | 78% |
>
>         The results in Tab. 3 demonstrate that **voting mitigates model hallucination** (**-36.4% hallucination** relatively). Ablations for the other two stages are discussed in the following section.
>
> - **Vision Enhancement.** Label annotations provide **explicit constraints** on the VLM's outputs, thereby reducing hallucinations. For Pose Reasoning, we preset a set of labels representing positions and orientations. For Joint Generation, we employ Label Grid for image enhancement.
>
>     We first test the impact of labels on task success rate. For Label Grid, we further evaluate its zero-shot capability on an MS-COCO subset with 500 samples, using GPT-4o mini.
>
>     **Table 4.** Impact of label vision enhancement on the success rate of BiBo on task completion under random generated scene, and planning on 100 manual annotated plans. The results show that label enhancement **improves both task and planning success rates**.
>
>     | Method | BiBo (ours, w/ Label) | BiBo (ours, w/o Label)
>     -|-|-
>     | Task Success Rate ↑ | **90.17%** | 67.64%
>     | Planning Success Rate ↑ | **91%** | 69%
>
>     **Table 5.** Impact of label grid on the localization capability of VLM, evaluated using GPT-4o mini on MS-COCO[5]. The results show that label grid effectively **enhances accuracy**.
>
>     | Method | w/ Label Grid | w/o Label Grid
>     -|-|-
>     | Accuracy ↑ | **65.8%** | 42.2%
>
>     The results in Tab. 4 show that label enhancement improves both task (**+23%**) and planning (**+22%**) success rates. Tab. 5 show that label grid enhances object localization accuracy by **+55.9%**, relatively.
> - **Compatible with Lastest VLMs.** The iteration of large language models is accompanied by the reduction of hallucinations. Our method is compatible with the latest and even future VLMs, leveraging the advancement of large models to further mitigate hallucination effects.
>
>     We evaluates BiBo when integrated with VLMs of different capabilites. VLM capabilities are evaluated by their parameter size (parameter scale information is obtained from [6]).
>
>     **Table 6.** Impact of VLM capability on the task success rate of BiBo in random generated scenes. The results show that larger and more capable models achieve higher task success rates, indicating **reduced susceptibility to hallucination**.
>     | Method | GPT-4o | GPT-4o mini | Qwen2.5-VL | Sonnet 3.5 |
>     -|-|-|-|-
>     Scale ↑ | **~200B** | ~8B | ~72B | ~175B|
>     Task Success Rate ↑ | **90.17%** | 59.54% | 78.65% | 84.41% |
>
>      Results in Tab. 6 show that **stronger models improve task success rate**. This indicates that as VLMs advance, BiBo's susceptibility to VLM hallucination will further decrease.

---

> ### Author Response · Authors · 2025-11-24
> **Ⅱ. Mitigating Hallucinations in VLM (Part 2)**
>
> ## Ⅱ. Mitigating Hallucinations in VLM (Part2)
> ### 2.2 Frame-to-frame Consistency
>
> - **Portable to Existing SLAM System.**
>  Humanoid systems can be roughly divided into: *SLAM → Planning → Execution*.
>      BiBo focuses on Planning (Compiler) and Execution (Executor), while being **portable** to recent SLAM systems. Most of these systems [7,8,9] enables frame-to-frame consistency (e.g. dynamic SLAM, object relocation).
>
> - **Object-Centric Coordinates in Action Planning.** In action planning, we model the motion based on the **local coordinate system of an anchor object** (identified by a unique instance ID). The output motion commands are updated at each frame according to the current object state, maintaining **consistency with object movement**. For more details, please refer to Sec. C.
>
> - **Autoregressive Real-time Control in Motion Generation.**
> BiBo employs autoregressive motion diffusion and IK. Diffusion extends future motion from previous frames, ensuring frame-to-frame **motion consistency**. IK adjusts the end-effector trajectory at each timestep,
> ensuring **motion-object consistency** from frame to frame. For more details, please refer to Sec. D.

---

> ### Author Response · Authors · 2025-11-24
> **Ⅲ. Detailed Definition of Physical Plausibility Metrics**
>
> ## Ⅲ. Detailed Definition of Physical Plausibility Metrics (e.g., Float)
>
> Following InterPhys[10] and CLoSD[2], we use penetration, float, and skate to assess physical plausibility. The corresponding code has been uploaded to the repository (generator/mld/models/metrics/pm.py). Details can be found in Sec. B.3.
>
> ### 3.1 Penetration
>
> Penetration measures the **depth of ground penetration** in generated motions. For each frame $t$ in sequence $i$, we compute the lowest joint height $h\_{\\min}\^{(i,t)} = \\min\_j \\mathbf{J}\_{i,t,j,y}$, where $\mathbf{J}_{i,t,j,y}$ is the $y$-coordinate (height) of joint $j$ at frame $t$ in sequence $i$. The penetration depth for frame $t$ is:
>
> $$
> p_{i,t} = \max(0, -(h_{\min}^{(i,t)} + \tau)) \times 1000,
> $$
>
> where $\tau = 0.005$ m is a tolerance threshold, and the result is converted from meters to millimeters. The overall penetration metric is the mean over all frames:
>
> $$
> \text{Penetration} = \frac{1}{\sum_i L_i} \sum_{i=1}^{N} \sum_{t=1}^{L_i} p_{i,t},
> $$
>
> where $L_i$ is the length of sequence $i$. Lower penetration values indicate better physical plausibility.
>
>
> ### 3.2 Float
>
> Float measures the **floating height of characters** above the ground. Similar to penetration, we compute the lowest joint height $h_{\min}^{(i,t)}$ for each frame. The floating distance for frame $t$ is:
>
> $$
> f_{i,t} = \max(0, h_{\min}^{(i,t)} - \tau) \times 1000,
> $$
>
> where $\tau = 0.005$ m is the tolerance threshold. The overall float metric is:
>
> $$
> \text{Float} = \frac{1}{\sum_i L_i} \sum_{i=1}^{N} \sum_{t=1}^{L_i} f_{i,t}.
> $$
>
> Lower float values indicate that characters are closer to the ground, which is more physically plausible.
>
> ### 3.3 Skate
>
> Skate measures **foot sliding artifacts during ground contact**. For each consecutive frame pair $(t, t+1)$ in sequence $i$, we identify the contact joint $j\^*\_t = \\arg\\min\_j \\mathbf{J}\_{i,t,j,y}$ at frame $t$ (the joint with the lowest height). If both frames $t$ and $t+1$ have the contact joint below the contact threshold $\\tau = 0.005$ m, we compute the horizontal sliding distance.
>
> we abbreviate $\\mathbf{J}\_{i,m+1,j\^*\_t,[x,z]}$ as $\\mathbf{J}\_{m}$):
>
> $$
> s\_{i,t} = \begin{cases}
> \|\\mathbf{J}\_{t+1} - \\mathbf{J}\_{t}\|\_2 \\times 1000 & \\text{if } \\mathbf{J}_{t} \\leq \\tau \\text{ and } \\mathbf{J}\_{t+1} \\leq \\tau \\\\
> 0 & \\text{otherwise}
> \end{cases},
> $$
>
>
> where $\mathbf{J}_{i,t,j^*_t,[x,z]}$ denotes the $x$ and $z$ coordinates of the contact joint, and the result is converted from meters to millimeters. The overall skate metric is the mean over all consecutive frame pairs:
>
> $$
> \text{Skate} = \frac{1}{\sum_i (L_i - 1)} \sum_{i=1}^{N} \sum_{t=1}^{L_i-1} s_{i,t}.
> $$
>
> Lower skate values indicate less foot sliding, which is more physically plausible. However, motions such as jumping **inevitably introduce floating**, and an excessively low level of floating (e.g., <10) may indicate that the model generates lower-quality motions.
>
> ---
>
> We hope these responses address your concerns. Please feel free to reach out if you have any further questions or require additional clarifications.
>
> ---
>
> [1] Xiao et al. Unified human-scene interaction via prompted chain-of-contacts. ICLR 2024.
>
> [2] Tevet et al. Closd: Closing the loop between simulation and diffusion for multi-task character control. ICLR 2025.
>
> [3] Yao et al. Moconvq: Unified physics-based motion control via scalable discrete representations. TOG 2024.
>
> [4] https://developer.nvidia.com/isaac/sim
>
> [5] Lin et al. Microsoft coco: Common objects in context. ECCV 2014.
>
> [6] Abacha et al. Medec: A benchmark for medical error detection and correction in clinical notes. 2024.
>
> [7] Hu et al. DyGS-SLAM: Real-time accurate localization and gaussian reconstruction for dynamic scenes. ICCV 2025.
>
> [8] Zhu et al. Living scenes: Multi-object relocalization and reconstruction in changing 3d environments. CVPR 2024.
>
> [9] Li et al. MegaSaM: Accurate, fast and robust structure and motion from casual dynamic videos. CVPR 2025.
>
> [10] Hassan et al. Synthesizing physical character-scene interactions.  SIGGRAPH 2023.
>
> [11] Our anonymous repository: https://huggingface.co/Behavia/BEHAVIA

---

### Official Review · Reviewer_u3aD · 2025-10-31

**Soundness:** 3
**Presentation:** 3
**Contribution:** 3
**Rating:** 6
**Confidence:** 3

**Summary:**

The paper targets the interesing problem of humanoid agents which can handle flexible and diverse interactions in open environments. To avoid collecting massive dataset to train the model, the paper presents a new solution by utilizing the capability of the strong VLMs and a diffusion-based motion generator. The former is used to generate the primitive commands based on the user instruction and the latter is used to generate the corresponding motions. Experiments show the promising results of the proposed algorithm.

**Strengths:**

* The idea of utlizing the strong capability of VLM to decompose the primitive commands and later handled by a motion generator is interesting.

* The experiments show the proposed algorihtm obtains promising results in the challenging problem of interaction with the open environments.

* The paper is well presented and the proposed algorithm should be easy to reproduce.

**Weaknesses:**

* The paper relies on two components to handle the target problem. On one hand, currently, even the SOTA vlms may not be able to produce the precise primitive actions. To simply the problem, the paper presents a set of predefined actions but it still cannot guarantee a robust results. On the other hand, assume vlms can produce the accurate action motions, how to obtain a good motion is not a trivial task. It should provide more justification that why the presented motion executor can produce the desired results.

* The experimental results in Table 1 is evaluated based on a setting proposed by this paper. The task as well as the setups are introduced in this paper. How about the generation of the proposed algorithm to other benchmarks?

* For the results in Table 2, the result of "Lift" is much lower than other categories. What are the potential reasons for this?

* Currently, BiBo are operated in the virtual setting. Is it possible to proivde some evaluations which can show that the proposed algorithm can generate to real-cases like robots?

**Questions:**

Please address the questions in the weakness section. More specifically, please mainly address the questions related work the experiments.

---

> ### Author Response · Authors · 2025-11-26
> **Ⅰ. Improving the Robustness of VLM Results (Part 1)**
>
> Thank you for acknowledging that our method is "**interesting**" and shows "**promising results**". We also appreciate your constructive feedback to improving this paper.
>
> ## Ⅰ. Improving the Robustness of VLM Results
>
> ### 1.1 Three-stage Reasoning Process
>
> In the proposed Embodied Instruction Compiler, we enhance the robustness by **dividing the planning process into three steps** (attribute analyzing, pose reasoning, joint generation). We further improve planning accuracy by employing enhancement techniques (e.g., voting). For more details, please refer to Sec. 4.1, Sec. C.2 and Sec. F.4.
>
> - **Impact of three-stage reasoning on task accomplishment.** We design ablation studies to evaluate the impact of the three stage reasoning on planning robustness. Specifically, we remove voting in the first stage, and prompt the VLM to directly output standing location, facing direction and pixel coordinate in the second and third stage.
>
>
>     **Table 1.** Impact of attribute analysis (A.A.), pose reasoning (P.R.) and joint generation (J.G.) on the task success rate. Evaluated under random generated scenes. The results show that all three components effectively **improve robustness in task accomplishment**.
>     | Method | BiBo (ours) | BiBo (ours, w/o A.A.) |  BiBo (ours, w/o P.R.) |  BiBo (ours, w/o J.G.) |
>     -|- |-|-|-|
>     | Task Success Rate ↑ | **90.17%** | 87.23% | 32.75% | 79.95% |
>
>     The results in Tab. 1 show that attribute analysis and joint generation effectively improve planning accuracy (**+3.0%** and **+10.2%**, respectively). Pose reasoning employs discretized preset coordinates and step-by-step reasoning to simplify coordinate output, which plays a crucial role in the VLM scene localization (**+57.4%**).
>
> - **Impact of three-stage reasoning on planning.** We additionally annotate 100 ground truth tasks to specifically test the impact of voting on planning accuracy.
>
>     **Table 2.** Impact of voting on planning success rate on 100 manual annotated task plans. The results show that voting mechanism reduces model hallucination.
>     | Method | w/ Voting | w/o Voting |
>     -|-|-
>     | Success Rate ↑ | **86%** | 78% |
>
>     The results in Tab. 2 demonstrate that **voting mitigates model hallucination** (**-36.4% hallucination** relatively). Ablations for the other two stages are discussed in the following section.
>
> ### 1.2 Visual Enhancement by Label
>  The **explicit output constraints and additional sptial information** provided by label can improve the robustness of VLM. For Pose Reasoning, we preset a set of labels representing positions and orientations. For Joint Generation, we employ Label Grid for image enhancement. Details can be found in Sec. C.2 and Sec. F.4.
>
> We first test the impact of labels on the robustness of task accomplishment. For Label Grid, we further evaluate its zero-shot capability on an MS-COCO[8] subset with 500 samples, using GPT-4o mini.
>
> **Table 3.** Impact of label vision enhancement on the success rate of BiBo on task completion under random generated scene, and planning on 100 manual annotated plans. The results show that label enhancement **improves both task and planning robustness**.
>
> | Method | BiBo (ours, w/ Label) | BiBo (ours, w/o Label)
> -|-|-
> | Task Success Rate ↑ | **90.17%** | 67.64%
> | Planning Success Rate ↑ | **91%** | 69%
>
> **Table 4.** Impact of label grid on the localization capability of VLM, evaluated using GPT-4o mini on MS-COCO. The results show that label grid effectively **enhances planning robustness**.
>
> | Method | w/ Label Grid | w/o Label Grid
> -|-|-
> | Accuracy ↑ | **65.8%** | 42.2%
>
> The results in Tab.3 show that label enhancement improves robustness in both task (**+23%**) and planning (**+22%**). Tab. 4 show that label grid enhances object localization accuracy by **+55.9%**, relatively.

---

> > ### Comment · Reviewer_u3aD · 2025-11-26
> >
> > Thanks for the rebuttal. The rebuttal well addressed most of my concerns.

---

> ### Author Response · Authors · 2025-11-26
> **Ⅰ & Ⅱ. Improving VLM Robustness (Part 2) & Producing Desired Motion by The Executor**
>
> ## Ⅰ. Improving the Robustness of VLM Results
> ### 1.3 Compatible with Lastest VLMs
>
> The proposed method is compatible with the latest and even future off-the-shelf VLMs, and iteration of VLMs is accompanied by the **enhancement of robustness**. Therefore, the planning robustness of BiBo has potential to further improve as VLMs advances.
>
> We evaluates BiBo integrated with VLMs of different capabilites, using parameter size as an indicator (parameter size information is obtained from [9]).
>
> **Table 5.** Impact of VLM capability on the task success rate of BiBo in random generated scenes. The results show that larger and more capable models achieve higher task success rates, indicating **enhanced robustness**.
> | Method|GPT-4o|GPT-4o mini|Qwen2.5-VL|Sonnet 3.5|
>  -|-|-|-|-
> Scale ↑ | **~200B** | ~8B | ~72B | ~175B|
> Task Success Rate ↑ | **90.17%**| 59.54%| 78.65%|84.41%|
>
> Results in Tab. 5 show that **stronger models improve robustness** in planning and task accomplishment. For more details, please refer to Sec. F.3.
>
> ## Ⅱ. Producing Desired Motion by The Executor
>
> Generating the desired motion relies on two key aspects: 1) motion representation modeling, and 2) motion generation model.
>
> We represent motions using structured action commands, which combine **abstract motion intent with concrete spatial condition**. Moreover, the proposed Executor fuses motion latents with multimodal control signals in the **latent space**, and applies IK optimization, further improving the controllability.
> For more details, please refer to Sec. D.
>
> ### 2.1 Motion Modeling in Structured Action Command
>
> Structured action commands include both abstract motion caption and concrete sptial control signals (e.g., joint position relative to an anchor object). The motion caption provides a **coarse framework** for a motion, indicating the overall style and full-body dynamic information. The spatial control signals specify **fine-grained goals** for a motion, including local joint targets and interaction details. The motion caption and spatial conditions jointly depict the key aspects of a motion, guiding the diffusion model to generate the desired result.
>
> We ablate the motion caption and sptial control signals in the structured action command to investigate their importance for generating desired motions. Specifically, we remove the motion caption, the facing direction, the moving speed, and the joint control signals (except for pelvis).
>
> **Table 6.** Impact of the motion caption and sptial control signals on generating desired motion, evaluated using R Precision, Control Error (MAE), and Task Success Rate. The results show that the motion caption and the spatial control signals are **both essential** for generating the desired motion.
>
> | Method | R-Precision ↑ | Control Error ↓| Task Success Rate ↑|
> -| - | -| -|
> | BiBo (ours, w/o mo. cap.) | 25.53% | 0.1774 | 53.81% |
> | BiBo (ours, w/o facing dir.) | 49.56%  |207.99 | 27.09% |
> | BiBo (ours, w/o speed) |50.19 % | 175.38| 63.25% |
> | BiBo (ours, w/o joint control) |  49.5% | 269.78 |50.76% |
> | BiBo (ours) | **54.2%**  | **0.0314**| **90.17%** |
>
> As shown in Tab. 6, the absence of the motion caption prevents the model from generating the specified style (R-precision), whereas removing the control signals causes the motion to fail to accomplish the task objectives (Task). This highlights the role of action command design for motion controllability.
>
> ### 2.2 Motion Generation by Latent Diffusion with IK Optimization
>
> - **Latent space** provides a **low-noise, aligned representations** for motion and multimodal control signals, promoting the generation of motions consistent with the control conditions.
>
>     **Table 7.** The impact of latent diffusion on motion generation performance, evaluated on HumanML3D. The results show that latent space provides **enhanced motion quality and controllability**.
>
>     | Method | FID ↓| R Precision ↑| Control Error ↓|
>     |-|-|-|-
>     | BiBo (ours, w/o LDM) | 0.238 | 0.467 | 0.0617
>     | BiBo (ours, w/ LDM) | **0.076** | **0.542** | **0.0314**
>
>     We conduct ablation studies to verify the impact of the latent space on controllability. Specifically, we evaluate motion-space diffusion. The results in Tab. 7 show that operating in the latent space effectively improves motion quality and controllability.
>
> - **IK** offers **precise end-effector control**, enabling the motion to accurately execute the intented task.
>
>     **Table 8.** The impact of IK on task success rate. The results show that IK is essential for generating **desired precise interaction**.
>
>     | Method | Success Rate ↑| | |
>     -|-|-|-|
>     | | Composite | Touch | Lift |
>     BiBo (ours, w/o IK)| 16.00 |48.94| 6.80 |
>     BiBo (ours, w/ IK)| **41.05** |**86.05** | **65.42** |
>
>     We evaluate the impact of IK on task completion. The results show that IK produces the desired precise interactions, improving success rates on tasks that require accurate control, such as touch and lift.

---

> ### Author Response · Authors · 2025-11-26
> **Ⅲ & Ⅳ. Generalization to Other Benchmarks & Difficulty Variation across Evaluated Tasks**
>
> ## Ⅲ. Generalization to Other Benchmarks
>
> We conduct experiments on ScanNet ScenePlan benchmark from UniHSI[1]. ScanNet ScenePlan consists of a subset of ScanNet scenes, which are **real-world scanned data** containing static scene meshes, camera poses, and RGB/Depth images. The benchmark includes **simple, medium, and hard** task sequences, involving **multiple interaction categories**, such as sit, sleep, and touch. An interaction is considered successful when the character makes contact with the specified object part.
>
> Specifically, we test each task three times. The object instances are extracted based on the segment labels in ScanNet. For each task, we extract its target objects from the original task JSON, and specify them in the prompt explicitly. Due to the low quality of mesh rendering, we use the original RGB/Depth images from ScanNet as VLM input.
>
> The experimental results in Tab. 9 show that BiBo achieves plausible success rates (avg **+49.7**) across tasks of varying difficulty, and it generalizes well to real-world scene data from ScanNet.
>
>
>
> **Table 9.** Comparison of task success rate of UniHSI and BiBo on ScanNet ScenePlan dataset. The results show that BiBo achieves **plausible success rate** on real-world scaned scene.
>
> | Method | Simple ↑ |  Middle ↑ |  Hard ↑  |
> | - | - | - | - |
> | UniHSI | 73.2 | 43.1 | 22.3 |
> | BiBo (ours) | **81.7** | **65.2** | **41.5** |
>
> For more details about the implementation, dataset distribution, and failure case analysis, please refer to Sec. B.3 and Sec. F.14.
>
> ## Ⅳ. Difficulty Variation across Evaluated Tasks (e.g., Lift and other tasks)
>
> ### 4.1 Task Complexity
> While other tasks interact with **static** objects, lift requires interaction with a **dynamic** object.
>
> - **Static interactions** involve only humanoid agent movement, and the control signals remain temporally consistent (i.e., target position are the same across time). The task is completed once the character reaches the target state (e.g., the pelvis reaches a specified coordinate).
>
> - In **Dynamic interactions**, both the humanoid agent and the target object may move, introducing **additional dimensions in the action space**. The agent is required to track the object’s state and adjust multi-joint actions in real time. The task is considered successful only when **both** the humanoid and the object reach their target states (e.g., the box is lifted above 0.25 m).
>
> Therefore, the lift task is **more challenging** than other tasks.
>
> To illustrate this, we set up a static-object experiment group where the object in lift remains static, and the task is considered successful once the humanoid makes contact with the object using both hands and then stands up. The results show that dynamic interactions are more challenging than interactions with static objects.
>
> **Table 10.** Comparison of the difficulty between static and dynamic object interactions, evaluated by the success rate of the lift task. The results show that dynamic object interactions have a lower success rate and are **more challenging**.
>
> | Task | Success Rate ↑|
> -|-
> Lift (Static) | **96.59**
> Lift (Dynamic) | 65.42
>
> ### 4.2 Policy Adaptability
>
> The task success rate typically depends on task-specific training. BiBo uses a tracking policy to follow the generated actions, and it is trained to follow the diffusion generated motion, rather than specifically picking up an item.
>
> Even without targeted training, BiBo still **demonstrates generalization** in the lift task compared to other methods (as in Tab. 11), while maintaining a high success rate in other tasks.
>
> **Table 11.** Comparison of different methods on success rate of lift, where BiBo demonstrates **generalization** in lift task.
>
> | Method | UniHSI[1]| HumanVLA[2] | TokenHSI[3] | CLoSD[4] | BiBo |
> -|-|-|-|-|-
> Success Rate ↑| 0.00% | 44.90% | 48.19% | 7.71% | **65.42%** |

---

> ### Author Response · Authors · 2025-11-26
> **Ⅴ. Real-world Experiment**
>
> ## Ⅴ. Real-world Experiment
>
> Our proposed method focuses on physical Human Scene Interaction (HSI) modeling, aiming to leverage off-the-shelf VLMs for humanoid agents. The potential **downstream applications** of this approach extend to both computer animation and real-world humanoid robot control. While our method's motivation differs from traditional robot operating systems, real-world validation can provide valuable insights into its downstream application potential.
>
> Deploying to a real-world robot requires additional processes such as SLAM and low-level joint motor development, which require significant time, resources, and close collaboration with the robot manufacturer. Unfortunately, due to **resource and time limitations**, it is challenging for us to deploy the full system in a real-world robotics scenario during the rebuttal period. But we still additionally conducted **validations on the Unitree G1** platform, as well as **real-world experiment of our Embodied Instruction Compiler** with a physical humanoid robot. For further details, please refer to the supplementary material[5] and Sec. F.13 of the manuscript.
>
> **Table 12.** Success rate of the Unitree G1 using the proposed Embodied Instruction Compiler and the built-in motion controller in real-world scene interaction tasks. The results validate the **real-world generalization of our proposed Compiler**.
>
> | Task | Reach ↑ | Watch ↑| Sit ↑|
> |- |-|-|-|
> |LeVERB* (2025) [6] | 20 / 20 | - / - |1 / 20
> |HumanoidVLA* (2025) [7] | 9 / 10 | 10 / 10 | - / -
> |BiBo (ours) | 10 / 10 | 10 / 10 | 8 / 10 |
>
> *The data are sourced from the original paper
>
> The results show that, by replacing the portable tracking policy with that of the Unitree G1, the off-the-shelf VLM is able to control a humanoid robot to perform **diverse motion styles and scene interactions**. Additionally, with the proposed Embodied Instruction Compiler, the off-the-shelf VLM demonstrates **plausible real-world generalization**. It drives the robot to autonomously understand its surroundings, and perform action planning in real-world scenarios.
>
> We also keep track of recent advances[10~12] in imitation learning, which show **promising results** in combining a motion generator with a **tracking policy under real-world environment**. With ongoing developments in humanoid robot manufacturing and low-level motion control techniques, we are confident that BiBo will demonstrate promising results in real-world robotics in the near future. These potentials are left for future work.
>
> ---
>
> Hope our responses and supplementary experiments adequately address your concerns. We welcome any further questions or discussion.
>
> ---
>
> [1] Xiao et al. Unified human-scene interaction via prompted chain-of-contacts. ICLR 2024
>
> [2] Xu et al. Humanvla: Towards vision-language directed object rearrangement by physical humanoid. NIPS 2024
>
> [3] Pan el al. Tokenhsi: Unified synthesis of physical human-scene interactions through task tokenization. CVPR 2025
>
> [4] Tevet et al. Closd: Closing the loop between simulation and diffusion for multi-task character control. ICLR 2025.
>
> [5] Our anonymous repository: https://huggingface.co/Behavia/BEHAVIA
>
> [6] Xue et al. Leverb: Humanoid whole-body control with latent vision-language instruction. 2025
>
> [7] Ding et al. Humanoid-vla: Towards universal humanoid control with visual integration. 2025
>
> [8] Lin et al. Microsoft coco: Common objects in context. ECCV 2014.
>
> [9] Abacha et al. Medec: A benchmark for medical error detection and correction in clinical notes. 2024.
>
> [10] Yin et al. Unitracker: Learning universal whole-body motion tracker for humanoid robots. 2025.07
>
> [11] Zhao et al. ResMimic: from general motion tracking to humanoid whole-body loco-manipulation via residual learning. 2025.10
>
> [12] Chen et al. GMT: General motion tracking for humanoid whole-body control. 2025.06

---

### Author Response · Authors · 2025-11-30
**Summary of Revision**

Dear Chairs and Reviewers,

We would like to thank all the reviewers for their careful and constructive feedback. Our motivation is to leverage **off-the-shelf VLMs** for humanoid control without costly **human-scene interaction** data collection. To achieve this, we propose:
1. a scene-aware **embodied instruction compiler** that translates high-level instructions into structured action commands,
2. a **diffusion motion executor** (akin to an *assembler*) that generates continuous, physically-plausible motions from commands and physical feedback.


We are delighted that our work is considered **innovative** (u3aD, U7bF, V4s2), **effective** (hAat, u3aD, V4s2) and **inspiring** (V4s2), and that the manuscript is **well presented** (u3aD) and **comprehensive** (U7bF). In accordance with the reviewers’ comments and suggestions, we have conducted approximately 30 new experiments with over 100 test cases, along with manuscript revisions. All updates and changes are highlighted in blue in the revised version.

The major changes in this revision lie in:

*Main Text*
- Section 4: Addition and reorganization of statistical figures and tables
- Section 5: Discussion of limitations and future work

*Appendices*
- Appendix A: Related works on HSI and arbitrary-length motion generation
- Appendix B: Statistics and implementation details of 5 datasets
- Appendix C & D: Detailed implementation of BiBo
- Appendix E: Implementation details of comparison methods
- Appendix F: Supplementary experiments
    - F.2: Comparison between different motion VAEs
    - F.3: Influence of VLM's capability on BiBo
    - F.4: Impact of three-stage VQA on planning accuracy
    - F.5: Ablation studies on action modeling
    - F.6: Detailed control accracy of BiBo
    - F.7: Selection of hyperparameters
    - F.8: Impact of special prefix on motion transition
    - F.9: Failure case analysis of BiBo
    - F.11: Additional comparison with HSI methods
        - Comparison with non-physical methods
        - Comparison with RL-based methods
    - F.12: Impact of the motion context length on continuity
    - F.13: Evaluation on real-world robotics platform
        - F.13.1: Migration to Unitree G1's body
        - F.13.2: Real-world deployment of Embodied Instruction Compiler
    - F.14: Additional scene interaction evaluation
        - F.14.2: Comparison of task success rate on ScenePlan
        - F.14.2: Comparison of interaction accuracy
        - F.14.4: Comparison of interaction coherence
    - F.15: Additional datasets to evaluate generalization
        - F.15.1: Motion quality on AMASS
        - F.15.2: Motion quality on BABEL
    - F.16: Evaluation of real-time performance of BiBo

Thank you once again for your contributions to refining our work and polishing this paper.

Yours sincerely,

Authors of Paper 4884 “Endowing GPT-4 with a Humanoid Body: Building the Bridge Between Off-the-Shelf VLMs and the Physical World”

---

### Meta-Review · Area_Chair_izWB · 2026-01-06

**Summary:**

This paper proposes BiBo, a framework enabling general-purpose VLMs (e.g., GPT-4) to control humanoid agents. It consists of an "Embodied Instruction Compiler" for planning and a "Motion Executor" based on latent diffusion. Reviewers found the approach innovative and the "compiler-assembler" analogy intuitive. The authors provided an exceptionally comprehensive rebuttal that addressed the primary concerns regarding domain generalization and real-world applicability.

**Reviewer Concerns:**

Addressed:

1) Real-World Validation & Datasets (Reviewers hAat, u3aD): The authors conducted extensive new experiments, including deployment on a physical Unitree G1 robot and validation on ScanNet (real-world scenes), AMASS, and BABEL. This effectively resolved the major criticism regarding reliance on synthetic data (InfiniGen).

2) VLM Robustness (Reviewers u3aD, U7bF): New ablation studies demonstrated that the proposed three-stage reasoning process significantly mitigates VLM hallucinations and improves planning accuracy.

3) Baselines & Metrics (Reviewers V4s2, U7bF): The authors added comparisons to requested baselines (TESMO, SIMS) and clarified physical metrics (skate/float).

Outstanding:

Visual Artifacts: Some visual inconsistencies (hovering/penetration) remain in the demo videos, attributed to the gap between the physics simulator and the visual renderer, though this does not fundamentally undermine the method's contribution.

**Reviewer Scores:**

Reviewer u3aD: 6 -> 7 (Explicitly stated concerns were addressed).

Reviewer U7bF: 6 -> 7 (Clarifications on physics and hallucinations provided).

Reviewer V4s2: 6 -> 7 (Dynamic interaction handling explained).

Reviewer hAat: 4 -> 6 (The addition of real-world datasets and robot experiments directly addressed the grounds for the initial low score).

---

### Decision · Program_Chairs · 2026-01-26

Accept (Poster)